# Abrupt excursions in water vapor isotopic variability at the Pointe Benedicte observatory on Amsterdam Island

Amaëlle Landais[1,*], Cécile Agosta[1,*], Françoise Vimeux[1,2], Olivier Magand[3], Cyrielle Solis[1], Alexandre Cauquoin[4], Niels Dutrievoz[1], Camille Risi[5], Christophe Leroy-Dos Santos[1], Elise Fourré[1], Olivier Cattani[1], Olivier Jossoud[1], Bénédicte Minster[1], Frédéric Prié[1], Mathieu Casado[1], Aurélien Dommergue[6], Yann Bertrand[6], Martin Werner[7]

[1] Laboratoire des Sciences du Climat et de l'Environnement, LSCE/IPSL, CEA-CNRS-UVSQ, Université Paris-Saclay, 91191 Gif-sur-Yvette, France

[2] HydroSciences Montpellier (HSM), UMR 5569 (UM, CNRS, IRD), 34095 Montpellier, France

[3] Observatoire des Sciences de l'Univers à La Réunion (OSU-R), UAR 3365, CNRS, Université de La Réunion, Météo France, IRD, 97744 Saint-Denis, La Réunion, France

[4] Institute of Industrial Science (IIS), The University of Tokyo, Kashiwa, Japan.

[5] Laboratoire de Météorologie Dynamique, Institut Pierre - Simon Laplace, Sorbonne Université / CNRS / École Polytechnique – IPP, Paris, France

[6] Univ. Grenoble Alpes, CNRS, INRAE, IRD, Grenoble INP[T], IGE, 38000 Grenoble, France ([T]Institute of Engineering and Management Univ. Grenoble Alpes)

[7] Alfred Wegener Institute, Helmholtz Centre for Marine and Polar Research, D-27570 Bremerhaven, Germany

* corresponding authors who contributed equally to the study: amaelle.landais@lsce.ipsl.fr and cecile.agosta@lsce.ipsl.fr

**Abstract**
In order to complement the picture of the atmospheric water cycle in the Southern Ocean, we
have continuously monitored water vapor isotopes since January 2020 on Amsterdam Island in
the Indian Ocean. We present here the first 2-year-long water vapor isotopic record on this site.
We show that the water vapor isotopic composition largely follows the water vapor mixing
ratio, as expected in marine boundary layers. However, we detect 11 periods of a few days
where there is a strong loss of correlation between water vapor $\delta^{18}O$ and water vapor mixing
ratio as well as abrupt negative excursions of water vapor $\delta^{18}O$. These excursions often occur
toward the end of precipitation events. Six of these events show a decrease in gaseous elemental
mercury suggesting subsidence of air from higher altitude.
Our study aims at further exploring the mechanism driving these negative excursions in water
vapor $\delta^{18}O$. We used two different models to provide a data-model comparison over this 2-year
period. While the European Centre Hamburg model (ECHAM6-wiso) at 0.9° was able to
reproduce most of the sharp negative water vapor $\delta^{18}O$ excursions hence validating the physics
process and isotopic implementation in this model, the Laboratoire de Météorologie
Dynamique Zoom model (LMDZ-iso) at 2° (3°) resolution was only able to reproduce 7 (1) of
the negative excursions highlighting the possible influence of the model resolution for the study
of such abrupt isotopic events. Based on our detailed model-data comparison, we conclude that
the most plausible explanations for such isotopic excursions are rain-vapor interactions
associated with subsidence at the rear of a precipitation event.

## 1. Introduction

The main sources of uncertainty in the atmospheric components of Earth System Models for future climate projections are associated with complex atmospheric processes, particularly those related to water vapor and clouds (Arias et al., 2021; Sherwood et al., 2014). Decreasing these uncertainties is of vital interest as the hydrological cycle is a fundamental element of the climate system because it allows, via the transport of water vapor, to ensure the Earth's thermal balance.

Stable water isotopes are a useful tool to study the influence of dynamical processes on the water budget at various spatial and temporal scales. They provide a framework for analyzing moist processes over a range of time scales from large-scale moisture transport to cloud formation, precipitation, and small-scale turbulent mixing (Bailey et al., 2023; Dahinden et al., 2021; Galewsky et al., 2016; Thurnherr et al., 2020).

The relative abundance of heavy and light isotopes in different water reservoirs is altered during phase change processes due to isotopic fractionation (caused by a difference in saturation vapor pressure and molecular diffusivity in the air and the ice). Each time a phase change occurs, the relative abundance of water vapor isotopes is altered. We express the abundance of the heavy isotopes D and $^{18}O$ with respect to the amount of light isotopes H and $^{16}O$, respectively, in the water molecules through the notation $\delta$:

$$\delta^{18}O = \left( \frac{\left( ^{18}O/_{16}O \right)_{Sample}}{\left( ^{18}O/_{16}O \right)_{VSMOW}} - 1 \right) \times 1000 \quad \text{(Eq. 1)}$$

$$\delta D = \left( \frac{\left( ^{D}/_{H} \right)_{Sample}}{\left( ^{D}/_{H} \right)_{VSMOW}} - 1 \right) \times 1000 \quad \text{(Eq. 2)}$$

where $(^{18}O/^{16}O)$ and $(D/H)$ represent the isotopic ratios of oxygen and hydrogen atoms in water and VSMOW (Vienna Standard Mean Ocean Water) is an international reference standard for water isotopes.

There are two types of isotopic fractionation: equilibrium fractionation, which is caused by the difference in saturation vapor pressure of different isotopes, and non-equilibrium fractionation, which occurs due to molecular diffusion (e.g. during ocean evaporation in undersaturated atmosphere or snowflakes condensation in oversaturated atmosphere). In the water vapor above the ocean, the proportion of non-equilibrium fractionation, and hence diffusive processes can

be estimated by the deuterium excess, a second order isotopic variable denoted d-excess,
defined as (Dansgaard, 1964):

$$\text{d-excess} = \delta D - 8 \times \delta^{18}O \qquad (Eq.3)$$

Over the recent years and thanks to the development of optical spectroscopy enabling
continuous measurements of water isotopes ratios in water vapor, an increasing number of
studies have focused on the use of water vapor stable isotopes to document the dynamics of the
water cycle over synoptic weather events, such as cyclones, cold fronts, atmospheric rivers
(Aemisegger et al., 2015; Ansari et al., 2020; Bhattacharya et al., 2022; Dütsch et al., 2016;
Graf et al., 2019; Lee et al., 2019;  Munksgaard et al., 2015; Tremoy et al., 2014) or water cycle
processes such as evaporation over the ocean or deep convection (Benetti et al., 2015; Bonne
et al., 2019). Several instruments have been installed either in observatory stations (e.g.
Aemisegger et al., 2012; Guilpart et al., 2017; Leroy-Dos Santos et al., 2020; Steen-Larsen et
al., 2013; Tremoy et al., 2012), on boat (e.g. Benetti et al., 2014; Thurnherr et al., 2019) or on
aircraft (Henze et al., 2022). In the aforementioned studies, the interpretation of the isotopic
records is often performed using a hierarchy of isotopic models, from conceptual models
(Rayleigh type) to general circulation models or regional weather prediction models equipped
with water isotopes (Ciais and Jouzel, 1994; Markle and Steig, 2022; Risi et al., 2010; Werner
et al., 2011). Such data comparisons enable one to test the performances of the models either in
the simulation of the dynamic of the atmospheric water cycle or in the implementation of the
water isotopes. Our study is part of these dynamics analyses and aims at improving the
documentation of climate and atmospheric water cycle in the Southern Indian Ocean, a region
which has been poorly documented until now.
Over the previous years, we have installed three water vapor analyzers on La Reunion Island at
the Maïdo observatory, 21.079°S, 55.383°E, 2160m (Guilpart et al., 2017) and in Antarctica
(Dumont d'Urville, 66,663°S, 140°E, 202m and Concordia, 75.1°S, 123.333°E, 3233m; Bréant
et al., 2019; Casado et al., 2016; Leroy-Dos Santos et al., 2021). These instruments have been
used for the following purposes. They document the diurnal variability of the isotopic signal
with the influence of the subtropical westerly jet on the water isotopic signal in night as well as
the cyclonic activity on La Réunion Island. In Antarctica, the records have shown a strong
influence of katabatic winds on the isotopic composition of water vapor (Bréant et al., 2019).
In order to complete the picture of the atmospheric water cycle over the Indian basin of the
Southern Ocean already measured by these three analyzers, we installed a new water vapor

isotopic analyzer at mid-latitude in the south Indian Ocean on Amsterdam Island (Figure 1) in November 2019. Amsterdam Island is one of the very rare atmospheric observatories in the southern hemisphere. Moreover, the south Indian Ocean is a significant moisture source for Antarctic precipitation, notably in the region encompassing Dumont d'Urville and Concordia stations (Jullien et al., 2020; Wang et al., 2020).

The objective of this study is to provide the first analyses of isotopic records (vapor and precipitation) on Amsterdam Island, with a comparison of meteorological data and environmental data collected in parallel on the Amsterdam Island Observatory (e.g. atmospheric mercury) to help with the interpretation of isotopic records. Indeed, previous studies have shown that gaseous elemental mercury decreases with increasing altitude in marine environment suggesting that gaseous elemental mercury can be used as a tracer of subsidence of air from the high altitude (e.g. Koening et al., 2023). This study includes analyses of meteorological maps, back trajectories as well as outputs from general circulation models equipped with water isotopes. After a description of the different records over the years 2020 and 2021, model simulations and back trajectories, we focus on some low-pressure events associated with a strong negative excursion of $\delta^{18}O_v$ over a few days and a decoupling between $\delta^{18}O_v$ and humidity. These events are then used for evaluation of atmospheric component of Earth system models equipped with water isotopes.

## 2. Methods

### 2.1 Site

Labelled as a global site for the Global Atmosphere Watch World Meteorological Organization, Amsterdam Island (37.7983° S, 77.5378° E) is a remote and very small island of 55 km$^2$ with a population of about 30 residents, located in the southern Indian Ocean at 3300 km and 4200 km downwind from the nearest lands, Madagascar, and South Africa, respectively (Sprovieri et al., 2016). Climate is temperate, generally mild with frequent presence of clouds (average total sunshine hours is 1581 hours per year over the period 1981 – 2010 from MeteoFrance data). Seasonal boundaries are defined as follows: winter from July to September and summer from December to February, in line with previous studies (Sciare et al., 2009). Average temperature is lower in winter compared to summer (10.5°C vs 15°C) while relative humidity and wind speed remain high (50-85% and 5 to 15 m s$^{-1}$ respectively) most of the year without a clear seasonal cycle.

Numerous atmospheric compounds and meteorological parameters are and were continuously
monitored at the site since 1960 (Angot et al., 2014; El Yazidi et al., 2018; Gaudry et al., 1983;
Gros et al., 1999, 1998; Polian et al., 1986; Sciare et al., 2000, 2009; Slemr et al., 2015; Slemr
et al., 2020). In particular, the Amsterdam (AMS) site hosts several dedicated atmospheric
observation instruments notably at the Pointe Bénédicte atmospheric observatory (70 m above
sea level) where greenhouse gases concentrations and mercury (Hg) are monitored. Hg species
have been continuously measured since 2012.

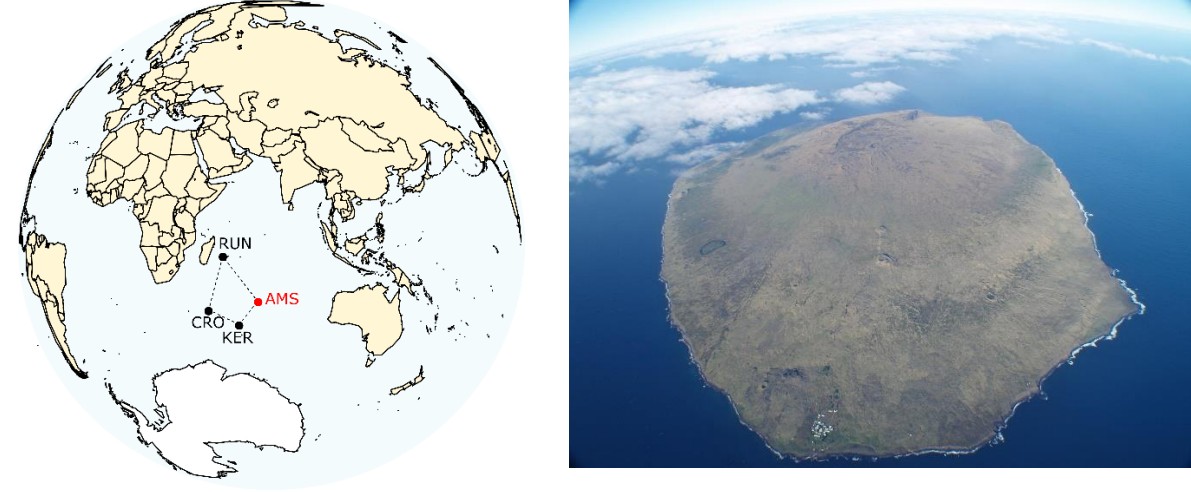

**Figure 1** : Location (left) and picture (right) of Amsterdam Island. CRO: Crozet Island;
RUN: La Réunion Island; KER: Kerguelen Island; AMS: Amsterdam Island.
Picture credit: left – from O. Magand adapted from Angot et al. (2016); right – photo
taken by O. Magand.

**2.2 Long term measurements**
2.2.1 Meteorological measurements

One meteorological station is installed at the top of an observation mast (25 m above ground
level, hence 95 m above sea level) at the Pointe Bénédicte observatory since 1980 (data used
during this study). Wind speed and direction, atmospheric pressure, air temperature and relative
humidity data are currently obtained at a minute resolution. Another meteorological station is
based on the island and is operated by Météo France at Martin-de-Viviès life base around 27 m
above sea level, about two kilometers east from the Pointe Bénédicte observatory collecting air
temperature, humidity, precipitation, wind speed and direction, pressure and solar radiation

2.2.2 Gaseous elemental mercury (GEM)

Atmospheric GEM (Gaseous Elemental Mercury) measurements have been conducted since
2012 in the framework of IPEV GMOStral-1028 observatory program at the Pointe Benedicte
atmospheric research facility (Magand and Dommergue, 2022). GEM is continuously measured
(15-minute data frequency acquisition) using a Tekran 2537 A/B instrument model (Angot et
al., 2014; Li et al., 2023; Slemr et al., 2015, 2020; Sprovieri et al., 2016). The measurement is
based on mercury enrichment on a gold cartridge, followed by thermal desorption and detection
by cold vapor atomic fluorescence spectroscopy (Bloom and Fitzgerald, 1988; Fitzgerald and
Gill, 1979). Concentrations are expressed in nanograms per cubic meters at standard
temperature and pressure conditions (273.15 K and 1013.25 hPa) with an instrumental detection
limit below 0.1 ng m$^{-3}$ and a GEM average uncertainty value around 10% (Slemr et al., 2015).
The instrument is automatically calibrated following a strict procedure adapted from that of
Dumarey et al. (1985). Ambient air is sampled at 1.2 L min$^{-1}$ through a heated (50°C) and UV
protected PTFE sampling line, with an inlet installed outside, 6 m above ground level (76 m
above sea level). The air is filtered through two 0.45 µm pore size polyether sulphone and one
PTFE (polytetrafluoroethylene) 47 mm diameter filters before entering in Tekran to prevent the
introduction of any particulate material into the detection system as well as to capture any
gaseous oxidized mercury or particulate bound mercury species ensuring that only GEM is
sampled. To ensure the comparability of mercury measurements around the world, the
instrument is operated according to the Global Mercury Observation System standard operating
procedures (Sprovieri et al., 2016; Steffen et al., 2012).
In this study, and even though long-range transport and a variable tropopause height may
modulate the signal, atmospheric GEM is used as potential tracer of stratosphere-to-troposphere
intrusion and/or subsidence of upper troposphere air (above 5-6 km) that may impact the
atmospheric records at the Pointe Benedicte Observatory where marine boundary layer air is
collected most of the time (Angot et al., 2014; Slmer et al., 2015, 2020; Sprovieri et al., 2016).
Mercury in the atmosphere consists of three forms: gaseous elemental mercury (GEM as
defined above), gaseous oxidized mercury and particulate-bound mercury. GEM, the dominant
form of atmospheric mercury, is ubiquitous in the atmospheric reservoir and originates from a
multitude of anthropogenic and natural sources (Edwards et al., 2021; Gaffney et al., 2014;
Gustin et al., 2020 ; Gworek et al., 2020). Near the surface (marine or terrestrial boundary layer)
and out of polar regions, gaseous oxidized mercury and particulate-bound mercury represent
only a few percent of the total atmospheric mercury (Gustin and Jaffe, 2010; Gustin et al., 2015;
Swartzendruber et al., 2006). Chemical cycling and spatiotemporal distribution of mercury in
the air is still poorly understood whatever atmospheric layer considered (surface, mixed or free
troposphere, stratosphere), and complete GEM oxidation schemes remain unclear (Shah et al.,
2021 and associated references). Still, several studies provided evidence that vertical
distribution of atmospheric mercury measurements from boundary layer to lower/upper
troposphere and stratosphere shows a decreasing trend in GEM concentration with increasing
altitude, in parallel with an increase in the concentration of divalent mercury resulting from
GEM oxidation mechanisms (Brooks et al., 2014; Fain et al., 2009; Fu et al., 2016; Koenig et
al., 2023; Lyman and Jaffe, 2012; Murphy et al., 2006; Swartzendruber et al., 2006, 2008; Sheu
et al., 2010; Talbot et al., 2007). The identification of such observational processes (lower
concentration of GEM  in high-altitude air masses compared to those in the marine boundary
layer ones) is used here to help characterize possible intrusions of high-altitude air masses at
the low altitude Pointe Benedicte observatory.

**2.3 Water vapor isotopic measurements**

The near-surface water vapor $\delta^{18}O$ and $\delta D$ (hereafter $\delta^{18}O_v$ and $\delta D_v$ expressed in ‰ versus
SMOW and enabling to calculate water vapor d-excess$_v$ as d-excess$_v$ = $\delta D_v - 8 \times \delta^{18}O_v$). The
water vapor mixing ratio ($q_v$ in ppmv) have been measured continuously since November 2019.
The measurements have been done with a Picarro Inc. instrument (L2130-i model) based on
wavelength-scanned cavity ring down spectroscopy. The instrument has been installed in a
temperature-controlled room at the Amsterdam Island observatory and the sampling of water
vapor is done outside at ~ 6 m above ground level (or 76 m above sea level) through a 5 m long
inlet tube made of PFA (perfluoroalkoxy alkanes) and heated at 40°C.

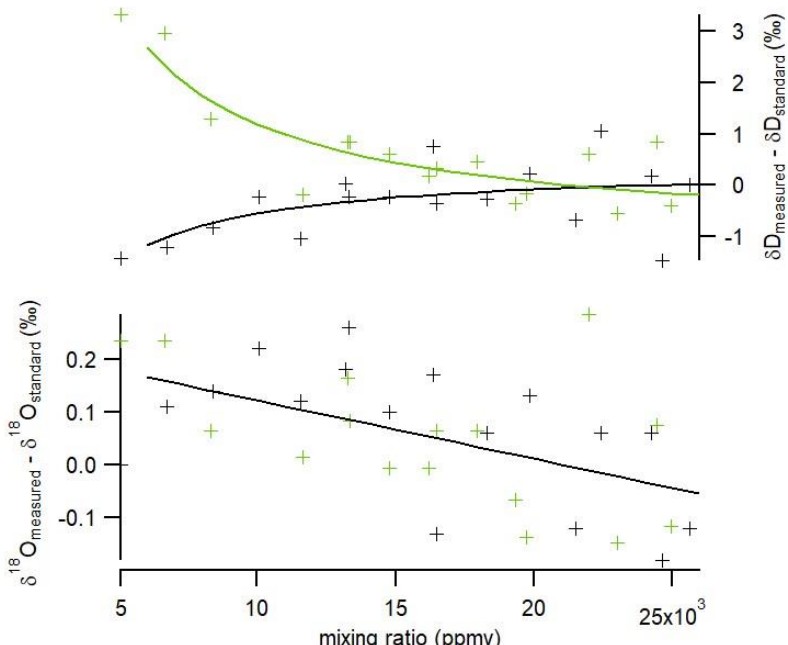


**Figure 2**: Influence of the water vapor mixing ratio on measured $\delta D$ (top) and $\delta^{18}O$ (bottom)

(anomaly from the true value of the standard). The results are shown for two different

standards (GREEN_AMS in green and EPB_AMS in black). The crosses indicate the data

obtained with the set-up and the solid lines are the best regression curves (same curve for

$\delta^{18}O$ for both standards).

The calibration of water vapour mixing ratio was performed in the laboratory before sending

the instrument to Amsterdam Island. In the field, we found an excellent agreement between

mixing ratio measured by the Picarro instrument and mixing ratio measured by the weather

station (the difference between the two records always stays below 2% and there is no

systematic shift between the two records).

The calibration of the water isotopic data is performed in several steps following previous

studies (Leroy-Dos Santos et al., 2020; Tremoy et al., 2011) and using a standard delivery

module by Picarro. First, we quantified the influence of the water vapor mixing ratio on the

water isotope ratios. This effect is large at very low humidity (Leroy-Dos Santos et al., 2021).

It can also depend on the isotopic composition of the standard water (Weng et al., 2020).

Here, we introduced two different water standards, EPB-AMS and GREEN-AMS, with

respective values of (-5.66 ‰, -47.31 ‰) and (-32.65 ‰, -263.76 ‰) for the couple ($\delta^{18}O$,

$\delta D$) which encompass the isotopic values observed on site. While we would expect a constant

null value for ($\delta^{18}O_{measured}$- $\delta^{18}O_{standard}$) in Figure 2 because we always inject the same water

standards, the measured $\delta^{18}O$ values of both EPB-AMS and GREEN-AMS standards in fact

decrease with increasing humidity with the same amplitude. The ($\delta D_{measured}$-$\delta D_{standard}$)
displayed in Figure 2 also shows variations but in contrast to the relative evolution of $\delta^{18}O$
with respect to water vapor mixing ratio, the $\delta D$ measurements of EPB-AMS and GREEN-
AMS standards exhibit different behavior: $\delta D$ of EPB-AMS increases by 1.5‰ and $\delta D$ of
GREEN-AMS decreases by 2.5 ‰ over the same 6,000-24,000 ppmv range for water vapor
mixing ratio $q_v$.
As a consequence, the raw $\delta^{18}O_v$ measurements are corrected with the following regression:

$$\delta^{18}O_{v,corr} = \delta^{18}O_{v,measured} + 1.1.10^{-5} \times q + 0.232 \qquad \text{(Eq 4)}$$

For the correction of the raw $\delta D_v$, we use two different regression splines for EPB-AMS and
GREEN-AMS (cf Figure 2):

$$\delta D_{EPB-AMS,corr} = \delta D_{EPB-AMS,measured} + \frac{9300}{q} - 0.383 \qquad \text{(Eq 5)}$$
$$\delta D_{GREEN-AMS,corr} = \delta D_{GREEN-AMS,measured} - \frac{22400}{q} + 1.05 \qquad \text{(Eq 6)}$$

The raw $\delta D_v$ are thus weighted-corrected according to their distance to the EPB_AMS and the
GREEN_AMS splines as follows:


$$\delta D_{v,corr} = \delta D_{GREEN-AMS,corr} + \frac{\delta D_{v,measured} - \delta D_{GREEN-AMS,measured}}{\delta D_{EPB-AMS,measured} - \delta D_{GREEN-AMS,measured}} \times (\delta D_{EPB-AMS,corr} - \delta D_{GREEN-AMS,corr})$$
$$\text{(Eq 7)}$$

This first calibration step (correction from the influence of mixing ratio on the isotopic
composition) has been performed every year over the whole range of mixing ratio values and
provided very similar results from one year to the other.The second calibration step consists in
the injection of the same two isotopic standards every 47 h at a water vapor mixing ratio of
13,000 ppmv to correct for any long-term drift. The correction associated with this drift is less
than 0.4 ‰ for $\delta^{18}O$ and 2.5 ‰ for $\delta D$ over the two years of measurements.
Precipitation were also sampled on a weekly basis in a rain gauge filled with paraffin oil which
permits to have measurements of water isotopic composition in the precipitation on a weekly
basis. The water samples are then sent for analyses to LSCE (Laboratoire des Sciences du
Climat et de l'Environnement) and measured with an isotopic analyzer L2130-i by Picarro. The
uncertainty associated with this series of measurements is of ±0.15 ‰ for $\delta^{18}$O and ±0.7 ‰ for
$\delta$D leading to an uncertainty of ±1.4 ‰ for d-excess.

**2.4 Back trajectories: FLEXPART**

The origin and trajectory of air masses were calculated by FLEXPART, which is a Lagrangian
particle dispersion model (Pisso et al., 2019). All the meteorological data used to simulate the
back trajectories are taken from the ERA5 atmospheric reanalysis (Hersbach et al., 2020) with
a 6-hourly resolution. The ERA5 reanalysis is carried out by the European Center for Medium-
Range Weather Forecasts (ECMWF), using ECMWF's Earth System model IFS (Integrated
Forecasting System), cycle 41r2. For a few selected events, we used FLEXPART to calculate
back trajectories over 5 days with 1000 launches of neutral particles (sensitivity test) of inert
air tracers released randomly (volume of 0.1°×0.1°×100 m) every 3 hours at 100 m above sea
level (Leroy-Dos Santos et al., 2020) centered around the coordinates of Amsterdam Island.
The results of the FLEXPART back trajectories are then displayed as particle probability
density  as well as through the location of their humidity weighted averages.
**2.5 General atmospheric circulation model equipped with water stable**
**isotopes**

**2.5.1 LMDZ-iso model (Laboratoire de Météorologie Dynamique Zoom model**
**equipped with water isotopes)**

LMDZ-iso (Risi et al., 2010) is the isotopic version of the atmospheric general circulation
model LMDZ6 (Hourdin et al., 2020). We have used LMDZ-iso version 20230111.trunk with
the physical package NPv6.1, identical to the atmospheric setup of IPSL-CM6A (Boucher et
al., 2020) used for phase 6 of the Coupled Model Intercomparison Project (CMIP6, Eyring et
al., 2016). We performed two simulations, one at very low horizontal resolution (VLR, 3.75°
in longitude and 1.9° in latitude, 96×95 grid cells) and the second at low horizontal resolution
(LR, 2.0° in longitude and 1.67° in latitude, 144×142 grid cells). Both simulations have 79
vertical levels and the first atmospheric level is located around 10 m above ground level. The
LMDZ-iso 3D-fields of temperature and wind are nudged toward the 6-hourly ERA5 reanalysis
data with a relaxation time of 3 hours. Surface ocean boundary conditions are taken from the
monthly mean SST and sea-ice fields from the CMIP6 AMIP Sea Surface Temperature and Sea
Ice dataset version 1.1.8 (Durack et al., 2022; Taylor et al., 2000). LMDZ-iso outputs are used
at a 3-hourly resolution. Amsterdam Island (58 km$^2$) is too small to be represented in the
LMDZ-iso model.

**325     2.5.2 ECHAM6-wiso model (European Centre Hamburg model equipped with water**

**326     isotopes)**


ECHAM6-wiso (Cauquoin et al., 2019; Cauquoin and Werner, 2021) is the isotopic version of
the atmospheric general circulation model ECHAM6 (Stevens et al., 2013). The
implementation of the water isotopes in ECHAM6 has been described in detail by Cauquoin et
al. (2019), and has been updated in several aspects by Cauquoin and Werner (2021) to make
the model results more consistent with the last findings based on water isotope observations
(isotopic composition of snow on sea ice considered, supersaturation equation slightly updated,
and kinetic fractionation factors for oceanic evaporation assumed as independent of wind
speed). We have used ECHAM6-wiso model outputs from a simulation with a T127L95 spatial
resolution (0.9° horizontal resolution and 95 vertical levels). ECHAM6-wiso is thus run with a
finer resolution than both LMDZ-iso simulations. The ECHAM6-wiso 3D-fields of
temperature, vorticity and divergence as well as the surface pressure field were nudged toward
the ERA5 reanalysis data every 6 hours (Hersbach et al., 2020). The orbital parameters and
greenhouse gas concentrations have been set to the values of the corresponding model year.
The monthly mean sea surface temperature and sea-ice fields from the ERA5 reanalysis have
been applied as ocean surface boundary conditions, as well as a mean $\delta^{18}O$ of surface seawater
reconstruction from the global gridded data set of LeGrande and Schmidt (2006). As no
equivalent data set of the $\delta D$ composition of seawater exists, the $\delta D$ of the seawater in any grid
cell has been set equal to the related $\delta^{18}O$ composition, multiplied by a factor of 8, in accordance
with the observed relation for meteoric water on a global scale (Craig, 1961). The ECHAM6-
wiso simulation is described in detail and evaluated by Cauquoin and Werner (2021).
ECHAM6-wiso outputs are given at a 6-hourly resolution. As for the LMDZ-iso model,
Amsterdam Island (58 km$^2$) is too small to be represented by ECHAM6-wiso.

# 3. Results

## 3.1 Data description

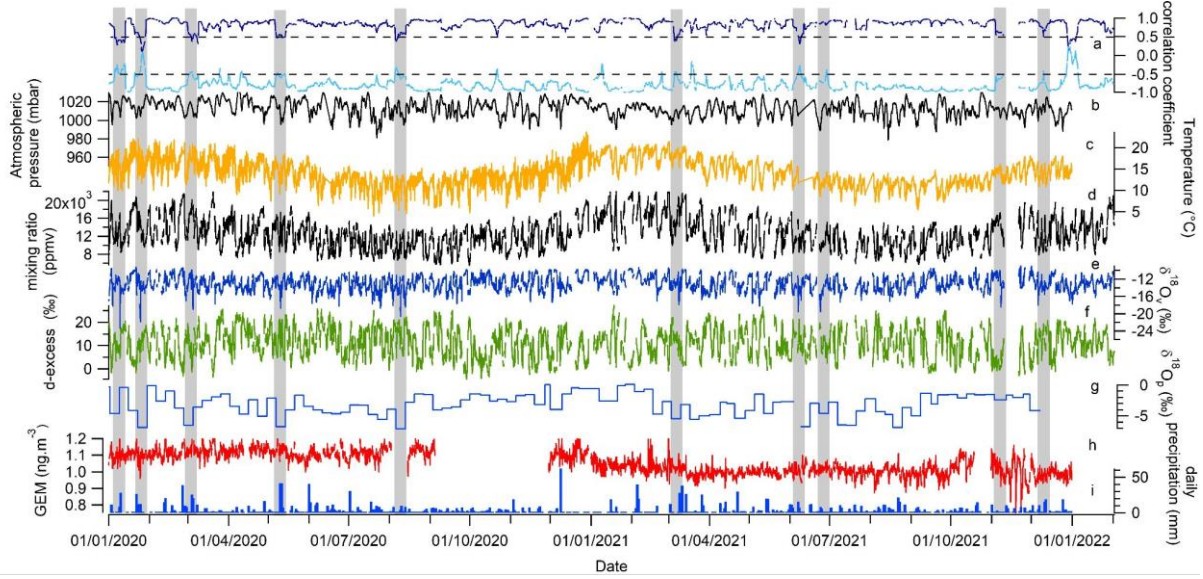

**Figure 3 :** Meteorological, isotopic and GEM records for the years 2020 and 2021 on the Amsterdam Island : (a) correlation coefficient between $\delta^{18}O_v$ and mixing ratio (dark blue, top) and between $\delta^{18}O_v$ and d-excess$_v$ (light blue, bottom) over a moving time window of 8 days, (b) atmospheric pressure (hourly average), (c) atmospheric temperature (hourly average), (d) water vapor mixing ratio (hourly average), (e) $\delta^{18}O_v$ (hourly average), (f) d-excess$_v$ (hourly average), (g) $\delta^{18}O$ of precipitation sampled on a weekly basis, (h) GEM concentration (hourly average), (i) daily precipitation. The grey shaded areas indicate the negative excursions in $\delta^{18}O_v$ associated with decorrelation between water vapor mixing ratio and $\delta^{18}O_v$ and a correlation coefficient >-0.5 between d-excess$_v$ and $\delta^{18}O_v$.

### 3.1.1 Temporal variability in the meteorological records

As mentioned earlier, there is a clear annual cycle at Amsterdam Island as recorded in the temperature and water vapor mixing ratio for the years 2020 and 2021. The December-February period (austral summer) has the highest temperatures with an average of 15.0°C, while in winter (July-September) the average temperature varies around 10.5°C. In parallel, we do not see clear patterns of a diurnal cycle in the temperature record except for some periods yet with a small amplitude (4-5 °C).

The impact of synoptic events at the scale of a few days is visible in the temperature and water mixing ratio with a covariation of temperature and water vapor mixing ratio and amplitudes of up to 10°C and more than 10,000 ppmv.

### 3.1.2 Temporal variability in the GEM record

Previous studies clearly showed that AMS is little influenced by anthropogenic sources of mercury, and greatly influenced by the ocean surrounding the island (Angot et al., 2014; Hoang et al., 2023; Jiskra et al., 2018; Li et al., 2023; Slemr et al., 2015, 2020). Angot et al., 2014 reported mean annual GEM concentrations of about $1.03 \pm 0.08$ ng m$^{-3}$ from 2012 to 2013. These concentrations are ~30% lower than those measured at remote sites of the northern hemisphere. Over the period 2012 to 2017, Slmer et al. (2020) confirmed that higher GEM concentrations can be found during austral winter. Lower GEM values are generally observed in October and November, as well as in January and February during austral summer. Using this 6-year long data set, mean annual GEM concentration is $1.04 \pm 0.07$ ng m$^{-3}$ (annual range: 1.014 to 1.080 ng m$^{-3}$) i.e. very close to the one observed by Angot et al. (2014).

Surprisingly, unlike the 2012-2017 data set, GEM presented in this study did not show a significant higher mean concentration during the austral winter months than during the summer months (Figure 3), with consequently no discernible seasonal amplitude of GEM. On a finer timescale, the lack of a clear pattern of GEM seasonal cycle is counterbalanced by days showing abrupt increases or decreases in concentrations. Some of the sudden GEM decreases appear concomitant with important negative peaks of several ‰ in $\delta^{18}O_v$.

### 3.1.3 Temporal variability of water isotopic composition

The isotopic composition of precipitation ($\delta^{18}O_p$) sampled on a weekly basis displays a quite large variability ($\delta^{18}O_p = -3.06 \pm 1.75$ ‰, n=104) with values slightly higher during austral summer (difference between summer and winter $\delta^{18}O_p$ values is about 2 to 3 ‰) (Figure 3). No

significant seasonal variations are observed in the record of d-excess of precipitation (not
shown).
No diurnal cycle can be detected in the $\delta^{18}O_v$ and d-excess$_v$. An annual cycle is not visible either
(1 ‰ difference between summer and winter mean $\delta^{18}O_v$ value while standard deviation of the
entire record at 1 h resolution is 1.7 ‰). Only the synoptic scale variability is well expressed in
the records of $\delta^{18}O_v$ and d-excess$_v$ with an anticorrelation between both parameters when
looking at the 2-year series at hourly resolution ($R^2 = 0.61$ with $R^2$ being the coefficient of
determination for a linear regression). Moreover, $\delta^{18}O_v$ is most of the time correlated with water
vapor mixing ratio ($R^2 = 0.55$ for the 2-year series at hourly resolution).
There are a few exceptions to the general correlation between water vapor $\delta^{18}O$ and water vapor
mixing ratio as illustrated in Figure 3. Short periods of a few days are associated with a decrease
of the correlation coefficient, R estimated from the correlation between $\delta^{18}O_v$ and $q_v$ (R is
calculated continuously from hourly records on an 8-day moving window). The periods of low
R are also often characterized by a negative peak of several ‰ in $\delta^{18}O_v$, which is not visible in
the d-excess$_v$. During these $\delta^{18}O_v$ excursions, the general anti-correlation between $\delta^{18}O_v$ and d-
excess$_v$ hence also breaks down. Our study mostly focuses on the 11 most prominent abrupt
events highlighted in the $\delta^{18}O_v$ record (only 10 visible on Figure 3 because of the scale). The
11 most abrupt events occurring when correlation coefficient R between $\delta^{18}O_v$ and d-excess$_v$ is
larger than -0.5 are associated with $\delta^{18}O_v$ negative excursion larger than 3 ‰ (at 6h resolution)
over a period of less than 24 h, the length of the event being measured between the mid-slopes
of the decrease and subsequent increase of the $\delta^{18}O_v$. The 11 selected negative excursions occur
at a rate larger than -0.5‰ h$^{-1}$ and the $\delta^{18}O_v$ increase at the end of each excursion has an
amplitude larger than half the amplitude of the corresponding initial decrease.

**3.2 Model-data comparison**

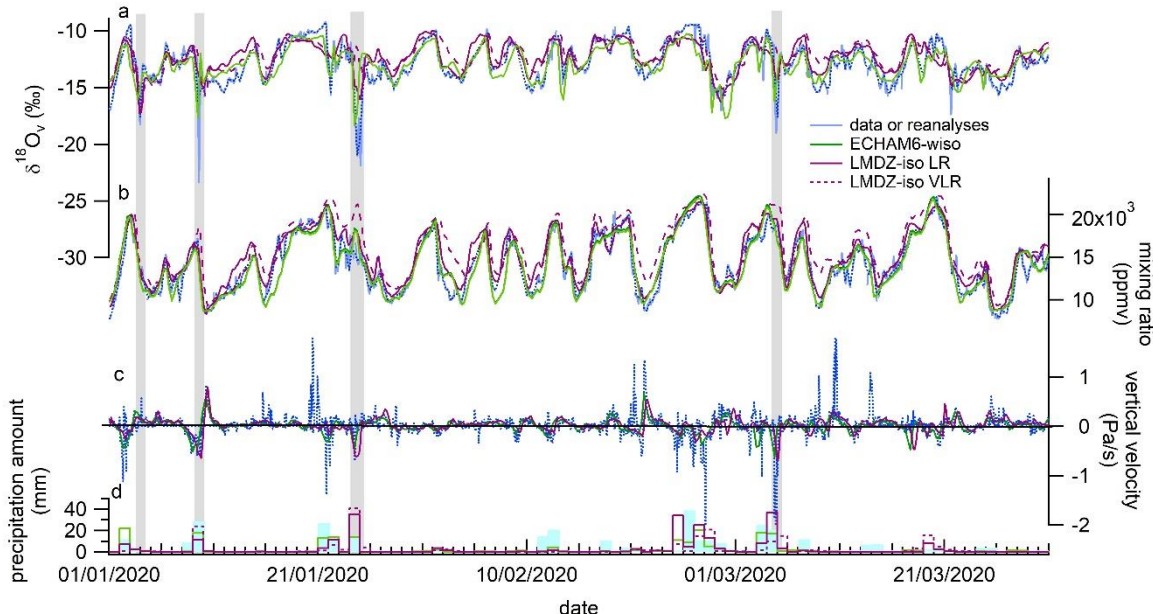


**Figure 4**: Model-measurement comparison (January – March 2020); a- $\delta^{18}O_v$ (light blue for
data on hourly average, dotted dark blue for data resampled at a 6-hour resolution); b- water
vapor mixing ratio from our data set; c- vertical velocity; d- Precipitation amount. The grey
shaded areas highlight the negative $\delta^{18}O_v$ excursions as defined in 3.1.3 (note that in this figure
the excursions of the 3$^{rd}$ and 9$^{th}$ of January 2020 are distinct while the distinction could not be
done on Figure 3 because of the scale).

We selected a 3-month period (January to March 2020) for the comparison between our dataset
and the outputs of the ECHAM6-wiso and LMDZ-iso models. This period has been selected
for display because it encompasses 4 out the 11 negative excursions of $\delta^{18}O_v$, but the extended
comparison over the whole 2 years period is displayed in Figure A1. There is an overall
agreement between the measured and modelled $\delta^{18}O_v$ and water vapor mixing ratio (Figure 4).
The best agreement over the 3-month series is obtained with the ECHAM6-wiso and LMDZ-
iso (LR) models (R$^2$ = 0.59 – 0.6 and 0.87 - 0.90 respectively for $\delta^{18}O_v$ and water vapor mixing
ratio series) while a slightly less good agreement is observed with the VLR simulation of the
LMDZ-iso model (R$^2$ = 0.49 and 0.79 respectively for $\delta^{18}O_v$ and water vapor mixing ratio
series). The same observation can be done on the entire 2-year time series. We also compare
the precipitation amount modelled by ECHAM6-wiso and LMDZ-iso to the precipitation
amount measured by the MeteoFrance weather station. The correlation between modeled and
measured precipitation is close to zero for LMDZ-iso (R$^2$ = 0.08 – 0.13 for VLR - LR) while
there is a better agreement when comparing measured precipitation amount to outputs of
ECHAM6-wiso ($R^2 = 0.45$). Finally, when focusing on the short term negative $\delta^{18}O_v$ excursions
(Figures 4 and A1), they are in general more strongly expressed in the measurement time series
than in the model series. Part of this disagreement can be explained by the fact that the $\delta^{18}O_v$
record has a higher temporal resolution (1h) than the model outputs (3h for LMDZ-iso and 6h
for ECHAM6-wiso). However, when interpolating the $\delta^{18}O_v$ record at a 6h resolution (dotted
dark blue), the negative excursions are still clearly visible while not captured by the LMDZ-iso
model (Figure 4 and Table 1). When looking at the whole 2-year series, the LMDZ-iso VLR
simulation fails to reproduce most of these $\delta^{18}O_v$ excursions (only the negative excursion of 3$^{rd}$
January, 2020 is reproduced) while the ECHAM6-wiso model is able to capture all the $\delta^{18}O_v$
excursions. The LMDZ-iso LR simulation produces a negative $\delta^{18}O_v$ excursion over many
events with a significantly lesser amplitude than in the data and in the ECHAM6-wiso model
(Table 1).

**4. Discussion**
The most remarkable pattern from this 2-year series is the succession of short negative
excursions of $\delta^{18}O_v$ associated with decorrelation between $\delta^{18}O_v$ and humidity, $\delta^{18}O_v$ and d-
excess$_v$, and which are highlighted with grey shaded areas in Figure 3, detailed in Figures 5 and
A2 and referenced in Table 1. These negative $\delta^{18}O_v$ excursions always occurred during low
pressure periods (atmospheric pressure below 1005 mbar) and we observe the presence of a
cold front within a distance of 100 km around Amsterdam Island in a 48h period covering the
time of the event (Supplementary Material Figure S1). The focus on the first three months of
the series presented in Figure 4 shows that these events are captured by ECHAM6-wiso at 0.9°
resolution, but not systematically by LMDZ-iso at 2x1.67° and even less by LMDZ-iso at
3.75x1.9° resolution. Such mismatch makes the understanding of the processes at play during
these events particularly important to investigate to further improve the performances of
atmospheric general circulation models equipped with water isotopes. .

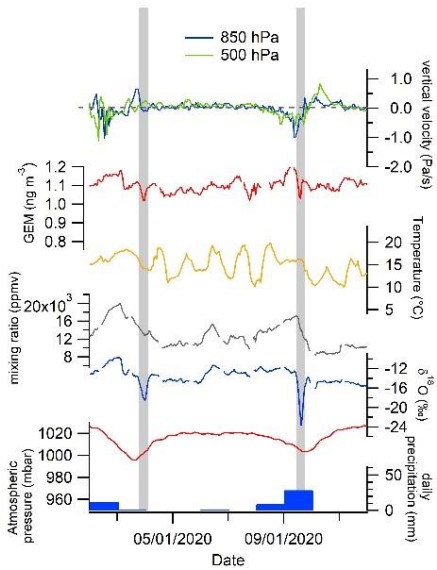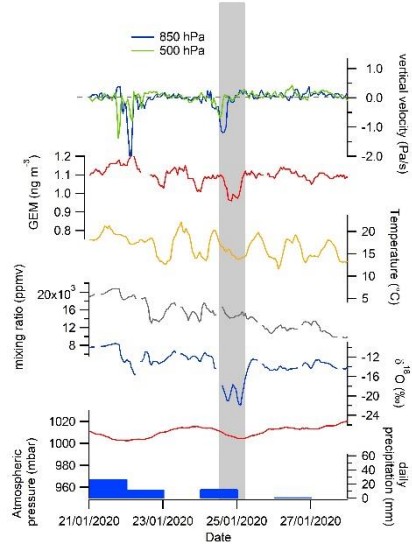

**Figure 5:** Evolution of GEM, $\delta^{18}O_v$, water vapor mixing ratio, meteorological parameters (surface temperature, surface atmospheric pressure, daily precipitation) measured by the MeteoFrance weather station and vertical velocity from the ERA5 reanalyses at 500 and 850 hPa over the three isotopic excursions of January 2020 identified on Figure 4. A focus on the other excursions is provided in Figure A2.

**Table 1**: List of the 11 events associated with both loss of correlation between $\delta^{18}O_v$ and $q_v$, $\delta^{18}O_v$ and d-excess$_v$ and negative excursions of $\delta^{18}O_v$ over 2020-2021. The amplitude of the negative $\delta^{18}O_v$ anomaly is calculated from the minimum of $\delta^{18}O_v$ on the record at hourly resolution (at 6h resolution). When the calculated amplitude is smaller than 1 ‰, we indicate only "-". When the vertical velocity is between -0.25 and 0.25 Pa/s, this is indicated in the table as "~0".

| Date of the event | Negative excursion of GEM | Low pressure (< 1005 mbar) | Rain | Relative Humidity at the surface (at minimum $\delta^{18}O_v$) | vertical velocity from reanalyses (850 hPa) | vertical velocity from reanalyses (500 hPa) | Length of the event (hours) | amplitude of the $\delta^{18}O_v$ peak in the data (‰) | amplitude of the $\delta^{18}O$ peak in ECHAM-wiso (‰) | amplitude of the $\delta^{18}O$ peak in LMDZ-iso VLR (‰) | amplitude of the $\delta^{18}O$ peak in LMDZ-iso LR (‰) |
|---|---|---|---|---|---|---|---|---|---|---|---|
| 06/12/2021 | Yes | Yes | Yes | 82% | ~0 | up | 3h | -6 (-5) | -2.3 | - | -2 |
| 08/11/2021 | Yes | Yes | No | 85% | ~0 | ~0 | 17h | -5.5 (-5.5) | -5 | - | -4 |
| 23/06/2021 | No | Yes | Yes | 75% | ~0 | ~0 | 10h | -5.5 (-5.4) | -6 | - | - |
| 07/06/2021 | No | Yes | Yes | 80% | up | ~0 | 9h | -6.5 (-5.8) | -5.8 | - | -2 |
| 08/03/2021 | Yes | Yes | Yes | 89% | down | up | 20h | -6 (-6) | -4 | - | - |
| 09/08/2020 | No data | Yes | Yes | 87% | down | up | 8h | -8 (-6) | -7 | - | -2 |
| 10/05/2020 | Small | Yes | Yes | 95% | down | down | 14h | -4.9 (-4) | -3 | - | -3 |
| 04/03/2020 | No data | Yes | Yes | 98% | up | up | 9h | -6.1 (-5.3) | -5 | - | - |
| 24/01/2020 (double peak) | Yes | Yes | Yes | 93% and 90% | 1st peak up and 2nd peak down | 1st peak up and 2nd peak down | 17h | -7.8 (-7.5) | -4.5 | - | -3.5 |
| 09/01/2020 | Yes | Yes | Yes | 94% | up | up | 4h | -9 (-4) | -5 | - | - |
| 03/01/2020 | Yes | Yes | No | 90% | down | ~0 | 6h | -2.8 (-2.5) | -2.4 | -3 | -3.5 |

Several hypotheses can be proposed to explain the negative excursions of $\delta^{18}O_v$. The beginning of these excursions is associated with a decrease of the water vapor mixing ratio and occurs in most cases during a precipitation event (Table 1). These events share similarities with negative $\delta^{18}O_v$ and $\delta^{18}O_p$ short events previously observed in temperate regions during a cold front passage (e.g. Aemisegger et al., 2015). Three possible processes at play to explain such events have already been listed in previous studies (e.g. Dütsch et al., 2016) (i) local interaction between the vapor and the rain droplets (rain equilibration and rain evaporation), (ii) vertical subsidence of water vapor with depleted isotopic composition, or (iii) horizontal advection through the arrival of a cold front. We explore below how we can gain information on the different processes using our data set, back trajectories and model-data comparison.

**4.1 $\delta^{18}O_v$ vs $q_v$ relationship**

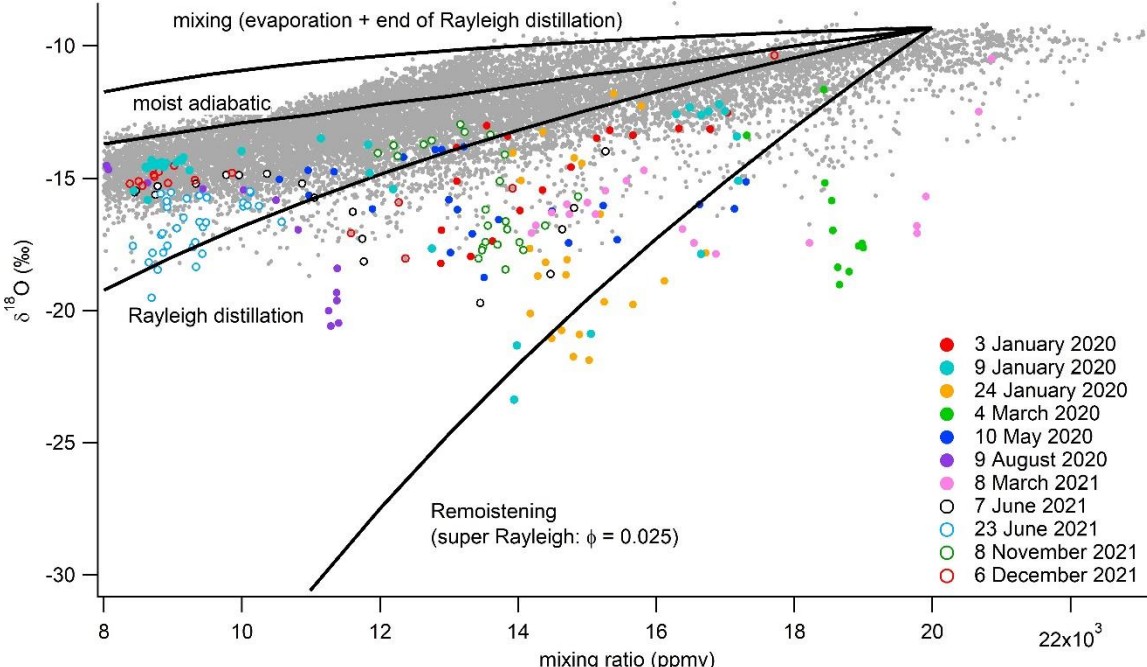

499

**Figure 6**: Relative evolution of $q_v$ and $\delta^{18}O_v$ for the different events (colors according to the date as explained in the graph) and for the entire 2 years records (grey). The solid lines are theoretical lines whose equations are detailed in Noone (2012) for different processes (remoistening associated with exchange between rain and water vapor; Rayleigh distillation assuming that all formed condensation is removed from the cloud; moist adiabatic process assuming that liquid condensation stays in the cloud with the water vapor; mixing of water vapor from ocean evaporation around Amsterdam Island and water vapor from the end of the Rayleigh distillation, i.e. high altitude water vapor). The water vapor for the calculation of Rayleigh distillation and for the evaporation above the ocean has a $q_{v,0}$ of 20,000 ppmv and a $\delta^{18}O_{v,0}$ of -9.3 ‰. The vapor at the end of the distillation line has a water vapor mixing ratio of 1,000 ppmv and a $\delta^{18}O_v$ of -40 ‰.

First, to test the hypothesis of vapor-droplet interactions, we looked at the $\delta^{18}O_v$ vs $q_v$ distribution following the approach already used by Guilpart et al. (2017) (Figure 6). We acknowledge that our approach is crude and should be taken as a first order approach since we can only look at the water vapor $\delta^{18}O_v$ vs $q_v$ distribution in the surface layer using adapted boundary conditions while it may be more relevant to look at this relationship in the free troposphere. In general, the $\delta^{18}O_v$ vs $q_v$ evolution lies on a curve which can be explained by condensation processes (Rayleigh distillation or reversible moist adiabatic process). However,

for the 11 events highlighted above, the water vapor $\delta^{18}O_v$ vs $q_v$ evolution follows an evolution
standing below the curve of the $\delta^{18}O_v$ vs $q_v$ evolution observed for the rest of the series.
Although the evolution of the water vapor $\delta^{18}O_v$ vs $q_v$ is rather abrupt, there is a certain
resemblance with the idealized theoretical remoistening curve initially calculated for the free
troposphere (Noone, 2012) and adapted here with initial conditions corresponding to the
isotopic composition of surface water vapor. Remoistening is described through a modification
of the equilibrium fractionation coefficient between water vapor and rain ($\alpha_e$) so that the
effective fractionation factor is $\alpha=(1+\phi)\times\alpha_e$, $\phi$ being the degree to which $\alpha$ deviates from
equilibrium. This effective fractionation coefficient is then introduced in the Rayleigh
distillation equation to deduce the link between $\delta^{18}O_v$ and mixing ratio as:

530                $$\delta^{18}O_v - \delta^{18}O_{v,0} = (\alpha-1)\times\ln(q_v/q_{v,0}) \qquad \text{(Eq 8)}$$

Despite the simplicity of our approach, the fact that the water vapor $\delta^{18}O_v$ vs $q_v$ evolution lies
below the idealized curve for condensation processes supports the depleting effect of vapor-
rain interactions for our negative water vapor $\delta^{18}O_v$ excursions (Noone, 2012; Worden et al.,
2007). Surface relative humidity remains relatively high during these events (values given in
Table 1 compared to a mean value of 77 %) which favors rain-vapor diffusive exchanges. This
interpretation is also supported by the stable d-excess$_v$ during these events.

**4.2 $\delta^{18}O_v$ vs GEM relationship**
Second, to test the hypothesis of subsidence of air from higher altitude, GEM is used. Indeed,
aircraft measurements as well as model simulations demonstrated that the upper
troposphere/lower stratosphere is depleted in GEM and enriched in species composed of
reactive gaseous mercury and particulate bound mercury (Lyman and Jaffe, 2012; Murphy et
al., 2006; Sillman et al., 2007; Swartzendruber et al., 2006, 2008; Talbot et al., 2007, 2008).
This leads to lower GEM concentrations than those usually observed when the lowest
atmosphere layer is only under marine influence (Angot et al., 2014; Lindberg et al., 2007). The
fact that GEM negative excursions are observed in phase with negative $\delta^{18}O_v$ excursions in
most of the events (6 events on a total of 9 events with GEM data, cf Figure 5 and A2, Table 1)
suggests that vertical subsidence of water vapor, $\delta^{18}O$-depleted by Rayleigh distillation and/or
rain-vapor interactions, can have an influence on the observed excursions of $\delta^{18}O_v$, in
agreement with the conclusion of Dütsch et al. (2016).

**4.3 Back trajectories information**

To further explore the processes leading to the decoupling of humidity and $\delta^{18}O_v$ as well as sharp negative excursions of $\delta^{18}O_v$ during the 11 events identified here, we also use information from the ERA5 reanalyses. In particular, the influence of atmospheric circulation (vertical and horizontal advection) and moisture origin can be studied through back trajectories. The back trajectories, presented here for 3 events (Figures 7, A3 and A4), confirm the information from wind directions that there is no systematic change in the horizontal origin of the trajectories for the different events. No systematic pattern is identified either in the vertical advection even if we note that for the event of January 3$^{rd}$, the average altitude of the envelope of the 5-day back trajectories increases when comparing the situation before the excursion and the situation when the most negative $\delta^{18}O_v$ values are reached. This observation may support the occurrence of air subsidence as indicated by the GEM record for this particular event (Figure 5).

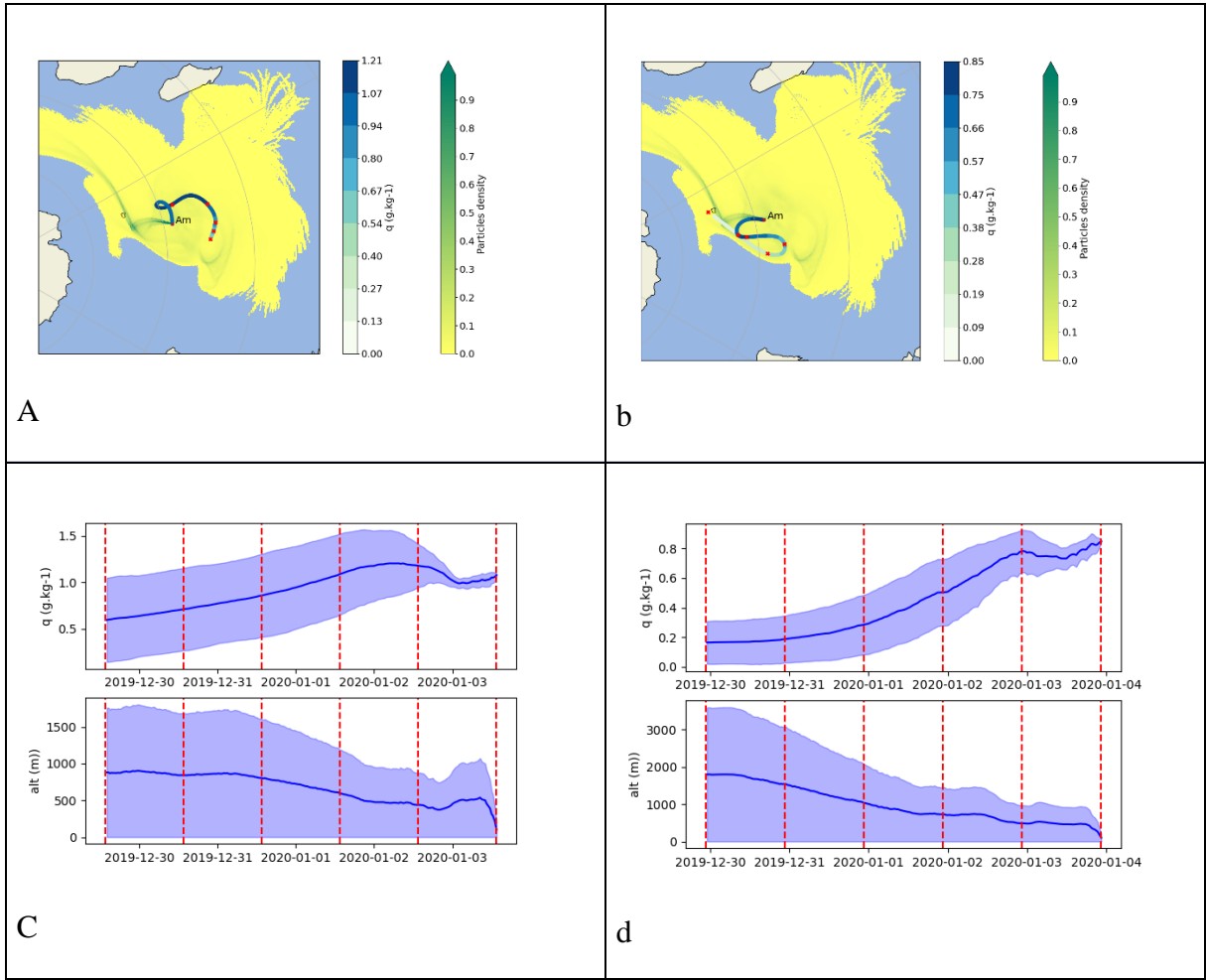

**Figure 7** : FLEXPART footprints of 5-day back trajectories for the event of the 3$^{rd}$-4$^{th}$ of January. (a) Latitude-longitude projection of the FLEXPART back trajectory footprints for January 3$^{rd}$ 2020 at 13h30. The yellow to green colors on each grid point of these projections

represent the density of particles. The white to blue colors indicate the water vapor mixing
ratio along the humidity-weighted average back trajectory. Each red point indicates the
location of the average back trajectory for each of the 5 days before the date of the considered
event. (b) Same as a for January 3$^{rd}$ 2020 at 22h30. (c) Top shows the evolution of the water
vapor mixing ratio of the back trajectories for January 3$^{rd}$ 2020 at 13h30; bottom shows the
altitude evolution of the back trajectory for January 3$^{rd}$ 2020 at 13h30. (d) same as (c) for
January 3$^{rd}$ 2020 at 22h30.

The subsidence over the different events can better be studied from the vertical velocity from
the ERA5 reanalyses (Figure 4 and A1). Subsidence (positive vertical velocity) is not
systematically associated with negative $\delta^{18}O_v$ excursions: subsidence at either 850 hPa or 500
hPa is observed only for 5 events over 11 (Table 1). In 4 cases, there is rather an ascending
movement of the atmospheric air associated with the rain event. In the other cases, there is no
clear vertical movement. However, we note that when negative $\delta^{18}O_v$ excursions are not
concomitant with subsidence, they occur at the end of an ascending movement which is
generally followed by subsidence (Figures A1 and A2).

**4.4 Model – data comparison and atmospheric dynamic**
With the information gathered above, both subsidence and isotopic depletion associated with
rain occurrence and further interaction between droplets and water vapor can explain the
negative excursions of $\delta^{18}O_v$. We note however that the data gathered so far do not permit to
provide a simple and unique explanation. Neither subsidence nor rain systematically occurred
for each of the $\delta^{18}O_v$ excursion. Still, the fact that at least ECHAM6-wiso is able to reproduce
every negative $\delta^{18}O_v$ excursion (whether they are associated or not with subsidence or rain-
water vapor reequilibration) shows that (1) the patterns of atmospheric water cycle are correctly
reproduced, a validation which can be performed using humidity and precipitation data for
some aspects but benefits from water isotopes implementation for the residence time of water
and (2) the isotopic processes are correctly implemented in this model. Such abrupt $\delta^{18}O_v$ events
can hence be used as a test bed of the performances of water isotopes enabled general circulation
models.
To further explore the $\delta^{18}O_v$ data-model comparison and associated processes, we compare the
performances of the ECHAM6-wiso and the LMDZ-iso models over the first months of 2020
in terms of atmospheric dynamics (Figures 4 and A1). First and as expected because of the
nudging, the two models reproduce rather well the evolution of the vertical velocity of the
ERA5 reanalyses with a stronger ascent for the model predicting the strongest precipitation
amount (e.g. LMDZ-iso for January 24$^{th}$ 2020). The event of January 3$^{rd}$ is the only one
reproduced by both ECHAM6-wiso and the two versions of the LMDZ-iso model: the three
simulations show a clear subsidence over the isotopic event and a clear negative $\delta^{18}O_v$
excursion. For the other events, neither LMDZ-iso nor ECHAM6-wiso show a clear signal of
subsidence neither at 500 nor at 850 hPa (not shown). However, the horizontal distribution of
vertical velocity obtained with ECHAM6-wiso and LMDZ-iso are significantly different
(Figure 8 for the event of the 9$^{th}$ of January, Supplementary Material Figures S2 and S3 for the
other events). While the LMDZ-iso modelled vertical velocity displays a rather strong
homogeneity on the vertical axis, ECHAM6-wiso modelled vertical velocity highlights
subsidence of air below the ascending column, with the maximum of negative $\delta^{18}O_v$ anomaly
at the surface located just at the limit between ascendance and subsidence (between 75°E and
77°E in Figure 8c). This subsidence of depleted $\delta^{18}O_v$ below the ascending column is
responsible for the sharp negative $\delta^{18}O_v$ excursion in the ECHAM6-wiso model. The fact that
subsidence of air occurs just below uplifted air, at the limit between ascendance and subsidence
(Figure 8k and Supplementary Material Figure S2), permits to reconcile the GEM data
suggesting subsidence and the sign of the vertical velocity of the ERA5 reanalyses at
Amsterdam Island suggesting that many excursions start with ascendance. Since the isotope
implementation was done similarly in the two models, the reason why the LMDZ-iso model
does not reproduce the water isotopic anomaly is its too coarse resolution as also supported by
the comparison between performances of the LMDZ-iso model at low resolution and very low
resolution for the event of the 24$^{th}$ of January (Table 1 and Figure 4). As already pointed by
Ryan et al. (2000), a fine resolution is necessary to correctly simulate front dynamics and we
extend this result here to the high resolution temporal patterns of surface $\delta^{18}O_v$.

**4.5 Synthesis**
Figure 9 summarizes the proposed mechanism for negative $\delta^{18}O_v$ excursions as inferred from
our data – model comparison when there is a clear rain event. A rain event is associated with a
strong ascending column in which $\delta^{18}O_v$ is depleted by progressive precipitation during the
ascent and by interaction between rain and water vapor. This ascending column is generally
associated with a cold front moving from South-West to North-Est (Fig. 8j and Supplementary
Material S1), with subsidence and $\delta^{18}O_v$ depleted air at the rear of the front (Fig. 8 and
Supplementary Material S2 and S3).

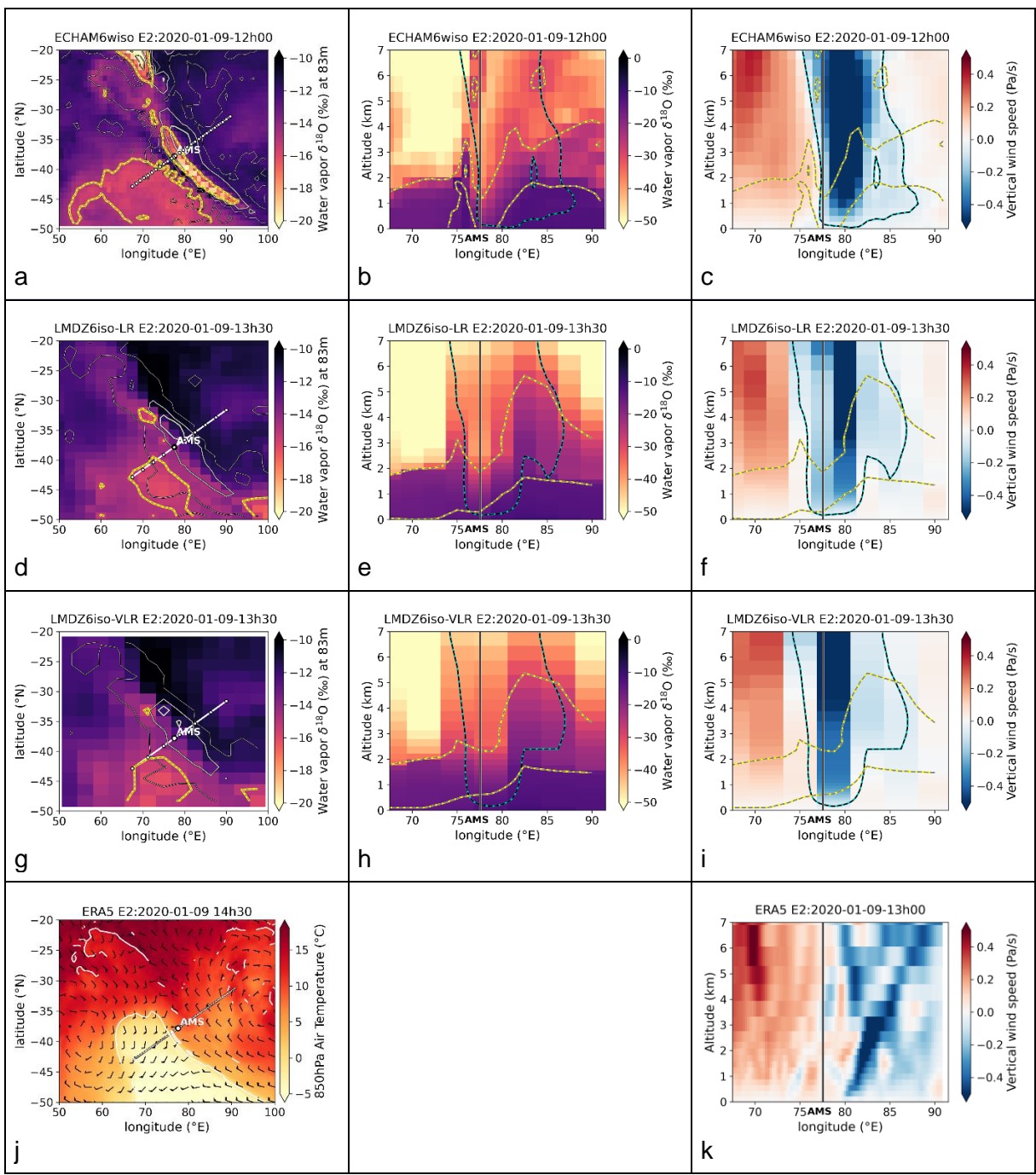

**Figure 8:** Modelled $\delta^{18}O_v$ and vertical velocity for the event of January 9th 2020. (a) Surface air $\delta^{18}O_v$ (~83 m, latitude vs longitude), with yellow line indicating -15 ‰ contour level and grey lines indicating precipitation contours at 0.5, 10, and 50 mm day$^{-1}$ (thin, medium and thick lines respectively); (b) $\delta^{18}O_v$ plotted on a vertical cross-section (altitude vs longitude) along the transect indicated by the white line on panel (a), with yellow lines indicating $\delta^{18}O_v$ contours at -30 ‰ and -15 ‰, blue lines indicating the contour of –0.05 Pa s$^{-1}$ vertical velocity (ascendance), and the vertical black line denoting the longitude of Amsterdam Island; (c) Vertical velocity plotted on a vertical cross-section as for (b), with same contour lines. (a), (b)

and (c) are drawn using outputs of the ECHAM6-wiso model ; (d), (e) and (f) are the same as (a), (b) and (c) but obtained from the LMDZ-iso model at low resolution (LR) ; (g), (h) and (i) are the same as (a), (b) and (c) but obtained from the LMDZ-iso model at very low resolution (VLR). (j) ERA5 air temperature at 850 hPa, with white lines marking front locations (see Supplementary Material S1); (k) ERA5 vertical velocity plotted on a vertical cross-section (altitude vs longitude) along the transect indicated by the black dotted line on panel (j).

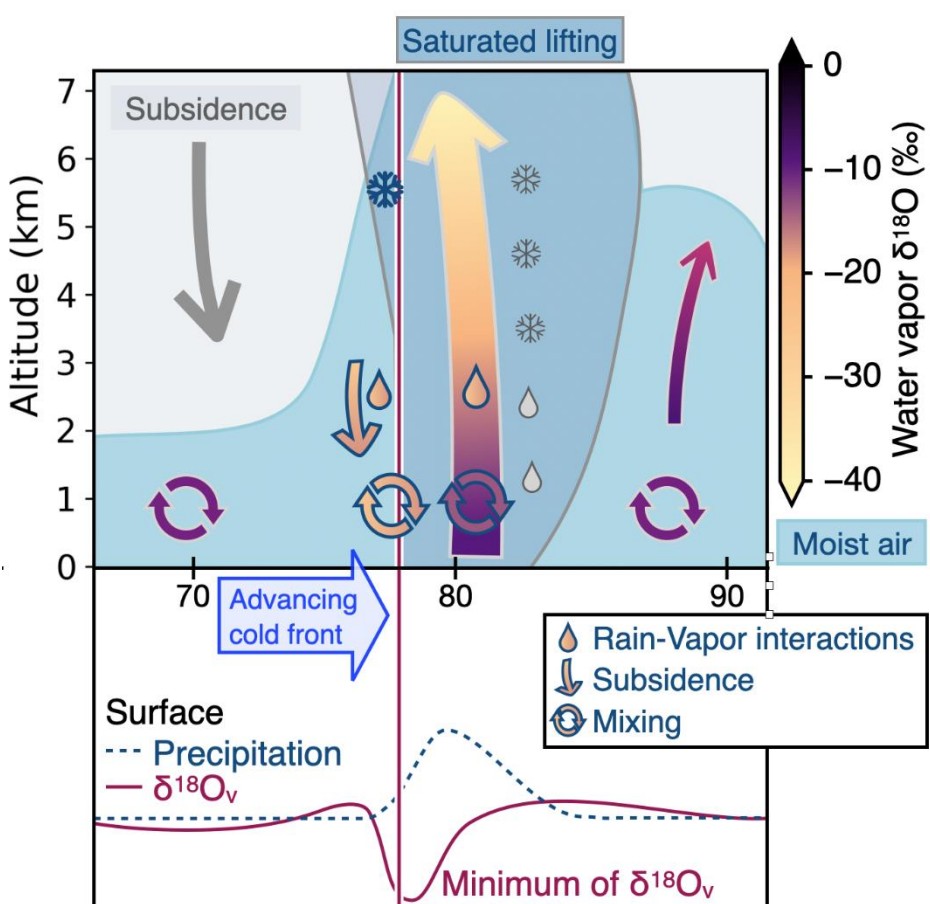

**Figure 9:** Scheme of the mechanism explaining the sharp negative excursion of $\delta^{18}O_v$ recorded at the surface for cold front events associated with precipitation. The scheme is based on the profile modelled by ECHAM6-wiso for event of January 9[th] 2020 (see Supplementary Material Figure S5 for other events). The top panel show the altitude vs longitude dynamics of air masses with vertical saturated lifting in the center and subsidence at the rear of the lifting. The bottom panel shows the associated evolution of $\delta^{18}O_v$ and precipitations on the same longitude scale than on the upper panel.

## 5. Conclusion

We presented here the first water vapor isotopic record over 2 years on Amsterdam Island. The water vapor isotopic variations follow at first order the variations of water vapor mixing ratio as expected for such a marine site. Superimposed to this variability, we have evidenced 11 periods of a few hours characterized by the occurrence of one or two abrupt negative excursions of $\delta^{18}O_v$ while the correlation between $\delta^{18}O_v$ and water vapor mixing ratio does not hold. These negative excursions are often occurring toward the end of precipitation events. They are most of the time occurring during a decrease in water vapor mixing ratio. Representation of these short events is a challenge for the atmospheric components of Earth System Models equipped with water isotopes and we found that the ECHAM6-wiso model was able to reproduce most of the sharp negative $\delta^{18}O_v$ excursions while the LMDZ-iso model at low (very low) resolution was only able to reproduce 7 (1) of the negative excursions. The good agreement between modeled and measured $\delta^{18}O_v$ when using ECHAM6-wiso validates the physics processes within the ECHAM6-wiso model as well as the implemented physics of water isotopes.

Using previous modeling studies as well as information provided by (1) the confrontation with other data sources (GEM, meteorology) obtained in parallel on this site, (2) back trajectory analyses and (3) the outputs of the two models ECHAM6-wiso and LMDZ-iso, we conclude that the most plausible explanations for such events are rain-vapor interactions and subsidence at the rear of a precipitation event. Both can be combined, since rain vapor interactions can help maintaining moist conditions in subsidence regions.

This study highlights the added value of combining different data from a surface atmospheric observatory to understand the dynamics of the atmospheric circulation, e.g. subsidence in the higher atmosphere. These 2-year records are also a good benchmark for model evaluation. We have especially shown that the isotopic composition of water vapor measured at the surface is a powerful tool to test the vertical dynamic of atmospheric models and the implementation of water isotopes for those that are equipped with them. In our case, we used it to test different horizontal resolutions which influence the representativity of the vertical dynamics and have important implication in the simulation of surface variations of water vapor $\delta^{18}O_v$. Our study highlights the importance to have high-resolution models (e.g. mesoscale models) equipped with isotopes to further study such abrupt isotopic events.

## Appendices:

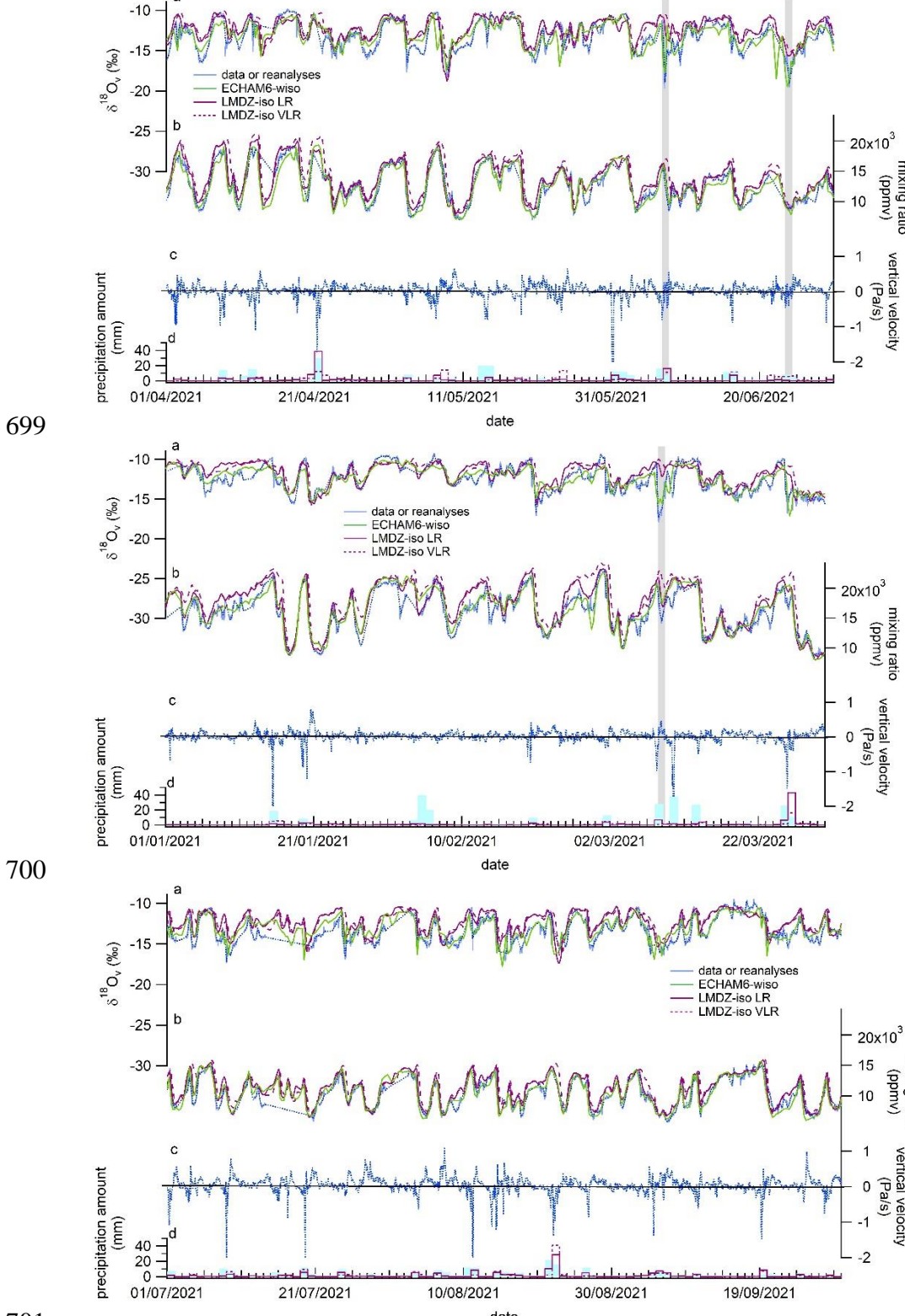




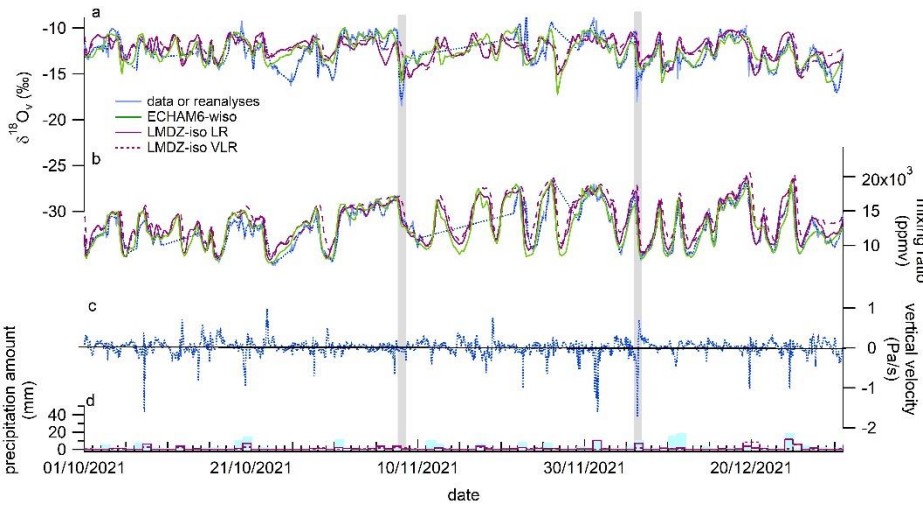

**Figure A1:** Model-measurement comparison (April 2020 – December 2021); a- $\delta^{18}O_v$ (light blue for
data on hourly average, dark blue for data resampled at a 6-hour resolution); b- water vapor mixing ratio
from our data set; c- vertical velocity; d- Precipitation amount. The grey shadings highlight the negative
$\delta^{18}O_v$ excursions.

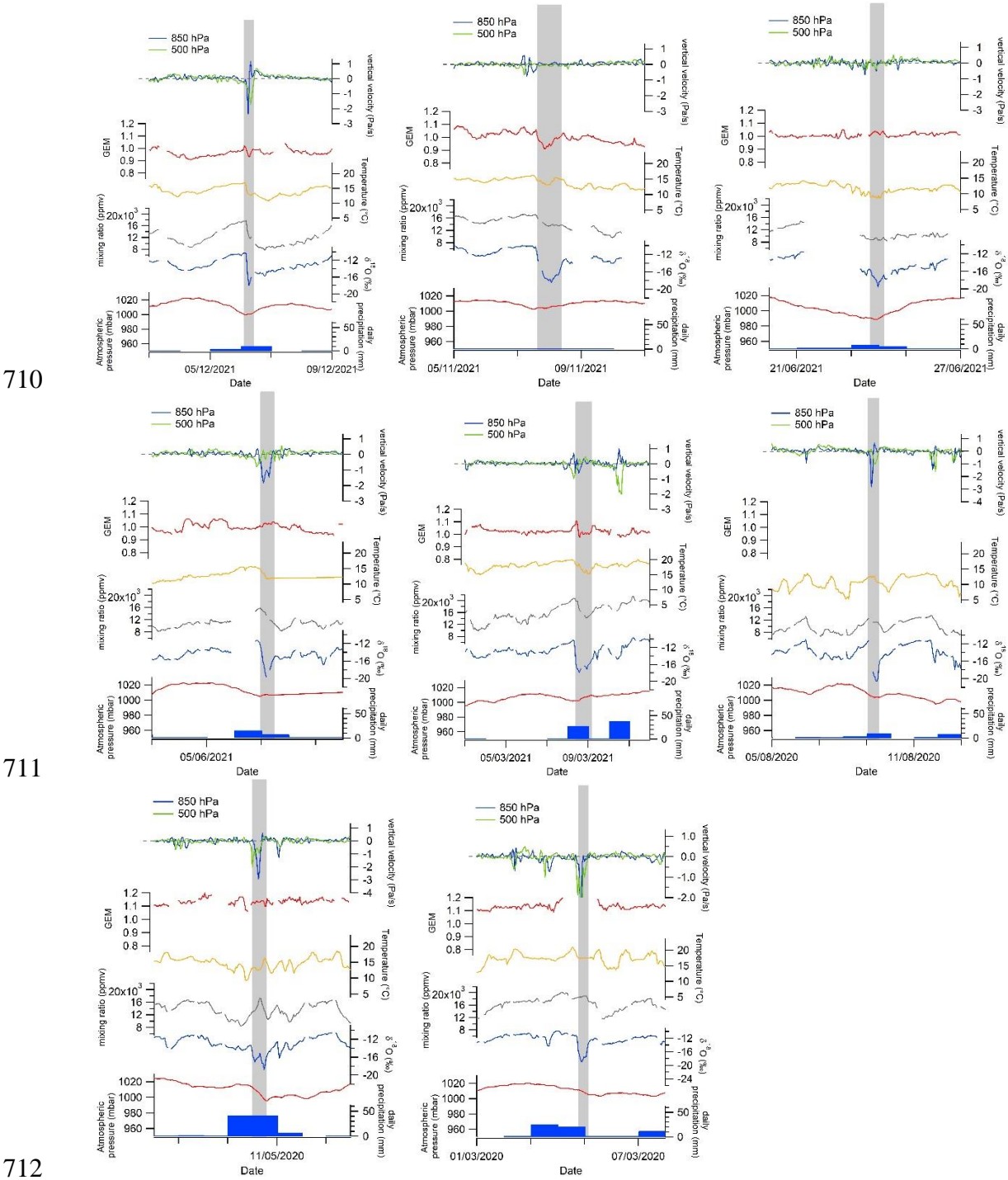




**Figure A2 :** Evolution of GEM, $\delta^{18}O_v$, water vapor mixing ratio, meteorological parameters

(surface temperature, surface atmospheric pressure, daily precipitation) measured by the

MeteoFrance weather station and vertical velocity from the ERA5 reanalyses at 500 and 850

hPa over the isotopic excursions between March 2020 and December 2021.



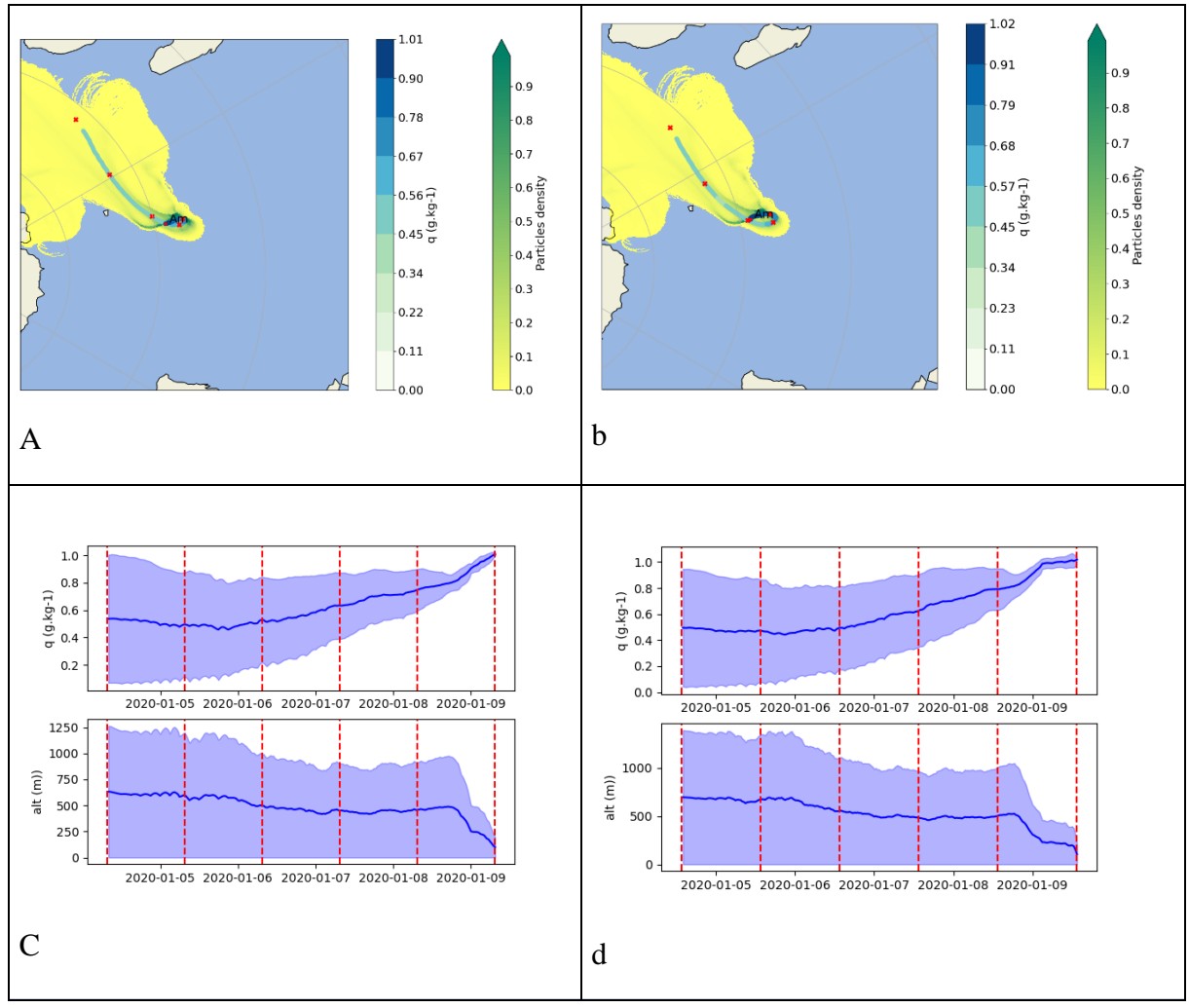



**Figure A3**: FLEXPART footprints of 5-day back trajectories for the event of January 9[th] 2020.
Panel (a) Latitude-longitude projection of the FLEXPART back trajectory footprint for January
9[th] 2020 at 7h30. The yellow to green colors on each grid point of these projections represent
the density of particles. The white to blue colors indicate the water vapor mixing ratio on the
humidity weighted average back-trajectory. Each red point indicates the location of the average
back-trajectory for each of the 5 days before the date of the considered event. Panel (b) Same
as a for January 9[th] 2020 at 13h30. Panel (c) Top shows the evolution of the water vapor mixing
ratio of the back trajectories for January 9[th] 2020 at 7h30; bottom shows the altitude evolution
of the back trajectory for January 9[th] 2020 at 7h30. Panel (d) same as panel (c) for January 9[th]
2020 at 13h30.

732

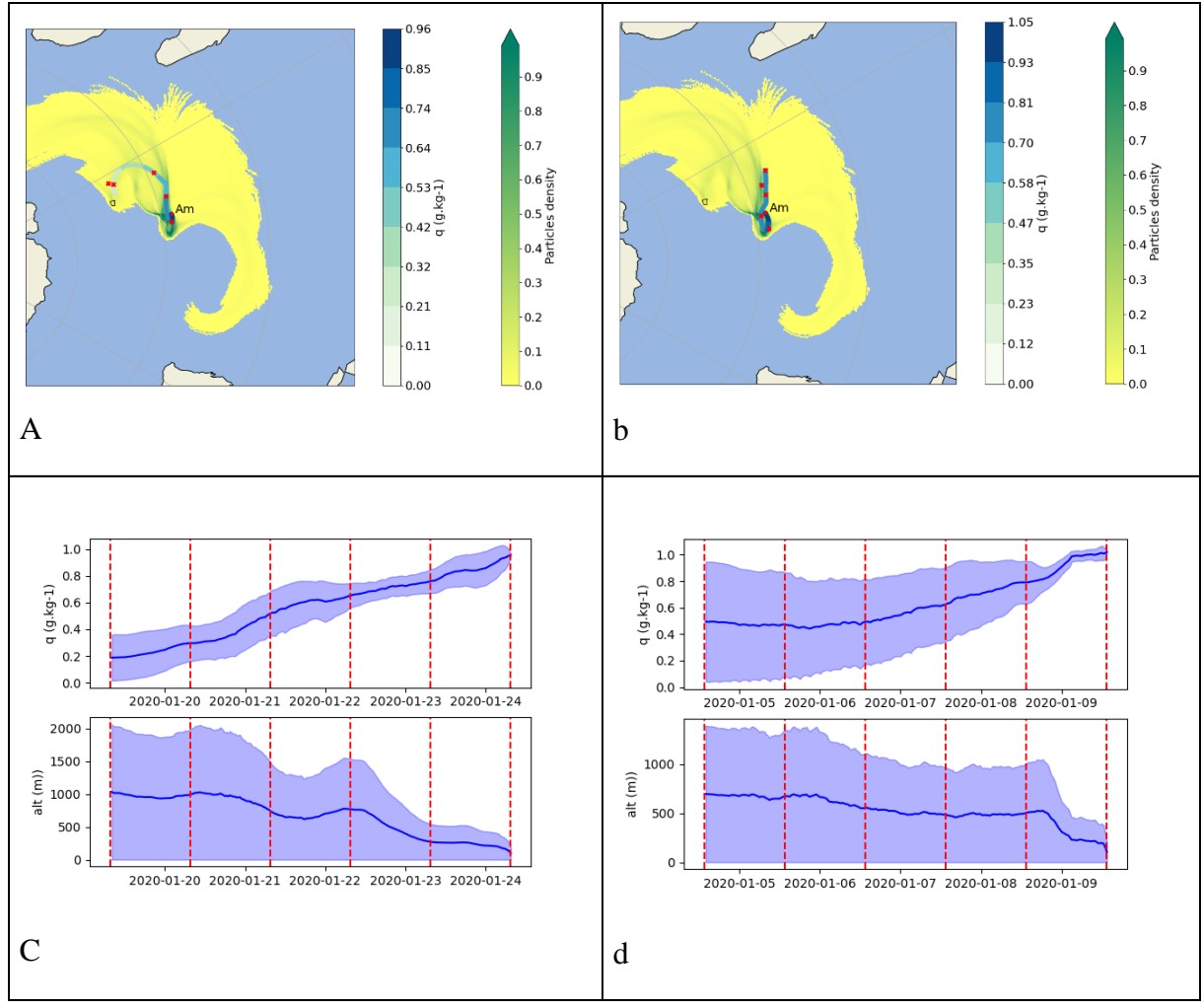

733
734

**Figure A4**: FLEXPART footprints of 5-day back trajectories for the event of January 21[st] 2020. (a) Latitude-longitude projection of the FLEXPART back trajectory footprint for January 21[st] 2020 at 7h30. The yellow to green colors on each grid point of these projections represent the density of particles. The white to blue colors indicate the water vapor mixing ratio on the humidity weighted average back-trajectory. Each red point indicates the location of the average back-trajectory for each of the 5 days before the date of the considered event. (b) Same as a for January 21[st] 2020 at 13h00. (c) Top shows the evolution of the water vapor mixing ratio of the back trajectories for January 21[st] 2020 at 7h30; bottom shows the altitude evolution of the back trajectory for January 21[st] 2020 at 7h30. (d) same as (c) for January 21[st] 2020 at 13h00.





**Data availability:** AMS L2 GEM data (https://doi.org/10.25326/168) are freely available (Magand and Dommergue, 2022) at https://gmos.aeris-data.fr/ from national GMOS-FR website data portal coordinated by IGE (Institut des Géosciences de l'Environnement, Grenoble, France; technical PI: Olivier Magand) with the support of the French national AERIS-SEDOO partners, data and services center for the atmosphere (last access: 08 December 2022). Hg species measurements belong to international monitoring networks (http://www.gos4m.org/). Water isotopic data and modeling outputs are available on the Zenodo platform (https://zenodo.org/record/8164392; https://zenodo.org/record/8160871).

**Acknowledgements:** We deeply thank all overwintering staff at AMS and the French Polar Institute Paul-Emile Victor (IPEV) staff and scientists who helped with the setup and maintenance of the experiment at AMS in the framework of the GMOStral-1028 IPEV program, the ICOS-416 program and the ADELISE-1205 IPEV program. Amsterdam Island Hg0 data, accessible in national GMOS-FR website data portal were collected via instruments coordinated by the IGE-PTICHA technical platform dedicated to atmospheric chemistry field instrumentation. GMOS-FR data portal is maintained by the French national center for Atmospheric data and services AERIS, which is acknowledged by the authors. The LMDZ-iso simulation were performed thanks to granted access to the HPC resources of IDRIS under the allocations 2022-AD010114000 and 2022-AD010107632R1 and made by GENCI. We deeply thank Sébastien Nguyen (CEA, LSCE) for his help and support in running LMDZiso simulation.

**Funding:** This work benefited from the IPSL-CGS EUR and was supported by a grant from the French government under the Programme d'Investissements d'avenir, reference ANR-11-IDEX-0004-17-EURE-0006, managed by the Agence Nationale de la Recherche. This project has also been supported by the LEFE IMAGO project ADELISE. Amsterdam Island GEM data, accessible in national GMOS-FR website data portal have been collected with funding from European Union 7th Framework Programme project Global Mercury Observation System (GMOS 2010-2015 Nr. 26511), the French Polar Institute IPEV via GMOStral-1028 IPEV program since 2012, the LEFE CHAT CNRS/INSU (TOPMMODEL project, Nr. AO2017-984931) and the H2020 ERA-PLANET (Nr. 689443) iGOSP program. This work is part of the AWACA project that has received funding from the European Research Council (ERC) under the European Union's Horizon 2020 research and innovation programme (Grant agreement No. 951596). The ERA5 reanalyses files for the ECHAM6-wiso nudging have been provided by

the German Climate Computing Center (DKRZ). The ECHAM6-wiso simulations have been
performed with support of the Alfred Wegener Institute (AWI) supercomputing centre.

**Author contributions:** AL designed the study and analyzed the data together with FV, CS, EF,
OM. OC installed the water vapor isotopic analyzer in Amsterdam Island and OJ was in charge
of the data calibration. BM and FP performed the measurements of the isotopic composition of
the precipitation samples. CA analyzed the modeling outputs, realized most of the simulations
and performed model-data analyses. CLDS performed the back trajectory analyses with help
from MC. OM, AD and YB provided expertise on GEM analyses and interpretation. AC, CR,
ND and MW provided model simulations. AL wrote the paper with contribution of all
coauthors.

**Competing interests:** One of the coauthors (AD) is a member of the editorial board of
Atmospheric Chemistry and Physics.

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
