# Peer review of "Abrupt excursions in water vapor isotopic variability at the Pointe"

_EGUsphere, 2023_

## Referee Comment (RC2)

Review egusphere-2023-1617

**Abrupt excursion in water vapor isotopic variability during cold fronts at the Pointe Benedicte observatory in Amsterdam Island**
*Amaëlle Landais,\*, Cécile Agosta,\*, Françoise Vimeux, Olivier Magand, Cyrielle Solis, Alexandre Cauquoin, Niels Dutrievoz, Camille Risi, Christophe Leroy-Dos Santos, Elise Fourré, Olivier Cattani, Olivier Jossoud, Bénédicte Minster, Frédéric Prié, Mathieu Casado, Aurélien Dommergue, Yann Bertrand, Martin Werner*

The manuscript "Abrupt excursion in water vapor isotopic variability during cold fronts at the Pointe Benedicte observatory in Amsterdam Island" presents two years of measurements of stable water isotopes in water vapour from Amsterdam Island. This time series is used together with measurements of gaseous elemental mercury and meteorological variables to analyse cold frontal passages at Amsterdam Island. Additionally, model simulation with ECHAM6-wiso and LMDZ-iso are evaluated using the measurements at Amsterdam Island.

This manuscripts presents a unique time series of stable water isotopes in water vapour in the under-sampled area of the Southern Indian Ocean. The calibration of the water vapour isotope data follows established standards in the community. While the dataset presented here is of unique coverage, the relevance of the analysis and the conclusions is not convincing. Furthermore, the figures presenting the analysis are often difficult to read and many important information is in the supplement figures. In its current state, the manuscript lacks a motivation for the presented analysis and how it improves the understanding of the atmospheric water cycle. It might be worth considering a submission to Earth System Science Data (ESSD) instead of ACP.

**Major comments:**

- **Model-measurement comparison**
  The comparison of ECHAM6-wiso and LMDZ-iso with the measurements at Amsterdam Island leads to the main conclusion that "the isotopic composition of water vapor is a powerful tool to identify aspects to be improved in the general circulation models, such as the horizontal resolution which may influence the representativity of the vertical dynamics." Why are *isotope-enabled* models needed to illustrate that horizontal resolution influences vertical dynamics in GCMs? This is, for example, already evident when comparing the vertical wind fields in Fig.7. Which "aspects to be improved", other than the horizontal resolution, were identified in this study?

- **Cold front analysis**
  The analysis is based on d18Ov excursions which are described as cold front events. The selection and analysis of these excursion with respect to cold frontal dynamics on a synoptic scale is missing. It is therefore difficult to interpret the described events with respect to the large-scale dynamics. In detail:
  - The analysis focuses on 11 events of cold fronts. These fronts and their spatial structure is not described in the manuscript and there are no synoptic figures describing a typical situation during a cold front passage, except for a few weather analysis charts in the supplementary information. How are the cold fronts identified? What are the properties of the cold fronts? How much of the annual precipitation is represented by cold frontal precipitation?
  - The 11 events are chosen using the following criteria: " The green rectangles indicate the period with (1) correlation coefficient >-0.5 between d-excess and d18O of water vapor and (2) occurrence of a negative excursion in water vapor d18O." There are (from eye) other events that could fall into these criteria. For example, before the event in ~March 2021 (6[th] green rectangle in Fig. 3), there is an event agreeing with criteria (1) and showing a strong decrease in d18Ov. Why are other events not included? And how is a negative excursion in d18Ov defined?
  - The analysis of the 11 events is mainly qualitative, and it is difficult to follow the description of the 4 events from 01-03/2020. All map plots and most of the vertical cross sections of these events are in the Supplementary Information, which makes it difficult to understand the synoptic situation during the events and the model performance. Further, it is not clear why these four events where chosen for a detailed description. Also, the description of the events is scattered in different paragraphs of

Sections 3 and 4. The paragraphs should be better structure to lead the reader through the evolution of the cold front events.

**Further general comments:**

- The section headings are not very specific. While Section 3 "Data description" has many short subsections, Section 4 "Discussion" has no subsection while introducding a lot of information and new analyses.

- Water isotope measurements:
  Various information is missing in the description of the water isotope measurements:
  - What was the material and length of the inlet line to the Picarro instrument? Was the inlet line heated?
  - Was the water vapour mixing ratio measured by Picarro calibrated? How does it compare to other humidity measurements on the Island?
  - When was the humidity-isotope dependency calibration done and what kind of calibration device was used?

**Detailed comments:**

Lines 68-69: What is (18O/16O) and (D/H)? Does this represent the isotopic ratio?

Lines 90-93: "For this objective, several instruments have been installed either in observatory stations ... or on boat …".
Is there a reason that this summary omits aircraft measurements?

Lines 97-99:  "Such data comparison enables one to test the performances of the models either in the simulation of the dynamic of the atmospheric water cycle or in the implementation of the water isotopes."
I agree with this statement but I don't see how this study adds any new knowledge on model performance or isotope parametrisations. Can you elaborate further?

Line 101-102: "This region is poorly documented with present-day observations despite its primary importance in governing CO2 sinks"
Do you have a reference for this statement?

Line 102-105: "Moreover, we lack precise descriptions of atmospheric processes associated with cloud microphysics and surface-atmosphere exchange in polar regions, and the evolution of westerly wind locations and strength (Fogt and Marshall, 2020)."
Why is this relevant for the presented study? The study site lies in the mid-latitudes.

Line 134: "Climate is temperate, generally mild with frequent presence of clouds."
What do you mean with frequent presence of clouds?

Lines 149-150: "$CO_2$, CO, $CH_4$ and Hg species have been continuously measured since 1980, 2014, and 2012 respectively."
There are four species but only three years are mentioned. It is not evident which species belongs to which year.

Lines 160-167: The elevation of the two meteorological stations at Pointe Bénédict observatory is given in meters agl, while the elevation of the station at Martin-de-Viviegs is given in meters asl. This makes it difficult to compare the elevations. Further, this paragraph gives a lot of detail on variables that are not shown later on.

Line 165: What is IGE?

Line 187: What is STP?

Line 202: "high altitude air masses (lower/ upper troposphere, or even above)"
This is very unspecific. What do you mean with lower/upper troposphere?

Lines 204-205: " As mentioned above, mercury in the atmosphere is detected in three defined forms:"
This has not been properly introduced earlier.

Line 233: "outside at ~ 6 m above ground level."
What is this relative to m agl/asl, i.e. compared to the other measurements?

Line 279: What do mean with "quadratic error"?

Line 290: Why are you starting the trajectories at 100m a.s.l.?

Line 324: "high spatial": 0.9° horizontal resolution is high compared to the LMDZiso simulation of this study but low compared to convection permitting climate simulations. I would therefore skip "high".

Line 389: "d-excess of the precipitation". I don't see this in Fig. 3.

Line 395, 395, 399: What is $R^2$? The correlation coefficient R to the power of 2? What kind of correlation are you calculating? $R^2$ is used before R is introduced.

Line 399-400. "...(R is calculated continuously from hourly records in 8 consecutive days)..."
Do you mean that you used an 8-day moving window?

Line 402-403: "d-excess$_v$" has not been introduced.

Lines 434 – 436: "...the agreement with measured precipitation amount is better for ECHAM6-wiso (R2 = 0.45) than for LMDZ-iso (R2 = 0.08 – 0.13 for VLR – LR)…"
The correlation of LMDZ-iso with the measurements is close to zero, i.e. there seems to be nearly no agreement. The statement that the agreement with measured precipitation is better for ECHAM6-wiso than for LMDZ-iso seems weak in this context.

Lines 437-439:  "...they are in general more strongly expressed in the data series than in the model series which is only partly due to the hourly resolution of the d18Ov record compared to the 3h and 6h resolution of the outputs of the LMDZ-iso…)."
What is the basis of this conclusion?

Linbes 450-451: " They always occurred during low pressure periods (atmospheric pressure below 1005 mbar)."
What is the synoptic situation leading to this low pressure and cold fronts?

Lines 501-503: "However, for the 11 events highlighted above, the d18Ov vs qv evolution follows an evolution characteristic of remoistening processes, i.e. a curve standing below the curve of the d18Ov vs qv evolution observed for the rest of the series…"
The single events show a much steeper evolution in the d18Ov-qv diagram than the remoistening curve. Why is this?

Lines 504-506: "Since relative humidity is relatively high during these events (values given in Table 1 compared to a mean value of 77 %), it more likely reflects rain-vapor diffusive exchanges than rain evaporation."
Are you referring to the relative humidity at the surface? How about relative humidity above that will also influence the interaction of the rain with its surroundings?

Line 519ff.: As the trajectories are only shown in the supplement, it is difficult to follow this paragraph. The beginning of the paragraph leads to think that the trajectories indicate that subsidence is import but in the end the conclusion is that "back trajectories are however not supporting systematic subsidence for other cases".

Lines 526-527: "... the maximum altitude of the envelope of the back trajectories increases from 5,000 to 8,000 m..."
What it the mean/standard deviation of the maximum trajectory height? How many days before arrival are the trajectories at their maximum height? How relevant is this for the isotopic composition upon arrival? E.g. if the trajectories descend over the ocean and take up moisture, their maximum height before the moisture take-up is less relevant for the isotopic composition at Amsterdam Island.
If you are using the full 10-days backward trajectories to calculate the maximum altitude, I don't think that the maximum altitude is a good measure of subsidence in front of the cold front.

Lines 543-555: " There is no evidence for changes in the horizontal advection of air over the 11 particular events from the observation of wind direction around these cold front events."
How is the cold front identified? Does it divide different air masses? A cold front normally implies a horizontal transport of air, why is this not the case for these cold fronts?

Lines 556-558: "Such abrupt d18Ov events can hence be used as a test of the performances of general circulation models equipped with water isotopes."
What was learned about the performance of the GCMs involved in this study? Was it necessary to include d18O in such a performance test instead of just using traditional humidity variables (e.g. relative humidity, specific humidity, precipitation)?

Line 562: What is "SOM"?

Lines 559-584: As both isotope-enabled models were nudged to ERA5 dynamics, it is to be expected that the GCMs reproduce the ERA5 reanalysis wind fields rather well with some caveats due to the lower horizontal resolution. This paragraph (and Fig.7) is mostly describing the smoothing of ERA5 due to the coarser resolution of the isotope-enables GCMs. Why do we need isotope measurements to see the effect of a coarser horizontal resolution? What do we learn about the GCM performances by decreasing the horizontal resolution?

Lines 586-590: "A rain event is associated with a strong ascending column in which d18Ov is depleted by progressive precipitation during the ascent and by interaction between rain and water vapor. This ascending column is coupled to the subsidence of d18Ov depleted air at the rear of the event which is pushed toward Amsterdam Island through a south west advection of cold air."
How is the isotopic composition of the subsiding air behind the cold front connected to the progressive precipitation during the ascent? Can the ascending and descending column be differentiated in the d18O excursions?

Line 621: "hours/days"
Is there a cold front passage that has a duration of several days?

Lines 635-640: "This study highlighted the added value of combining different data from an atmospheric observatory to understand the dynamic of the atmospheric circulation. The two-year records are also a good benchmark for model evaluation. We have especially shown that the isotopic composition of water vapor is a powerful tool to identify aspects to be improved in the general circulation models, such as the horizontal resolution which may influence the representativity of the vertical dynamics."
As also mention above, why are stable water isotopes needed to show that the horizontal resolution may influence the vertical dynamics? The vertical cross sections of vertical wind speed (Fig. 7) illustrates this already quite well.

Line 642: Data availability: The isotope measurements are poorly documented on the Zenodo platform, and the dataset does not include the water vapour mixing ratio.

**Figures**:
General: The figures are often difficult to read, especially the described phenomena are small (e.g. d18O excursions of a few hours in a 2-year or 3 month timeline). Additionally, the colors are not color-blinded friendly and the caption are not concise.

Fig.1: Is this figure needed? Fig.1 is not mentioned in the text.

Fig.3: x-axis too coarse, light green shading difficult to see.

Fig. 4: Colors red/blue/green are not colorblind friendly. The caption text includes many repetition and should be improved. A legend in the figure could improve the readability.

Fig. 6: What is φ=0.025?

Figure 8: What is SBL? The ascent of air in front of the cold front rises nearly vertical at a constant longitudinal position. As a cold front is moving system (mostly associated with an extratropical cyclone), the ascent does not occur at a constant location (in latitude and longitude). Further, all precipitation seems to fall in front of the cold front, which is unlikely. How is the subsidence at 100°E and ascent at 90°E related to the cold front? What does a composite of precipitation and d18Ov for all cold front events look like? Can it reproduce the schematic as shown in the "surface box"?

**Technical comments**

Generally: there are many abbreviations in the text that are only used a few times. Can you reduce the number of abbreviations?

Line 60-61 (and many others): The references are not in chronological order.

Line 65-66: "We express the abundance of the heavy isotopes D and 18O with respect to the amount of light isotopes 16O and H in the water molecules…" should be "We express the abundance of the heavy isotopes D and 18O with respect to the amount of light isotopes **H and 16O, respectively,** in the water molecules…"

Line 68: Eq. 1 has strange symbols (squares).

Line 88-89: "water cycle processes such as  "

Lines 106-109: "Over the previous years, we have installed 3 water vapor analyzers on Reunion Island at the Maïdo observatory (21.079°S, 55.383°E, 2160m) (Guilpart et al., 2017) and in Antarctica (Dumont d'Urville and Concordia; (Leroy-Dos Santos et al., 2021; Bréant et al., 2019; Casado et al., 2016). "
Check usage of brackets.

Line 133: "from the nearest lands, Madagascar, and"

Lines 140-141: "...and were continuously monitored at the site since 1960…"

Lines 145-150: The section is very difficult to read, the websites and datasets should better be included as references. Same for link to AERIS on line 178 and 200.

Line 178: "AERIS ((Magand and Dommergue, 2022))"

Lines 180-181: "instrument models (Tekran Inc., Toronto, Canada) (Angot et al., 2014; Slemr et al., 2015, 2020; Sprovieri et al., 2016; Li et al., 2023). "

Line 202: "may possibly" Doubling, omit either.

Lines 211-221: This sentence is too long. Can you divide into two sentences?

Line 228: "The isotopic composition of near-surface water vapor (d18Ov and dDv )"

Line 242: "The calibration of the data is performed in **several** steps following previous studies"

Lines 301-303: "...identical **to** the atmospheric setup of IPSL-CM6A (Boucher et al., 2020) used for phase 6 of the Coupled Model Intercomparison Project (CMIP6, (Eyring et al., 2016)). "

Line 373:  "… very close to the one observed in Angot et al. (2014)."

Line 374: "During the period (2020-2021) of water vapor isotope measurements in AMS…"
Do you mean: During the period 2020-2021 of water vapor isotope measurements in AMS.. ?

Lines 390-391: "The annual cycles are also not visible…" Do you mean: "An annual cycle is not visible…" ?

---

## Editor Comment (EC1)

**Editor comments on egusphere-2023-617**

Since I have trouble finding a second referee, I will provide a referee comment as editor of this manuscript. However, I will continue trying to find an additional referee so that there will be still two independent referee comments.

This manuscript presents 2 years of continuous water vapor isotope observations obtained at Amsterdam Island in the southern Indian Ocean. The authors find that the observed data do not show a clear seasonality, and that temporal isotopic variations in the data set are associated with synoptic-scale atmospheric weather patterns. The authors highlighted that the abrupt negative water vapor isotope excursions occurred in connection with cold front passages. To understand the physical isotope related mechanisms that cause the isotopic depletion, the authors performed several analyses, such as: comparison with the measurement of gaseous elemental mercury as an indicator of subsidence, back trajectory analysis, and the model experiments using the isotope enabled atmospheric general circulation models. Based on the results, the authors conclude that both the vertical subsidence of water vapor with depleted isotopic content and the isotopic exchange process between rain drops and the surrounding vapor were responsible for the negative excursion of the water vapor isotopes Further, by comparing two isotope enabled general circulation models, the authors find that a higher resolution model is needed to reproduce the observed negative excursions.

The main contribution of this manuscript to the scientific community is to provide new observational data for the southern Indian Oceanic region, where the measurements are sparse. Their interpretation that the abrupt isotopic depletion at the surface is caused by subsiding air with depleted isotopic contentt sounds reasonable. Convective precipitation usually occurs in connection with cold fronts. Further, sudden isotopic depletion in surface vapor is known to occur during or just after the passage of convective precipitation such as squall lines and mesoscale convective systems. In addition, previous studies have also highlighted the contribution of depleted isotopic moisture from the free troposphere. Additionally, taking into account the model experiment, it is well known that the mesoscale or finer resolution model simulations are required to resolve the convective clouds associated with the cold front. Thus, it makes sense that the only higher resolution model can properly reproduce the observed negative isotopic excursions.

**General comments**

The quality of this draft is above standard and the subject is suitable for publication in ACP. However, I think it may not be easy for every reader to understand this manuscript. Especially for those who are not experts on water isotope processes. I recommend major revisions of the manuscript so that the contents and results of this study become better understandable for any potential reader.

Further, the graphical presentation is rather of low quality. Most of the figures should be improved (increasing the size of the figure itself as well as the font size and the line thickness). Some of the rather important figures of this study are in the supplement and may be moved to the main text or in an appendix to the main text (so that these do not appear in an extra document).

The usage of terms etc. should be done more consistently one way or the other. In some occasions the term "vapor mixing ratio", "mixing ratio" , "water mixing ratio" or "water

vapor mixing ratio" are used making reading this manuscript very confusing. Then the notation "v" as subscript is used in some occasions, but in many others not. I would suggest to use "v" and "p" to differentiate between vapor and precipitation water vapor and water vapor isotopes throughout the manuscript.

**Specific comments**

Abstract: The abstract is quite confusing and some transitional sentences, e.g. the first and second paragraph as well as a sentence stating that you use two models for comparison are missing. What is the main focus of your study? The two paragraphs feel like two independent abstracts. One describing the measurements and the conclusions you derive from these and then the measurement-model comparison and the according results of this part of the study. The problem with the focus of study continues throughout the entire study. Additionally, several technical issues in the abstract need to be corrected (see below the list of technical corrections). Since there were so many issues I provide you here a corrected/improved version of your manuscript as a suggestion how it would read much better:

*In order to complement the picture of the atmospheric water cycle in the Southern Ocean, we have continuously monitored water vapor isotopes (δ18O) since January 2020 on Amsterdam Island in the Indian Ocean. We present here the first 2-year-long water vapor isotopic record monitored on this site. We show that the vapor isotopic composition, as expected in marine boundary layers, largely follows the vapor mixing ratio. However, we detect 11 cold front periods lasting for a few days where there is a strong degradation of correlation between δ18O and water vapor mixing ratio. These periods are associated with abrupt negative excursions of δ18O, often occurring toward the end of precipitation events. Six of these events show a decrease in gaseous elemental mercury suggesting subsidence of air from higher altitude.*
*To proof this hypothesis we additionally consider model simulations of these processes, although accurately representing the water isotopic signal during these cold fronts is a real challenge for the atmospheric components of Earth System models equipped with water isotopes. We compare here two of these models. While the European Centre Hamburg model (ECHAM6-wiso) was able to reproduce most of the sharp negative water vapor δ18O excursions, the Laboratoire de Météorologie Dynamique Zoom model (LMDZ-iso) at 2° (3°) resolution was only able to reproduce 7 (1) of the negative excursions. Based on this detailed model-data comparison, we conclude that the most plausible explanations for such isotopic excursions are rain-vapor interactions associated with subsidence at the rear of a precipitation event.*

P2, L51: Also in the introduction still the question remains what the purpose of your study is. Is it to confirm/better understand the measurements or to test the capability of the isotope enabled models to reproduce the isotopic processes?

P5, L113-114: The Durmont d'Urville and Concordia stations in Antarctica are not really in the Indian sector, but rather in the Pacific sector. Thus, how these are suitable for understanding the atmospheric water cycle over the Indian basin of the Southern Ocean does not become clear.

Figure 1 caption: What is meant with Magand? Add a link or reference?

P8, L223: What do you mean here with "low altitude"? That the observatory is located at low altitude? Or that the air from higher altitude is transported down to lower altitudes?

P8, L228: Here now subscripts "v" used, but before not.

P9, Figure 2 caption: Why is here the anomaly used?

P9, L247: What exactly are these standards and how are these derived? Not clear! Are these typical relationships between these species? Are these documented somewhere else?

P9, L247: What exactly is denoted by these numbers? The data range?

P9, L254ff: I could not follow you. Why does the data need to be corrected? What did you find here in the relationship that is not as it should be?

P15, L402: How do you know that these peaks occurred during a cold front? No analyses of meteorological parameters indicating a front passage are shown or discussed here.

P14, L403-404: You only picked the 11 excursions with low correlation coefficient between d18Ov and qv. However, according to Fig. 3, there are other negative excursions of d18O besides these 11 cases. If the goal of this study is to show the isotopic features associated with the passage of the cold fronts, the authors should rather pick up the events from the weather chart showing a cold front passage, not from the low correlations alone.

P16, Figure 4: Add a legend so that we can understand which color indicates which data just by looking at it.

P17, L454-455: Sentence not clear since it is grammatically incorrect. Please rephrase.

P18, L475: I still have not seen how you can be sure that there was a cold front passage. How have the cold fronts been detected?

Line 504-506: Rain evaporation occurs under the cloud base, moistening the boundary layer. So, the authors should not underestimate the role of rain evaporation because of the high relative humidity near the surface.

P19, Figure 6: This figure is not clear at all and needs more explanation. The plot represents the observations in the boundary layer, but the theoretical curves are the isotopic changes in the free troposphere? At least  Noone (2012) used them to investigate processes in the tropical mid troposphere. What exactly is meant with "inspired" by Noone (2012)? How have these curves been derived? Do you take these from the Noone (2012) paper?  Have you calculated/estimated these yourself?

P20, L523ff: Since subsidence is an important aspect of this study and you use Figure S1-S3 for the discussion, I don't understand why these figures are in the supplement instead of in the main part of the manuscript.

P21, L623: I still haven't seen any proof that there has been a cold front passage.

P25, L640: It is still not clear what the function of the models are. Are these only used to be evaluated or are these also used to understand the processes behind the peaks in the d18O time series?

P26, L682: What do you mean with "of the s"?

Supplement: The supplement contains too many figures. I think not all of them are really necessary and the number could be reduced. Further, the formatting should be the same as for the ACP paper, that means no underlined headers and the same style for the figures (no underline of the figure caption title and no italic text for the caption text.)

**Technical corrections**

P1, L33: in → on

P1, L34: remove coordinates. It is not necessary to provide these in the abstract.

P1, L35: either add "water" or just write "isotopic".

P1, L36: add "water"

P1, L37: Rather "detect" than "evidence".

P1, L38: move "water vapor" before "mixing ratio".

P1, L39: Omit "water vapor".

P1, L44 and 45: Abbreviations "ECHAM5-wiso" and "LMDZ-iso" not introduced.

P1, L46: detail → detailed

P4, L85: stable isotopes in water vapor → stable water vapor isotopes

P4, L85: dynamic → dynamics

P4, L89: repetition of water cycle processes.

P4, L90: Delete "For this objective" (since this makes no sense in context with the previous sentence) and start sentence with "Several instruments…..".

P4, L97: comparison enables -> comparisons enable

P4, L108: Add coordinates and elevation for the Durmont d'Urville and Concordia stations.

P4, L111: in → at

P5, L113: delete "the" before water vapor

P5, L114: Rather "measured" or "observed" than "documented".

P5, L118: in→ on

P5, L120: It should either read "……to help with the interpretation" or "interpreting" of isotopic records.

P5, L123: delete "methodology" and change "trajectory" to "trajectories".

P5, L124: remove "the" and "water vapor" before and after, respectively, $\delta^{18}O$ or write "water isotope $\delta^{18}O$".

P5, L125: "expressed strongly in the water vapor isotope record" -> delete? This seems to be a repetition to what is said in the previous sentence.

P5, L131: degree sign not correct.

P5, L138: Move "respectively" at the end of the text in parentheses.

P6, L145: concentration → concentrations

P6, L145: delete "species"

P6, L147: Rather "belong" or "report to" than "respond".

P6, Fig. 1 caption: Full stop at the end of the last sentence is missing.

P7, L180: models → model

P7, L186. Measurement → measurements

P7, L197: Abbreviations CAMNET and AMNET have not been introduced.

P7, L199: dataset → datasets

P8, L203: add "air" so that it reads "boundary layer air"?

P8, L206: Abbreviation PBM not introduced.

P8, L207: remove "species".

P8, L211-213: Sentence makes no sense ("Even if"……."is still poorly understood"). Please check and rephrase.

P8, L223: Rather "excursions" than "intrusions".

P9, Figure 2 caption: Rather "Correlation between" than "Dependency of".

P10, L277: at LSCE → to LSCE

P10, L277: What does the abbreviation LSCE stand for?

P10, L283: Rather "calculated with" than "assessed".

P10-11, L288-291: Rephrase. Not FLEXPART is calculating, but you are calculating with FLEXPART. So the sentence should read "Back trajectories…………..were calculated with FLEXPART".

P11, L292: add "the" →  of the FLEXPART

P11, L292: in→ as

P11, L293: probability functions?

P11, L299 and L317: Introduce abbreviations of the models LMDZ and ECHAM6.

P12, L324: at → with a

P12, L329: gases → gas

P12, L332: Parentheses around the reference are not correct.

P12, L336: in Cauqoin -> by Cauqoin

P13, Figure 3 caption: In several occasions "water vapor" can be omitted. Use subscripts "v" and "p" instead.

P14, L353: In several occasions: add "vapor"? Or do you refer here to total water?

P14, L368: in → at

P14, L382. Add "a" → in a few hours

P14, L383: delete "water vapor".

P15, L389: delete "the" before "precipitation".

P15, L395: Add "vapor". Or do you mean total water?

P15, L398: on → in

P15, L399: It should rather read "……..based on the / estimated from the correlation of d18Ov and qv".

P15, L401: "which is" rather than "which are"?

P16, L408: Data model comparison → Model-measurement comparison

P16, L408: delete "water vapor"

P16, L423: set of data → data set

P16, L426: on → in

P16, L427 and 429: What mixing ratio? Water vapor?

P16, L432: two-year → 2-year and add "time" before "series" so that it reads "time series".

P16, L437: I would rather call it "grey shaded areas" than "grey rectangles".

P16, L437: data series → measurement time series

P17, L441: two-year → 2-year

P17, L447: Section header should appear without being underlined.

P17, L448: two-year → 2-year

P17, L449: decorrelation → anti-correlation ?

P17, L450: Also here, I would rather call it "shaded areas" than "rectangles".

P17, L452: the series presented on → the time series presented in

P18, L476: water mixing ratios → water vapor mixing ratio

P19, L484: mixing ratio ….of water vapor -> water vapor mixing ratio

P19, L486: correct parentheses around Noone reference.

P21, L539: movement of the atmosphere → movement of the (atmospheric) air

P21, L557: ….a test of the performance …… → as a test bed for the performance

P21, L558: general circulation models equipped with water isotopes -> water isotope enabled general circulation models.

P21, L562: What does SOM stand for?

P24, L619: add "vapor" → water vapor mixing ratio

P25, L624: same here

P25, L636: two-year → 2-year

Figure S1 caption, L73: mixing ratio → water vapor mixing ratio

Figure S1 caption, L69: water vapor $\delta^{18}$O → $\delta^{18}$O

Figure S1 caption, L80: grey rectangles → grey shaded areas

Supplement, P6, L87: flexpart -> FLEXPART

Figure S2 caption, L89: in → as

Figure S2 caption, L91: ten-day → 10-day

Figure S3 caption, L101: in → as

Figure S3 caption, L102-103 and Figure S3 caption, L114: ten-day → 10-day

---

## Author Comment (AC1)

We thank very much the editor and the two reviewers for their very detailed and helpful comments. We have addressed all comments below and are willing to submit a manuscript taking into account all comments as explained in the answers to comment below.

Many thanks again for your help.

**Review 1**

I would like to compliment the authors for having prepared such a well-written paper and I strongly recommend the paper for publication after a very few adjustments/corrections. In general, the interpretation and the discussion of results is sounding and easy to follow. All figures are clear (see my only comment on Fig.4).

>> Many thanks for this general comment

I only have one comment/question about the interpretation of the results. Why d-excess has been (almost) left out of the discussion? The authors clearly state that during the depletion events both $\delta^{18}O_v$-q and $\delta^{18}O_v$-dexcess correlations break down. But how d-excess signal looks like during the event? If the d-excess doesn't change much, it would provide support to the hypothesis of rain-vapor interaction close to equilibrium than to rain-evaporation and to atmospheric subsidence, since evaporation of raindrops and free tropospheric air are associated with high d-excess.

>> Many thanks for this suggestion. d-excess of water vapor is indeed not changing much over the $\delta^{18}O_v$ excursion. We thus agree that it may not be explained by strong rain drop evaporation at the surface and it is also in agreement with the relatively high relative humidity at the surface. We thus agree that it is then in better agreement with rain-vapor interaction close to equilibrium for the acquisition of this signal. Such rain-vapor interaction is indeed indicated on our summary on Figure 8 and we will add in the manuscript that the stable d-excess signal supports this interpretation.

Minor comments:

L149 Please rearrange number of gasses (4) and monitoring years (3).

>> We propose this new sentence:

"From 1980 to 2011, $CO_2$ and CO species have been monitored by non-dispersive infrared measurements (NDIR) systems. From 2012, dry air mole fractions of these two greenhouse species are currently measured by cavity ring down spectroscopy (CRDS) using commercial Picarro analyzers model G2401. Methane ($CH_4$) and nitrous oxide ($N_2O$) are also measured continuously using the same instrument model, but since 2014 and 2018 respectively, while Hg species are monitored since 2012 (see section 2.2.2.)."

L187 STP conditions: 273.15 K

>> Thank you, this will be changed.

Figure 4 Including a legend and reducing the size of the caption could improve readability.

>> Thank you, we propose the updated figure:

[Figure]

Figure 4: Data model comparison (January – March 2020); a- water vapor d$^{18}$O (light blue for data on hourly average, dark blue for data resampled at a 6-hour resolution); b- mixing ratio from our data set; c- vertical velocity; d- Precipitation amount. The grey rectangles highlight the negative d$^{18}$O excursions (note that in this figure the excursions of the 3$^{rd}$ and 9$^{th}$ of January 2020 are distinct while the distinction could not be done on Figure 3 because of the scale).

---

## Author Comment (AC2)

We thank very much the editor and the two reviewers for their very detailed and helpful comments. We have addressed all comments below and are willing to submit a manuscript taking into account all comments as explained in the answers to comment below.

Many thanks again for your help.

**Review 2**

This manuscripts presents a unique time series of stable water isotopes in water vapour in the under-sampled area of the Southern Indian Ocean. The calibration of the water vapour isotope data follows established standards in the community. While the dataset presented here is of unique coverage, the relevance of the analysis and the conclusions is not convincing. Furthermore, the figures presenting the analysis are often difficult to read and many important information is in the supplement figures. In its current state, the manuscript lacks a motivation for the presented analysis and how it improves the understanding of the atmospheric water cycle. It might be worth considering a submission to Earth System Science Data (ESSD) instead of ACP.

>> Many thanks for this review and detailed comments. The reason why we chose to submit to ACP is because we aim at combining for the first time water isotopes (data + 2 different models) and atmospheric species, here gaseous elemental mercury, over a long time series. The observation of concomitant water isotopes and Hg excursions suggests subsidence of air from high altitude.
We believe that water isotopes and elemental mercury records present an added value to the understanding of the dynamic of the atmospheric water cycle since they provide a surface diagnostic of processes happening higher in the atmosphere. They are very powerful for identification of change in vertical velocity which is a parameter which, otherwise, can only be reconstructed from the models. We agree with the reviewer that this objective was not clear enough in the manuscript, nor the added values of combining water isotopes, Hg and models and this will be corrected in the next version. Also, we will work on improving the readability (and colorscale) of the figures and move some figures (backtrajectories for example) from the supplement to the main text or in appendix (see also suggestions from the editor).

**Major comments:**
• **Model-measurement comparison**
The comparison of ECHAM6-wiso and LMDZ-iso with the measurements at Amsterdam Island leads to the main conclusion that "the isotopic composition of water vapor is a powerful tool to identify aspects to be improved in the general circulation models, such as the horizontal resolution which may influence the representativity of the vertical dynamics." Why are *isotope-enabled* models needed to illustrate that horizontal resolution influences vertical dynamics in GCMs? This is, for example, already evident when comparing the vertical wind fields in Fig.7. Which "aspects to be improved", other than the horizontal resolution, were identified in this study?

Our study aims at exploring the mechanism driving the variations of water vapor $\delta^{18}O_v$ and especially the 11 events identified in the water vapor $\delta^{18}O_v$. With the aim to understand the associated processes, the comparison of the data series with outputs of atmospheric components of Earth System models equipped with water isotopes is a very useful tool. In addition to propose a mechanism for the water vapor d18$O_v$ excursions through our model – data comparison, we could also propose some ideas to explain why some models could not reproduce the water isotopic excursions at the surface. Isotope implementation is quite similar in the 2 models and is probably not the reason for the differences, a result that we should highlight in the next version of the manuscript. Nudging is slightly different between the two models (vorticity nudging in ECHAM6-wiso, shorter relaxation time in LMDZ-iso) but as the reviewer rightly points out, the conclusion is that if isotopes are controlled during these events

by vertical dynamics, then it is probably necessary to have high-resolution models to unravel the mechanisms. We will adda couple of sentences saying that our model-data comparison validates the implemented physics of water isotopes and introduce in the conclusion the importance to have high-resolution models (mesoscale models) equipped with isotopes if one wants to study such events.

Finally, vertical wind fields from figure 7 are from reanalyses, themselves based on model. They were not measured. So we believe that our study, unique in this region, still has an added values by comparing a new measured data series (the records of water vapor $\delta^{18}O_v$ at the surface) with model outputs.

**• Cold front analysis**

The analysis is based on d18Ov excursions which are described as cold front events. The selection and analysis of these excursion with respect to cold frontal dynamics on a synoptic scale is missing. It is therefore difficult to interpret the described events with respect to the large-scale dynamics. In detail:

◦ The analysis focuses on 11 events of cold fronts. These fronts and their spatial structure is not described in the manuscript and there are no synoptic figures describing a typical situation during a cold front passage, except for a few weather analysis charts in the supplementary information. How are the cold fronts identified? What are the properties of the cold fronts? How much of the annual precipitation is represented by cold frontal precipitation?

>> This comment is very sound. Our aim was actually not to focus on cold fronts but on the isotopic record and especially on periods when the isotopic record is providing an added value to understand features associated with the atmospheric water cycle, that is to say a different signal than the meteorological signals (humidity, temperature). This is why we concentrate on the abrupt negative water vapor $\delta^{18}O_v$ excursions which are not seen in the humidity signal. We then simply observed that the periods during which we observed water vapor $\delta^{18}O_v$ negative excursions were also associated with a cold front but there are certainly many cold fronts that are not exhibiting any water vapor $\delta^{18}O_v$ excursions. We definitively need to make it more clear in the manuscript and we will also remove the term "cold fronts" from the title and in other parts of the text since it was misleading.

In the previous manuscript, we had identified the cold fronts from the synoptic figures. All weather charts corresponding to the 11 water vapor $\delta^{18}O_v$ negative excursions are shown below:

[Figure]

| | Day-1 of event at 00:00 UTC | Day of event at 00:00 UTC | Day+1 of event at 00:00 UTC |
|---|---|---|---|
| 03/01/2020 | | | |
| 09/01/2020 | | | |

[Figure]

| | | | |
|---|---|---|---|
| 24/01/2020 | | | |
| 05/03/2020 | | | |
| 10/05/2020 | | | |
| 09/08/2020 | | | |
| 08/03/2021 | | | |
| 07/06/2021 | | | |

[Figure]

**Figure R3** : Weather analysis charts provided once a day at 00:00 UTC by the Analysis Chart Archive service of the Australian Government Bureau of Meteorology http://www.bom.gov.au/australia/charts/archive/index.shtml.  Red dot on weather charts displays Amsterdam Island location.

The idea was to check for the presence of a cold front in a distance of 100 km around Amsterdam Island in a 48h period covering the time of the event. We indeed see that we systematically have cold fronts in the vicinity of the Amsterdam Island at the time of the water vapor $\delta^{18}O_v$ excursions. Still, in some cases, such as the 06/12/2021, there are cold fronts identified on the weather charts in the vicinity of the Amsterdam Island 1 day before and 1 day after the excursions but no clear cold front over Amsterdam Island at the exact time of the water vapor $\delta^{18}O_v$ excursion.

To address the occurrence of cold fronts in a more detailed manner, we propose here a second synoptic analysis with the frontal passage, computed as the maximum gradient of 850 hPa potential temperature, when this gradient is greater than 2 K/100 km, following Schemm et al. (2015). These results are displayed in the figure below:

[Figure]

E2

E3

E4

E5

E6

g

[Figure]

[Figure]

[Figure]

**Figure R4**: synoptic analysis using hourly ERA5 fields at the time of observed minimum $\delta^{18}O_v$ correspondint to the 11 events identified in the man,uscript (numbered E1 to E11): (a) air temperature at 850 hPa, (b) precipitation, and (c) vertical velocity at 850 hPa. White and black lines represent frontal passage, located at the maximum gradient of 850 hPa potential temperature. Front is computed as the zero-line of the gradient of the magnitude of the gradient of 850hPa air temperature, when the gradient of 850hPa air temperature is greater than 2 K/100 km, following Schemm et al. (2015).

*Reference :*

Schemm, Sebastian, Irina Rudeva, et Ian Simmonds. « Extratropical Fronts in the Lower Troposphere–Global Perspectives Obtained from Two Automated Methods ». Quarterly Journal of the Royal Meteorological Society 141, no 690 (2015): 1686‑98. https://doi.org/10.1002/qj.2471.

◦ The 11 events are chosen using the following criteria: " The green rectangles indicate the period with (1) correlation coefficient >-0.5 between d-excess and d18O of water vapor and (2) occurrence of a negative excursion in water vapor d18O." There are (from eye) other events that could fall into these criteria. For example, before the event in ~March 2021 (6th green rectangle in Fig. 3), there is an event agreeing with criteria (1) and showing a strong decrease in d18Ov. Why are other events not included? And how is a negative excursion in d18Ov defined?

>>> Many thanks for this comment and we agree that the definition of a negative excursion was not clear enough. The reason why the event in March 2021 has not been selected is that after the decrease of the $\delta^{18}O_v$, the $\delta^{18}O_v$ increases again but not to the level it had before the initial decrease. It stays several days on a low plateau.
We will thus precise in the revised manuscript that the $\delta^{18}O_v$ excursions are associated with $\delta^{18}O_v$ negative excursion larger than 2.5 permil (at 6h resolution) on a total length smaller than 24 h (definition of the length of the event is given in the caption of Table 1). We will also precise in the text that the average $\delta^{18}O_v$ 24h before and 24h after the event should not be larger than 1/4th of the amplitude of the $\delta^{18}O_v$ excursion. Note that some excursions were also discarded because of a too large interruption in the water isotopic record (21 March 2020).

◦ The analysis of the 11 events is mainly qualitative, and it is difficult to follow the description of the 4 events from 01-03/2020. All map plots and most of the vertical cross sections of these events are in the Supplementary Information, which makes it difficult to understand the synoptic situation during the events and the model performance. Further, it is not clear why these four events where chosen for a detailed description. Also, the description of the events is scattered in different paragraphs of Sections 3 and 4. The paragraphs should be better structure to lead the reader through the evolution of the cold front events.

The reason why we chose to show only the events from 01-03/2020 in the main text is simply to be able to read the figures easily and avoid too much repetitions. This period was favored because it encompasses several events which were then easily to see on a graph covering 3 months. As we wrote, « This period has been selected for display because it encompasses 4 out the 11 negative excursions of $\delta^{18}O_v$, but the extended comparison over the whole 2 years period is displayed on Figure S1. » and the same conclusions can be drawn if we consider the 11 events (all discussed in Table 1). To support this assumption, we provide online the figures corresponding to each event (such as current Figure 5 for the events of January 2020).

Following this comment, we will also reorganise the sections 3 and 4 to improve readibility and add more figures from the supplement in the main text. It is probably not possible to have the figures for all models in the main text but probably some figures from the ECHAMwiso model showing the best agreement with the data.

**Further general comments:**

• The section headings are not very specific. While Section 3 "Data description" has many short subsections, Section 4 "Discussion" has no subsection while introduducding a lot of information and new analyses.

>>> This is a very valid comment and we will include subsection titles in the discussion so that it will be easier to follow the argumentation.

• Water isotope measurements:
Various information is missing in the description of the water isotope measurements:
◦ What was the material and length of the inlet line to the Picarro instrument? Was the inlet line heated ?

>>> The inlet line was indeed heated (40°C) and the 5 m inlet tube was of PFA. This will be explained in the new manuscript.

◦ Was the water vapour mixing ratio measured by Picarro calibrated? How does it compare to other humidity measurements on the Island?

>>> The calibration of water vapour mixing ratio was done in the laboratory before sending the instrument and this protocol is valid as this calibration depends on the laser cavity configurations. On the field, we found an excellent agreement between mixing ratio measured by the Picarro and by the weather station (the difference between the two records always stays below 2% and there is no systematic shift between the two records). This will be added in the revised version.

◦ When was the humidity-isotope dependency calibration done and what kind of calibration device was used?

>>> The humidity-isotope dependency calibration was checked every year and the calibration device is the standard delivery module by Picarro. These explanations will be given in the new manuscript.

**Detailed comments:**
Lines 68-69: What is (18O/16O) and (D/H)? Does this represent the isotopic ratio?

>> Indeed, these are isotopic ratios between heavy and light isotopes. This will be added.

Lines 90-93: "For this objective, several instruments have been installed either in observatory stations ... or on boat ...".
Is there a reason that this summary omits aircraft measurements?

>> Many thanks for this comment, we will also mention aircraft measurements adding a reference to the following study as an example :

Henze, D., Noone, D., and Toohey, D.: Aircraft measurements of water vapor heavy isotope ratios in the marine boundary layer and lower troposphere during ORACLES, Earth Syst. Sci. Data, 14, 1811–1829, https://doi.org/10.5194/essd-14-1811-2022, 2022.

Lines 97-99: "Such data comparison enables one to test the performances of the models either in the simulation of the dynamic of the atmospheric water cycle or in the implementation of the water isotopes."
I agree with this statement but I don't see how this study adds any new knowledge on model performance or isotope parametrisations. Can you elaborate further?

>>> In the two models presented here, a very similar approach has been followed for water isotopes implementation. Because one model is able to reproduce well the observed water vapor $\delta^{18}O_v$ excursions, we can conclude that the isotopes parameterisation is appropriate (at least in this region). It means that the reason why the other model is not able to reproduce the excursion is not due to isotopic parameterisation but to the modeled atmospheric dynamic, most probably the horizontal resolution. As mentionned in the answer to a comment above, we will have a couple of sentences explaining that we can validate the implementation of water isotopes physics in the models by this model – data comparison.

Line 101-102: "This region is poorly documented with present-day observations despite its primary importance in governing CO2 sinks"
Do you have a reference for this statement?

>>> We propose to add the following reference.

Khatiwala, S., Primeau, F., & Hall, T. (2009). Reconstruction of the history of anthropogenic $CO_2$ concentrations in the ocean. *Nature*, 462(7271), 346–349. https://doi.org/10.1038/nature08526

Line 102-105: "Moreover, we lack precise descriptions of atmospheric processes associated with cloud microphysics and surface-atmosphere exchange in polar regions, and the evolution of westerly wind locations and strength (Fogt and Marshall, 2020)."
Why is this relevant for the presented study? The study site lies in the mid-latitudes.

>> We agree that this sentence is more general and not really adapted for this study in particular. Actually, it was more referring to the following paragraph where we explain that our initial aim to have an instrument at Amsterdam Island was to fill a gap between our observation in polar regions and in La Réunion. In the polar regions for example, we are concerned by the surface-atmosphere exchange. We will thus rewrite these paragraphs to better explain this general strategy.

Line 134: "Climate is temperate, generally mild with frequent presence of clouds."
What do you mean with frequent presence of clouds?

>> We will precise this statement and better say that the average total sunshine hours is 1581 hours per year from the period 1981 - 2010 day statistics of MeteoFrance (https://donneespubliques.meteofrance.fr/FichesClim/FICHECLIM_98404002.pdf)

Lines 149-150: "$CO_2$, CO, $CH_4$ and Hg species have been continuously measured since 1980, 2014, and 2012 respectively."There are four species but only three years are mentioned. It is not evident which species belongs to which year.

We propose to replace: "$CO_2$, CO, $CH_4$ and Hg species have been continuously measured since 1980, 2014, and 2012 respectively." by

« From 1980 to 2011, $CO_2$ and CO species have been monitored by non-dispersive infrared measurements (NDIR) systems. From 2012, dry air mole fractions of these two greenhouse species are currently measured by cavity ring down spectroscopy (CRDS) using commercial Picarro analyzers model G2401. Methane ($CH_4$) and nitrous oxide ($N_2O$) are also measured continuously using the same instrument model, but since 2014 and 2018 respectively, while Hg species are monitored since 2012 (see section 2.2.2.). »

Lines 160-167: The elevation of the two meteorological stations at Pointe Benedict observatory is given in meters agl, while the elevation of the station at Martin-de-Viviegs is given in meters asl. This makes it difficult to compare the elevations. Further, this paragraph gives a lot of detail on variables that are not shown later on.

The pointe Benedicte station is located 70 m above sea level. As a consequence, the meteorological stations referred here are 95-100 m above sea level or 25 m above ground level. We will try to simplify this paragraph.

Line 165: What is IGE?

>> IGE is the "Institut des Geosciences de l'Environnement", it will be explained in the revised version.

Line 187: What is STP?

>> It means "Standard temperature and pressure" and will be explained in the new manuscript. Also there was a mistake in the first manuscript since the temperature should be 273.15 K.

Line 202: "high altitude air masses (lower/ upper troposphere, or even above)" This is very unspecific. What do you mean with lower/upper troposphere?

>> There is not a precise altitude above which the GEM profile shows a decrease and is replaced by Hg oxidized species. The observations show that when we are above the free troposphere (in general above 5-6 km) and in the low stratosphere and when there is no biomass burning transportation from Africa, the GEM concentrations decrease with height (and this is the inverse for oxidized species). We thus propose to replace the following text:

"In this study, atmospheric GEM is used as potential tracer of intrusion and/or subsidence of high altitude air masses (lower/ upper troposphere, or even above) that may possibly impact the atmospheric records in Pointe Benedicte Observatory which collects marine boundary layer most of the time"

by:

"In this study, and even if long-range transport and a variable tropopause height may modulate, atmospheric GEM is used as potential tracer of stratosphere-to-troposphere intrusion and/or subsidence of upper troposphere (above 5-6 kms) that may possibly impact the atmospheric records in Pointe Benedicte Observatory which collects marine boundary layer most of the time"

Lines 204-205: " As mentioned above, mercury in the atmosphere is detected in three defined forms:"This has not been properly introduced earlier.

>>> We propose to replace the first sentence of section 2.2 by:

"Mercury (Hg) in the atmosphere consists of three forms: gaseous elemental Hg (GEM), gaseous oxidized Hg (GOM) and particulate-bound Hg (PBM) . Gaseous Elemental Mercury, dominant form (> 90%) of natural and anthropogenic Hg emissions transported globally through the atmosphere (Krabbenhof and Sunderland, 2013; Driscoll et al., 2013; Sprovieri et al., 2016), is the one the IPEV GMOStral-1028 observatory program is concentrating its analytical strengths, at the Pointe Benedicte atmospheric research facility. Data are freely...."

With the additional references:

Krabbenhof, D. P. & Sunderland, E. M. Global change and mercury. Science 341, 1457–1458 (2013).

Driscoll, C. T., Mason, R. P., Chan, H. M., Jacob, D. J. & Pirrone, N. Mercury as a global pollutant: sources, pathways, and efects. Environ. Sci. Technol. 47, 4967–4983 (2013).

Sprovieri, F. et al. Atmospheric mercury concentrations observed at ground-based monitoring sites globally distributed in the framework of the GMOS network. Atmos. Chem. Phys. 16, 11915–11935 (2016).

Line 233: "outside at ~ 6 m above ground level." What is this relative to m agl/asl, i.e. compared to the other measurements?

>> See previous answers to comment. It was indeed not very clear, 6 m above ground level (agl) is then 76 m above sea level. We will give this last number in the revised version.

Line 279: What do mean with "quadratic error"?

>> Because d-excess is defined as d-excess = $\delta D - 8 \times \delta^{18}O$ (to be added in the new version), the "quadratic error" on d-excess ($\sigma^2$) is calculated as the root square of ($\sigma_{\delta D}^2 + 64 \times \sigma_{\delta 18O}^2$). The term "quadratic error" is confusing and will be replaced by "uncertainty" as for the $\delta^{18}O$ and $\delta D$ uncertainties given above.

Line 290: Why are you starting the trajectories at 100m a.s.l.?

>> Because our instrument is at the surface and the Pointe Bénédicte observatory is 70 m above the ground, we chose the 100 m level. But you are right that we could also have taken a lower level. We have checked that we obtain exactly the same pattern when starting the back-trajectories at 100 m or 50 m high.

Line 324: "high spatial": 0.9° horizontal resolution is high compared to the LMDZiso simulation of this study but low compared to convection permitting climate simulations. I would therefore skip "high".

>> OK

Line 389: "d-excess of the precipitation". I don't see this in Fig. 3.

>> Indeed, we did not show this record because it does not add much to the study. This will be explained in the new version of the manuscript.

Line 395, 395, 399: What is $R_2$? The correlation coefficient R to the power of 2? What kind of correlation are you calculating? $R_2$ is used before R is introduced.

>> R2 is the coefficient of determination for a linear regression between the time series at hourly resolution. We will precise "with R2 being the coefficient of determination for a linear regression".

Line 399-400. "…(R is calculated continuously from hourly records in 8 consecutive days)…"
Do you mean that you used an 8-day moving window?

>> Yes, this is correct and the wording will be changed

Line 402-403: "d-excess$_v$" has not been introduced.

>> It will be introduced as"d-excess$_v$ = $\delta D_v$ - 8×$\delta^{18}O_v$"

Lines 434 – 436: "…the agreement with measured precipitation amount is better for ECHAM6-wiso (R2 = 0.45) than for LMDZ-iso (R2 = 0.08 – 0.13 for VLR – LR)…"
The correlation of LMDZ-iso with the measurements is close to zero, i.e. there seems to be nearly no agreement. The statement that the agreement with measured precipitation is better for ECHAM6-wiso than for LMDZ-iso seems weak in this context.

>>> We will change the wording and say that the correlation for LMDZiso is close to zero while there is a correlation for ECHAM6-wiso.

Lines 437-439: "…they are in general more strongly expressed in the data series than in the model series which is only partly due to the hourly resolution of the d18Ov record compared to the 3h and 6h resolution of the outputs of the LMDZ-iso…)." What is the basis of this conclusion?

>> Our idea is simply to see if part of the disagreement between data and model is linked to the fact that the models have only a 3h or 6h resolution while the data were displayed at 1h resolution. To test this, we have reinterpoled our data at 6h resolution in the figure 4 and calculated the amplitude of the water vapor $\delta^{18}O_v$ excursions with data resampled at 6h resolution. We will better explain that our aim was here to look if the difference resolution between the models and between models and data can explain the bad agreement between model outputs and data for the LMDZ-iso model.

Linbes 450-451: " They always occurred during low pressure periods (atmospheric pressure below 1005 mbar)." What is the synoptic situation leading to this low pressure and cold fronts?

>> As mentioned above, we will remove this reference to cold fronts because it was confusing and we actually wanted to focus on the water vapor $\delta^{18}O_v$ excursions. So we do not think that we need to include detailed descriptions of the synoptic situation.

Lines 501-503: "However, for the 11 events highlighted above, the d18Ov vs qv evolution follows an evolution characteristic of remoistening processes, i.e. a curve standing below the curve of the d18Ov vs qv evolution observed for the rest of the series…"
The single events show a much steeper evolution in the d18Ov-qv diagram than the remoistening curve. Why is this?

>> You are correct. As mentioned as answer to the editor, this figure is a "first order" approach following previous study of Guilpart et al. (2017) but is not 100% appropriate since the simple modeling curves are idealized $\delta^{18}O_v$ vs $q_v$ trajectories calculated for the free troposphere and we are looking at surface records which are strongly influenced also by vertical and horizontal atmosphere dynamic. We thus do not expect the curves to be aligned with our events and we simply use them at first approach to show that enhanced remoistening or water-rain interactions may be a good candidate to explain the $\delta^{18}O_v$ vs $q_v$ relationship during the events.

We propose to better explain as:
" Even if the water vapor $\delta^{18}O_v$ vs $q_v$ evolution is rather steep, there is some resemblance with the idealized theoretical curve for remoistening calculated for the free troposphere (Noone, 2012). Even if the analogy between our measurements at the surface and the free troposphere should be taken with cautious, the fact that the water vapor $\delta^{18}O_v$ vs $q_v$ evolution lies below the idealized curve for condensation processes supports the depleting effect of vapor-rain interactions for our negative water vapor $\delta^{18}O_v$ excursions (Worden et al., 2007; Noone, 2012). "

Lines 504-506: "Since relative humidity is relatively high during these events (values given in Table 1 compared to a mean value of 77 %), it more likely reflects rain-vapor diffusive exchanges than rain evaporation." Are you referring to the relative humidity at the surface? How about relative humidity above that will also influence the interaction of the rain with its surroundings?

>> This is correct. We should refer here to local surface re-evaporation only. We fully agree that rain - vapor interaction (including rain evaporation) in the upper atmosphere can have an influence as shown in Figure 8 and in l. 585 - 588. This will be explained better in the new version.

Line 519ff.: As the trajectories are only shown in the supplement, it is difficult to follow this paragraph. The beginning of the paragraph leads to think that the trajectories indicate that subsidence is import but in the end the conclusion is that "back trajectories are however not supporting systematic subsidence for other cases".

>> Indeed, the backtrajectories do not evidence systematic air subsidence for the water vapor $\delta^{18}O_v$ excursions and this is the reason why the figures were initially put in the supplement but we agree that it is better to have them in the main text. Actually, when looking in details the atmospheric dynamic, it is clear that the $\delta^{18}O_v$ excursions occur at the transition between ascendence and subsidence and this is probably the reason why we could not easily detect it from 10 days back-trajectories. This will be better explained in the text with back-trajectories of 5 days only (cf next answer to comment and figure R5).

Lines 526-527: "… the maximum altitude of the envelope of the back trajectories increases from 5,000 to 8,000 m…"

What it the mean/standard deviation of the maximum trajectory height? How many days before arrival are the trajectories at their maximum height? How relevant is this for the isotopic composition upon arrival? E.g. if the trajectories descend over the ocean and take up moisture, their maximum height before the moisture take-up is less relevant for the isotopic composition at Amsterdam Island. If you are using the full 10-days backward trajectories to calculate the maximum altitude, I don't think that the maximum altitude is a good measure of subsidence in front of the cold front.

>> Actually, what we wanted to test initially was if there is a change of the origin of the air mass during the excursion which may explain different isotopic signature of the water vapor. To answer the questions, we propose a new representation of the back-trajectories on only 5 days and with the indication of the location of the average (using humidity weighting) back-trajectory. We propose also to better explain that back-trajectories are used to mainly study the change of air origin.

As an example, we show below the new figure S2 that we propose (on the excursion of the 3-4 January 2020):

[Figure]

**Figure R5** : FLEXPART footprints in 2D projections for the event of the 3rd-4th of January. The colors on each grid point of these projections represent the density of particles over the 5-day back trajectories (1000 particules per launch). A dark red color indicates a zone with a high concentration of particles, hence a region from which a large part of the air mass originates. a: latitude-longitude projection of the FLEXPART back trajectory footprint for the 3rd of January 2020 at 13h30. b: same as

a for the 3rd of January 2020 at 22h30. c: left is the longitude-altitude projection of the FLEXPART back trajectory footprint for the 3rd of January 2020 at 13h30; right is the latitude-altitude projection of the FLEXPART back trajectory footprint for the 3rd of January 2020 at 13h30. d: same as (a) for the 3rd of January 2020 at 22h30.

Lines 543-555: " There is no evidence for changes in the horizontal advection of air over the 11 particular events from the observation of wind direction around these cold front events."

How is the cold front identified? Does it divide different air masses? A cold front normally implies a horizontal transport of air, why is this not the case for these cold fronts?

>> The cold front were initially identified with the synoptic weather charts as explained in answers to other comments. As mentioned elsewhere, the reference to these cold fronts will be muted in the new version.

Lines 556-558: "Such abrupt d18Ov events can hence be used as a test of the performances of general circulation models equipped with water isotopes."

What was learned about the performance of the GCMs involved in this study? Was it necessary to include d18O in such a performance test instead of just using traditional humidity variables (e.g. relative humidity, specific humidity, precipitation)?

>> You are right that many performances could already be tested using only meteorological data. This is the reason why we focused here only on the periods when water vapor $\delta^{18}O_v$ was showing a different signal than the one inferred from humidity to study what we can learn from this signal. Also, the combination with Hg measurements suggesting subsidence was useful to complement the traditional variable records over these excursions. Finally, because more and more models are equipped with water isotopes, we wanted to show an example of the peculiar signal seen in the water isotopes and not in traditional humidity variable to test the performances of these models, both on the implementation of water isotopes and on the dynamic of the atmospheric water cycle. This will be explained better in the revised version of the manuscript.

Line 562: What is "SOM"?

>> Supplementary Online Material, it will be explained in the revised version

Lines 559-584: As both isotope-enabled models were nudged to ERA5 dynamics, it is to be expected that the GCMs reproduce the ERA5 reanalysis wind fields rather well with some caveats due to the lower horizontal resolution. This paragraph (and Fig.7) is mostly describing the smoothing of ERA5 due to the coarser resolution of the isotope-enables GCMs. Why do we need isotope measurements to see the effect of a coarser horizontal resolution? What do we learn about the GCM performances by decreasing the horizontal resolution?

>> It is true that the GCMs are nudged to reanalyses. As mentioned in other answers to comments, the idea with water isotopes is to have a record at the surface of what is happening above in the atmosphere (without relying on reanalyses only). Also, we wanted to test, with different resolutions,

if the implementation of water isotopes was correct in the model or if the disagreement between models and data.

Lines 586-590: "A rain event is associated with a strong ascending column in which d18Ov is depleted by progressive precipitation during the ascent and by interaction between rain and water vapor. This ascending column is coupled to the subsidence of d18Ov depleted air at the rear of the event which is pushed toward Amsterdam Island through a south west advection of cold air."

How is the isotopic composition of the subsiding air behind the cold front connected to the progressive precipitation during the ascent? Can the ascending and descending column be differentiated in the d18O excursions?

>> This is a very valid question which is not so simple to answer with the model outputs since we do not have water tagging. Still, we did a simple analysis using the outputs of the ECHAM6-wiso model looking at the water vapor $\delta^{18}O_v$ vs $q_v$ in front of the event (ascending column), during the event and at the rear of the event (subsiding column) (Figure R6).

[Figure]

**Figure R6**: (left) Evolution of the water vapor $\delta^{18}O_v$ vs $q_v$ in ECHAM6wiso, at the surface at the nearest grid cell to Amsterdam island for all time steps between 01/01/2020 and 31/03/2020 (grey dots), at the surface for event 3 (24$^{th}$ of January 2020) at the latitude of Amsterdam and for all longitudes between 50°E and 100°E (black dots), and for three atmospheric columns at the time of the event (plain colored lines). The three vertical atmospheric columns are taken as best representations of the situation before the water vapor $\delta^{18}O_v$ anomaly of the 24th of January 2020, as an example. Ascending column is represented by the vertical atmospheric column at 81.6°E (upstream of / before the anomalous $\delta^{18}O_v$ event, green line); the situation during the event can be visualized as the vertical atmospheric column at 77.8°E, orange line; the vertical atmospheric column after the event (downstream) can be visualized by the column at 73.1°E, blue line. Vertical velocity directed downward is represented by a thick line (only present for the blue line). The black line indicates the distribution $\delta^{18}O_v$ vs $q_v$ for marine mixing and the dashed lines show Rayleigh distillation distributions. (right) as in Figure 7b of the article, but for event 3, and showing the location of the three atmospheric columns with same color lines as in the right.

The period before the $\delta^{18}O_v$ anomaly corresponds in most cases to the end of the rain event. This period is associated with a strong lift of the moist air in which we see a water vapor $\delta^{18}O_v$ vs qv distribution (low $\delta^{18}O_v$ with high humidity, green curve on Figure R6) which looks like an extreme case of remoistening (super Rayleigh as described in the Figure 6 of the main manuscript). We can interpret this as ongoing rain-steam exchanges. The situation is different from what happens at the surface where water vapor $\delta^{18}O_v$ vs $q_v$ is not showing any anomalous behavior yet.

During the event (orange curve on Figure R6), the water vapor $\delta^{18}O_v$ vs $q_v$ evolution is completely vertical and really difficult to explain by only remoistening effect. We thus believe that a dynamic aspect (mixing) is also involved in bringing in the surface boundary layer low water vapor $\delta^{18}O_v$ with relatively high humidity.

After the event (blue curve on Figure R6), we are back to a classic Rayleigh situation, so the water vapor $\delta^{18}O_v$ returns to its initial value.

Line 621: "hours/days". Is there a cold front passage that has a duration of several days?

>> You are right, we do not have cold fronts lasting several days over the Amsterdam Island. As noted in previous answers to comments, we did put too much emphasis on cold fronts which was misleading. We will remove "associated with cold fronts" on l. 623.

Lines 635-640: "This study highlighted the added value of combining different data from an atmospheric observatory to understand the dynamic of the atmospheric circulation. The two-year records are also a good benchmark for model evaluation. We have especially shown that the isotopic composition of water vapor is a powerful tool to identify aspects to be improved in the general circulation models, such as the horizontal resolution which may influence the representativity of the vertical dynamics."

As also mention above, why are stable water isotopes needed to show that the horizontal resolution may influence the vertical dynamics? The vertical cross sections of vertical wind speed (Fig. 7) illustrates this already quite well.

>> This comment has been addressed in the list of major comments above.

on the Zenodo platform, and the dataset does not include the water vapour mixing ratio.

>>> We will complete the dataset with a version 2 after acceptation of the manuscript including water vapour mixing ratio.

**Figures**:

General: The figures are often difficult to read, especially the described phenomena are small (e.g. d18O excursions of a few hours in a 2-year or 3 month timeline). Additionally, the colors are not color-blinded friendly and the caption are not concise.

>> We agree that it is difficult to see the excursions on a 2-year timeline and this is the reason why it has been decomposed in several figures of 3-months. In addition, some excursions are detailed in the main text. We propose to provide in the supplementary material or in the appendix a focus on each excursion like Figure 5. The colors have been changed on Figure 4 as shown below.

Fig.1: Is this figure needed? Fig.1 is not mentioned in the text.

>> We will mention it in the text at first instance. If the editor wants to remove it, it is also fine of course.

Fig.3: x-axis too coarse, light green shading difficult to see.

We will change to the following figure:

[Figure]

Fig. 4: Colors red/blue/green are not colorblind friendly. The caption text includes many repetition and should be improved. A legend in the figure could improve the readability.

>> We propose to have this modified figure (new color code) which will also enable to shorten the caption (since information is on the legend in the figure):

[Figure]

On this panel, we also removed the vertical velocity at 500 hPa from panel c to improve readability and because it did not bring much additional information (the information is given in Table 1).

Fig. 6: What is ɸ=0.025?

This value is indicated together with the "remoistening curve" which has been obtained following the expression in Noone (2012). Basically, the idea is to express remoistening through a modification of the equilibrium fractionation coefficient between water vapor and rain ($\alpha_e$) so that the effective fractionation factor will be $\alpha=(1+\phi)\times\alpha_e$

This effective fractionation coefficient is then introduced in the Rayleigh distillation equation to deduce the link between $\delta^{18}O_v$ and mixing ratio:

$(\delta-\delta_0)=(\alpha-1)*\ln(q/q_0)$

We will provide this information in the revised version of the manuscript.

Figure 8: What is SBL? The ascent of air in front of the cold front rises nearly vertical at a constant longitudinal position. As a cold front is moving system (mostly associated with an extratropical cyclone), the ascent does not occur at a constant location (in latitude and longitude). Further, all precipitation seems to fall in front of the cold front, which is unlikely.

SBL is Surface Boundary Layer, it will be changed by Marine Boundary Layer.

We agree that we need to include a more detailed analysis of the synoptic situation during the events. In Figure R4 shown above, we show front location, precipitation and 850 hPa vertical velocities from ERA5 at the time of the events. Precipitation generally falls just ahead of the cold front.

Figure 8 scheme is based on the profiles modelled by ECHAM for event 2 (09/01/2020). We show these profiles below (Figure R7). We will make this clear in the new legend to Figure 8 and add these profiles to the Supplement. We will show the same profiles for all events in the revised Supplement, but you can see in the current Supplement that these profiles share similar patterns for all events.

[Figure]

**Figure R7**: ECHAM6wiso profiles during Event 2, used to design Figure 8.

How is the subsidence at 100°E and ascent at 90°E related to the cold front?

The subsidence at 100°E seems to be linked to background conditions, while ascent at 90°E is caused by the cold front moving eastwards, causing precipitations just ahead of the front (Figures R4 and R7)

What does a composite of precipitation and d18Ov for all cold front events look like?

>> We already tried to make a composite before the first version of the manuscript but because the water $\delta^{18}O_v$ anomalies have different amplitude and durations (Table 1) and precipitation amount is very different from one event to the other (some event being associated with no rain, Table 1), a composite does not show any useful additional value.

Can it reproduce the schematic as shown in the "surface box"?

The surface box of the schematic follows the surface state modeled by ECHAM for event 2 (09/01/2020) (Figure R7, lower right plots).

We propose this new version for the Figure 8:

[Figure]

**Technical comments**

Generally: there are many abbreviations in the text that are only used a few times. Can you reduce the number of abbreviations?

>> This is true and will be modified in the next version.

Line 60-61 (and many others): The references are not in chronological order.

>> We will correct this.

Line 65-66: "We express the abundance of the heavy isotopes D and 18O with respect to the amount of light isotopes 16O and H in the water molecules…" should be "We express the abundance of the heavy isotopes D and 18O with respect to the amount of light isotopes **H and 16O, respectively,** in the water molecules…"

>> This will be modified

Line 68: Eq. 1 has strange symbols (squares).*

>> It is a problem from the word to pdf conversion, it should be possible to change it

Line 88-89: "water cycle processes such as water cycle processes such as "

>> This will be corrected

Lines 106-109: "Over the previous years, we have installed 3 water vapor analyzers on Reunion Island at the Maido observatory (21.079°S, 55.383°E, 2160m) (Guilpart et al., 2017) and in Antarctica (Dumont d'Urville and Concordia; (Leroy-Dos Santos et al., 2021; Breant et al., 2019; Casado et al., 2016). " Check usage of brackets.

>> This will be corrected following the editorial style

Line 133: "from the nearest lands, Madagascar**,** and"

>> It will be added

Lines 140-141: "…and were continuously monitored at the site **since** 1960…"

>> This will be corrected

Lines 145-150: The section is very difficult to read, the websites and datasets should better be included as references. Same for link to AERIS on line 178 and 200.

>> References will be given instead

Lines 180-181: "instrument models (Tekran Inc., Toronto, Canada) (Angot et al., 2014; Slemr et al., 2015, 2020; Sprovieri et al., 2016; Li et al., 2023). "

>>> This will be simplified in the new version.

Line 202: "may possibly" Doubling, omit either.

>> We will remove "possibly"

Lines 211-221: This sentence is too long. Can you divide into two sentences?

>>> Sure, we propose this new formulation:

"Chemical cycling and spatiotemporal distribution of mercury in the air is still poorly understood whatever atmospheric layer considered (surface, mixed or free troposphere, stratosphere), and complete GEM oxidation schemes remain still unclear (Shah et al., 2021 and associated references). Still, several studies provided evidence that vertical distribution of atmospheric mercury measurements from boundary layer to lower/upper troposphere and stratosphere shows a decreasing trend in GEM concentration with increasing altitude, in parallel with an increase in the concentration of divalent mercury (GOM + PBM) resulting from GEM oxidation mechanisms (Murphy et al., 2006 ; Swartzendruber et al., 2006, 2008 ; Talbot et al., 2007 ; Fain et al., 2009 ; Sheu et al., 2010 ; Lyman and Jaffe, 2012 ; Brooks et al., 2014 ; Fu et al., 2016 ; Koenig et al., 2023). "

Line 228: "The isotopic composition of near-surface water vapor (d18Ov and dDv in ‰ versus SMOW)"

>> It will be modified

Line 242: "The calibration of the data is performed in different**several** steps following previous studies"

>> It will be modified

Lines 301-303: "…identical **to** the atmospheric setup of IPSL-CM6A (Boucher et al., 2020) used for phase 6 of the Coupled Model Intercomparison Project (CMIP6, (Eyring et al., 2016)). "

>> It will be modified

Line 373: "… very close to the one observed in Angot et al**.** (2014)."

>> It will be modified

Line 374: "During the period (2020-2021) of water vapor isotope measurements in AMS…"

Do you mean: During the period 2020-2021 of water vapor isotope measurements in AMS.. ?

>> Yes, this will be changed

Lines 390-391: "The annual cycles are also not visible…" Do you mean: "An annual cycle is not visible…" ?

>> Yes, this will be changed

---

## Author Comment (AC3)

We thank very much the editor and the two reviewers for their very detailed and helpful comments. We have addressed all comments below and are willing to submit a manuscript taking into account all comments as explained in the answers to comment below.

Many thanks again for your help.

**Editor comments on egusphere-2023-617**

General comments:

The quality of this draft is above standard and the subject is suitable for publication in ACP. However, I think it may not be easy for every reader to understand this manuscript. Especially for those who are not experts on water isotope processes. I recommend major revisions of the manuscript so that the contents and results of this study become better understandable for any potential reader. Further, the graphical presentation is rather of low quality. Most of the figures should be improved (increasing the size of the figure itself as well as the font size and the line thickness). Some of the rather important figures of this study are in the supplement and may be moved to the main text or in an appendix to the main text (so that these do not appear in an extra document).

>> Many thanks for this suggestion. Indeed, the idea of an appendix to present the figures is a very good one and we will use it preferentially than the supplementary material for some figures.

The usage of terms etc. should be done more consistently one way or the other. In some occasions the term "vapor mixing ratio", "mixing ratio" , "water mixing ratio" or "water vapor mixing ratio" are used making reading this manuscript very confusing. Then the notation "v" as subscript is used in some occasions, but in many others not. I would suggest to use "v" and "p" to differentiate between vapor and precipitation water vapor and water vapor isotopes throughout the manuscript.

>> This is indeed again a good suggestion and we will do this systematically for the new version of the manuscript.

Specific comments

Abstract: The abstract is quite confusing and some transitional sentences, e.g. the first and second paragraph as well as a sentence stating that you use two models for comparison are missing. What is the main focus of your study ? The two paragraphs feel like two independent abstracts. One describing the measurements and the conclusions you derive from these and then the measurement-model comparison and the according results of this part of the study. The problem with the focus of study continues throughout the entire study. Additionally, several technical issues in the abstract need to be corrected (see below the list of technical corrections). Since there were so many issues I provide you here a corrected/improved version of your manuscript as a suggestion how it would read much better:

>> Many thanks for your suggestion. We slightly adapted it and propose the following abstract:

"In order to complement the picture of the atmospheric water cycle in the Southern Ocean, we have continuously monitored water vapor isotopes since January 2020 in Amsterdam Island (37.7983 °S, 77.5378 °E) in the Indian Ocean. We present here the first 2-year-long water vapor isotopic record on this site. We show that the vapor isotopic composition largely follows the water vapor mixing ratio, as expected in marine boundary layers. However, we evidence 11 periods of a few days where there is a strong loss of correlation between water vapor $\delta^{18}O$ and water vapor mixing ratio. These periods are associated with abrupt negative excursions of water vapor $\delta^{18}O$, often occurring toward the end

of precipitation events. Six of these events show a decrease in gaseous elemental mercury suggesting subsidence of air from higher altitude.

Our study aims at exploring the mechanism driving the variations of water vapor $\delta^{18}O$ and especially the 11 events identified in the water vapor $\delta^{18}O$. With the aim to understand the associated processes, the comparison of the data series with outputs of atmospheric components of Earth System models equipped with water isotopes is a very useful tool. We thus used two different models to provide a data-model comparison over this 2-year water vapor $\delta^{18}O$ record. While the European Centre Hamburg model (ECHAM6-wiso) was able to reproduce most of the sharp negative water vapor $\delta^{18}O$ excursions, the Laboratoire de Météorologie Dynamique Zoom model (LMDZ-iso) at 2° (3°) resolution was only able to reproduce 7 (1) of the negative excursions. Based on a detailed model-data comparison, we conclude that the most plausible explanations for such isotopic excursions are rain-vapor interactions associated with subsidence at the rear of a precipitation event."

P2, L51: Also in the introduction still the question remains what the purpose of your study is. Is it to confirm/better understand the measurements or to test the capability of the isotope enabled models to reproduce the isotopic processes?

>> In this manuscript, we mainly look for controls on negative isotopic excursions. This is done by using models to unravel processes and while suggesting a mechanism, we could also point on limitations of the performances of some of them, which may support future use of mesoscale isotope models to go further. We propose to add a sentence making the link between atmospheric components of Earth System Models and water isotopes and introducing in l. 61 the importance of the implementation of water isotopes in the atmospheric components of Earth System Models and to confront them with the models as:

"They can also be used as additional tools to test the performance of atmospheric components of some Earth System Models in which water isotopes have been added (Risi et al., 2010; Schmidt et al., 2005; Werner et al., 2011). «

P5, L113-114: The Dumont d'Urville and Concordia stations in Antarctica are not really in the Indian sector, but rather in the Pacific sector. Thus, how these are suitable for understanding the atmospheric water cycle over the Indian basin of the Southern Ocean does not become clear.

>> Many thanks for this question. We will make a more broader statement about South Indian Ocean being a significant moisture source for Antarctic precipitation, notably in the region encompassing Dumont d'Urville and Concordia stations. We will add references to this statement, including Wang et al. (2020) and Jullien et al. (2020) (see figures R1 and R2 below),

*References :*

Wang, H., Fyke, J. G., Lenaerts, J. T. M., Nusbaumer, J. M., Singh, H., Noone, D., Rasch, P. J., and Zhang, R.: Influence of sea-ice anomalies on Antarctic precipitation using source attribution in the Community Earth System Model, The Cryosphere, 14, 429–444, https://doi.org/10.5194/tc-14-429-2020, 2020.
Jullien, N., Vignon, É., Sprenger, M., Aemisegger, F., and Berne, A.: Synoptic conditions and atmospheric moisture pathways associated with virga and precipitation over coastal Adélie Land in Antarctica, The Cryosphere, 14, 1685–1702, https://doi.org/10.5194/tc-14-1685-2020, 2020.

[Figure]

Figure R1 : Spatial distribution of fractional contribution (%) to annual mean precipitation at the surface (right) from individual source regions in the mean case (left). Left panel is taken from Wang et al. 2020, Fig. 2 ; right panel is taken from Wang et al. 2020, Fig. 6

[Figure]

Figure R2 : Composite maps of moisture uptakes for precipitating air parcels over Dumont d'Urville from Jullien et al. (2020) Fig. 10.

We suggest to add the following information in the revised version:

"Amsterdam Island is the only oceanic observatories dedicated to atmospheric studies in the southern hemisphere. This regions in the Southern Indian Ocean is a significant moisture source for Antarctic precipitation, notably in the region encompassing Dumont d'Urville and Concordia stations (Wang et al., 2020; Jullien et al., 2020) "

Figure 1 caption: What is meant with Magand? Add a link or reference?

>> We explain better with "photo taken by O. Magand"

P8, L223: What do you mean here with "low altitude"? That the observatory is located at low altitude? Or that the air from higher altitude is transported down to lower altitudes?

>> Indeed, this sentence if not clear. We propose to modify as:

"The identification of such observational processes (lower GEM concentrations in high-altitude air masses versus marine boundary layer ones) is used here to help indentifying possible intrusions of high altitude air masses down to the surface in Pointe Benedicte Observatory."

P8, L228: Here now subscripts "v" used, but before not.

>> Actually, when we were writing either "water vapor $\delta^{18}O$" or simply "$\delta^{18}O_v$" all along the manuscript but we will simplify and put systematic $\delta^{18}O_v$, $q_v$ and $\delta D_v$ everywhere.

P9, Figure 2 caption: Why is here the anomaly used?

>> This is the standard way to correct the Picarro $\delta^{18}O_v$ and $\delta D_v$ measurements as detailed in the references given in the manuscript « (Tremoy et al., 2011; Leroy-Dos Santos et al., 2020)".

P9, L247: What exactly are these standards and how are these derived? Not clear! Are these typical relationships between these species? Are these documented somewhere else?

>> This is very classical when using isotopic composition of water vapor but we agree that it is difficult to follow when not working with this tool. We thus propose to explain better with the modified sentences:

"Here, we introduced two different water standards, EPB-AMS obtained from tap water at LSCE and GREEN-AMS obtained from melting Greenland surface snow. The water isotopic composition of these two standards lie on the global meteoric water line (Craig, 1961): they have respective values of (-5.66 ‰, -47.31 ‰) and (-32.65 ‰, -263.76 ‰) for the couple ($\delta^{18}O$, $\delta D$) which also encompass the range of isotopic values of $\delta^{18}O_v$ and $\delta D_v$ in Amsterdam Island. "

P9, L247: What exactly is denoted by these numbers? The data range?

>> These are the $\delta^{18}O$ and $\delta D$ values has explained in the sentence above (answer to previous comment)

P9, L254ff: I could not follow you. Why does the data need to be corrected? What did you find here in the relationship that is not as it should be?

>> From Figure 2 of the manuscript, you can see that the value of $\delta^{18}O_{measured}$ - $\delta^{18}O_{standard}$ is not flat as it should be when we inject the same standard at different humidity. This is a well known artefact of the water vapor analyzer which needs to be corrected as in all previous studies dealing with such kind of measurements.

We suggest the modified text:

"While we expect a constant null value for $\delta^{18}O_{measured}$- $\delta^{18}O_{standard}$ on Figure 2 because we always inject the same water standards at different humidity, the $\delta^{18}O$ measurements of both EPB-AMS and GREEN-AMS standards decrease with increasing humidity with the same amplitude. $\delta D_{measured}$-$\delta D_{standard}$ displayed on Figure 2 also shows variations but in contrast to the relative evolution of $\delta^{18}O$ with respect to water vapor mixing ratio, the $\delta D$ measurements of both EPB-AMS and GREEN-AMS standards exhibit different behavior: $\delta D$ of EPB-AMS increases by 1.5‰ and $\delta D$ of GREEN-AMS decreases by 2.5 ‰ over the same 6,000-24,000 ppmv range for water vapor mixing ratio $q_v$. "

P15, L402: How do you know that these peaks occurred during a cold front? No analyses ofmeteorological parameters indicating a front passage are shown or discussed here.

>> Indeed, we simply wanted to say that these $\delta^{18}O_v$ excursions occurred near the presence of a cold front as observed on weather synoptic maps for each excursion (see answer to reviewer 2, Figure R3). But as explained in the answer of reviewer 2, many cold fronts are also not associated with $\delta^{18}O_v$ excursions and we will refrain from referring to much to the cold fronts in the revised version. We agree that this was confusing and should be improved in the next version.

P14, L403-404: You only picked the 11 excursions with low correlation coefficient between d18Ov and qv. However, according to Fig. 3, there are other negative excursions of d18O besides these 11 cases. If the goal of this study is to show the isotopic features associated with the passage of the cold fronts, the authors should rather pick up the events from the weather chart showing a cold front passage, not from the low correlations alone.

>> As mentioned above, we concentrate on water vapor $\delta^{18}O_v$ excursions and not cold fronts. So we will still pick the events using low correlation coefficient between water vapor $\delta^{18}O_v$ and $q_v$ and the occurrence of $\delta^{18}O_v$ excursions. As explained in our answer to reviewer 1, we will thus precise in the revised manuscript that the $\delta^{18}O_v$ excursions are associated with water vapor $\delta^{18}O_v$ negative excursion larger than 2.5‰ (at 6h resolution) on a total length smaller than 24h (definition of the length of the event is given in the caption of Table 1). We will also precise in the text that the average water vapor $\delta^{18}O_v$ 24h before and 24h after the event should not be larger than 1/4th of the amplitude of the $\delta^{18}O_v$ excursion. Note that some excursions were also discarded because of a too large interruption in the water isotopic record (21st March 2020).

P16, Figure 4: Add a legend so that we can understand which color indicates which data just by looking at it.

>> This has been added, see answers to comments to reviewers 1 and 2.

P17, L454-455: Sentence not clear since it is grammatically incorrect. Please rephrase.

>> We propose this new sentence "Such mismatch makes the understanding of the processes at play during these events particularly important to test to further improve the performances of atmospheric general circulation models equipped with water isotopes."

P18, L475: I still have not seen how you can be sure that there was a cold front passage. How have the cold fronts been detected?

>> You are right. This sentence leads to confusion since we wanted to focus on excursions only. We will better add in table 1 if there is a presence of a cold front during the event (or a few hours before or after) but not state that cold fronts are associated with $\delta^{18}O_v$ excursions which was actually not our aim but it was indeed badly phrased. We will modify this sentence as:

"Several hypotheses can be proposed to explain the negative excursions of water vapor $\delta^{18}O_v$."

Line 504-506: Rain evaporation occurs under the cloud base, moistening the boundary layer. So, the authors should not underestimate the role of rain evaporation because of the high relative humidity near the surface.

>> This is true, we will precise this with this modified sentence making the distinction between what is occurring at the surface and the rain -vapor interaction (including rain evaporation) which can occur higher in the atmosphere:

"Since relative humidity at the surface stays relatively high during these events (values given in Table 1 compared to a mean value of 77 %), it more likely reflects rain-vapor exchanges in the atmosphere than rain evaporation under a transition to dry conditions at the surface."

P19, Figure 6: This figure is not clear at all and needs more explanation. The plot represents the observations in the boundary layer, but the theoretical curves are the isotopic changes in the free troposphere? At least Noone (2012) used them to investigate processes in the tropical mid troposphere. What exactly is meant with "inspired" by Noone (2012)? How have these curves been derived? Do you take these from the Noone (2012) paper? Have you calculated/estimated these yourself?

>> The aim of figure 6 is to look at simple relationships between water vapor $\delta^{18}O_v$ and $q_v$ using simple relationships. Calculation of the curves has been done using the formulas given in the Noone (2012) paper. These formulas are quite simple and could be used for different applications. Of course, it is better adapted to studies in the free troposphere but it has also been used in the past for studies of the isotopic composition of water vapor near the surface (e.g. Guilpart et al., JGR, 2017). We will explain in the revised manuscript that this simple representation is a first order approach for discussing the mechanisms. But because it is not enough, we then use general circulation models equipped with water isotopes in a second part of the discussion.

We hence propose to add the following explanation to introduce our analysis:

"First, to test the hypothesis of vapor-droplet interactions, we looked at the water vapor $\delta^{18}O_v$ and $q_v$ distribution following the approach already used by Guilpart et al. (2017) (Figure 6). We acknowledge that our approach is simple and should be taken as first order approach since we can only look at the water vapor $\delta^{18}O_v$ and $q_v$ distribution in the surface layer while it would be more adapted to look at this relationship in the free troposphere. "

>> We also changed the word "inspired" in the caption of Figure 6 and propose:

"The solid lines are theoretical lines whose equations are detailed in Noone (2012) "

*Reference* :

Guilpart, E., Vimeux, F., Evan, S., Brioude, J., Metzger, J. M., Barthe, C., ... & Cattani, O. (2017). The isotopic composition of near-surface water vapor at the Maïdo observatory (Reunion Island, southwestern Indian Ocean) documents the controls of the humidity of the subtropical troposphere. *Journal of Geophysical Research: Atmospheres*, *122*(18), 9628-9650.

P20, L523ff: Since subsidence is an important aspect of this study and you use Figure S1-S3 for the discussion, I don't understand why these figures are in the supplement instead of in the main part of the manuscript.

>> You are right and the modified figures according to the corresponding answer to reviewer 2 (figure R5) will be added in the main text.

P21, L623: I still haven't seen any proof that there has been a cold front passage.

>> As mentioned in other parts, this will be modified. Not all cold fronts are associated with $\delta^{18}O_v$ excursions (even if all water vapor $\delta^{18}O_v$ excursions occur in the vicinity of a cold front, Figure R4). We propose to remove the expression "associated with cold fronts" here.

P25, L640: It is still not clear what the function of the models are. Are these only used to be evaluated or are these also used to understand the processes behind the peaks in the d18O time series?

>> Indeed, in this conclusion, we explain that we used the models for the two applications. In the second paragraph (l. 629 - 634), we say that the model-data comparison permits to explain the mechanism at play to explain the $\delta^{18}O_v$ excursion. And in the last paragraph (l. 635 - 640), we say that the kind of water isotopic records provided can be useful to evaluate model equipped with water isotopes.

We propose a new version of the last paragraph to improve clarity:

"We have especially shown that the isotopic composition of water vapor measured at the surface can be a powerful tool to identify aspects to be improved in the atmospheric component of the general circulation models. In our case, we compared our data tomodel outputs with different horizontal resolutions. We showed that, as expected, resolution influences the representativity of the vertical dynamics and have important implication in the simulation of surface variations of water vapor $\delta^{18}O_v$ which can support the future use of mesoscale atmospheric models equipped with isotopes to better explain such abrupt isotopic excursions."

P26, L682: What do you mean with "of the s"?

>> to be changed to "realized most of the simulations"

Supplement: The supplement contains too many figures. I think not all of them are really necessary and the number could be reduced. Further, the formatting should be the same as for the ACP paper, that means no underlined headers and the same style for the figures (no underline of the figure caption title and no italic text for the caption text.)

>> We will indeed reduce the numbers of figures and put some of them in the appendix.

Technical corrections

>> All technical corrections will be taken into account. Many thanks again for your careful reading

---

## Author Response (AR1)

Dear Prof Farahnaz Khosrawi, dear reviewers

We thank the editor very much as well as the two reviewers for their very detailed and helpful comments. We have addressed all comments and are submitting a manuscript taking into account all comments as explained in the answers to comment below.

Many thanks again for your help and work on this manuscript,

Amaëlle Landais and Cécile Agosta on the behalf of all co-authors

**Editor comments on egusphere-2023-617**

General comments:

The quality of this draft is above standard and the subject is suitable for publication in ACP. However, I think it may not be easy for every reader to understand this manuscript. Especially for those who are not experts on water isotope processes. I recommend major revisions of the manuscript so that the contents and results of this study become better understandable for any potential reader. Further, the graphical presentation is rather of low quality. Most of the figures should be improved (increasing the size of the figure itself as well as the font size and the line thickness). Some of the rather important figures of this study are in the supplement and may be moved to the main text or in an appendix to the main text (so that these do not appear in an extra document).

>> Many thanks for this suggestion. Indeed, the idea of an appendix to present the figures is a very good one and we now use it preferentially than the supplementary material for some figures. We also include the figures for the back trajectories in the new manuscript and did an effort to better explain some aspects of the paper following your comments.

The usage of terms etc. should be done more consistently one way or the other. In some occasions the term "vapor mixing ratio", "mixing ratio" , "water mixing ratio" or "water vapor mixing ratio" are used making reading this manuscript very confusing. Then the notation "v" as subscript is used in some occasions, but in many others not. I would suggest to use "v" and "p" to differentiate between vapor and precipitation water vapor and water vapor isotopes throughout the manuscript.

>> This is indeed again a good suggestion and we will use the p and v subscripts and use the common term « water vapor mixing ratio » in the new version of the manuscript.

Specific comments

Abstract: The abstract is quite confusing and some transitional sentences, e.g. the first and second paragraph as well as a sentence stating that you use two models for comparison are missing. What is the main focus of your study ? The two paragraphs feel like two independent abstracts. One describing the measurements and the conclusions you derive from these and then the measurement-model comparison and the according results of this part of the study. The problem with the focus of study continues throughout the entire study. Additionally, several technical issues in the abstract need to be corrected (see below the list of technical corrections). Since there were so many issues I provide you here a corrected/improved version of your manuscript as a suggestion how it would read much better:

>> Many thanks for your suggestion. We propose the following abstract:

" In order to complement the picture of the atmospheric water cycle in the Southern Ocean, we have continuously monitored water vapor isotopes since January 2020 on Amsterdam Island in the Indian Ocean. We present here the first 2-year-long water vapor isotopic record

on this site. We show that the vapor water isotopic composition largely follows the water vapor mixing ratio, as expected in marine boundary layers. However, we detect 11 periods of a few days where there is a strong loss of correlation between water vapor $\delta^{18}O$ and water vapor mixing ratio. These periods are associated with abrupt negative excursions of water vapor $\delta^{18}O$, often occurring toward the end of precipitation events. Six of these events show also a decrease in gaseous elemental mercury suggesting subsidence of air from higher altitude.

Our study aims at further exploring the mechanism driving these negative excursions in water vapor $\delta^{18}O$. We used two different general atmospheric circulation models to provide a data-model comparison over this 2-year. While the European Centre Hamburg model (ECHAM6-wiso) at 0.9° is able to reproduce most of the sharp negative water vapor $\delta^{18}O$ excursions, hence validating the physics process and isotopic implementation in this model, the Laboratoire de Météorologie Dynamique Zoom model (LMDZ-iso) at 2° (3°) resolution only reproduces 7 (1) of the negative excursions highlighting the possible influence of model's resolution for the study of such abrupt isotopic events. Based on our detailed model-data comparison, we conclude that rain-vapor interactions associated with subsidence at the rear of a precipitation event are the most plausible explanations for such isotopic excursions. "

P2, L51: Also in the introduction still the question remains what the purpose of your study is. Is it to confirm/better understand the measurements or to test the capability of the isotope enabled models to reproduce the isotopic processes?

>> In this manuscript, we mainly look for controls on negative isotopic excursions. This is done by using models to unravel processes and while suggesting a mechanism, we could also point on limitations of the performances of some of them, which may support future use of mesoscale isotope models to go further. We propose to add a sentence making the link between atmospheric components of Earth system models and water isotopes and introducing in l. 61 the importance of the implementation of water isotopes in the atmospheric components of atmospheric general circulation models and to confront them with the models as:

"They can also be used as additional tools to test the performance of atmospheric components of some Earth system models in which water isotopes have been added (Risi et al., 2010; Schmidt et al., 2005; Werner et al., 2011). «

P5, L113-114: The Dumont d'Urville and Concordia stations in Antarctica are not really in the Indian sector, but rather in the Pacific sector. Thus, how these are suitable for understanding the atmospheric water cycle over the Indian basin of the Southern Ocean does not become clear.

>> We now make a more broader statement about South Indian Ocean being a significant moisture source for Antarctic precipitation, notably in the region encompassing Dumont d'Urville and Concordia stations. We add references to this statement, including Wang et al. (2020) and Jullien et al. (2020) (see figures R1 and R2 below),

*References :*

Wang, H., Fyke, J. G., Lenaerts, J. T. M., Nusbaumer, J. M., Singh, H., Noone, D., Rasch, P. J., and Zhang, R.: Influence of sea-ice anomalies on Antarctic precipitation using source attribution in the Community Earth System Model, The Cryosphere, 14, 429–444, https://doi.org/10.5194/tc-14-429-2020, 2020.

Jullien, N., Vignon, É., Sprenger, M., Aemisegger, F., and Berne, A.: Synoptic conditions and atmospheric moisture pathways associated with virga and precipitation over coastal Adélie Land in Antarctica, The Cryosphere, 14, 1685–1702, https://doi.org/10.5194/tc-14-1685-2020, 2020.

[Figure]

Figure R1 : Spatial distribution of fractional contribution (%) to annual mean precipitation at the surface (right) from individual source regions in the mean case (left). Left panel is taken from Wang et al. 2020, Fig. 2 ; right panel is taken from Wang et al. 2020, Fig. 6

[Figure]

Figure R2 : Composite maps of moisture uptakes for precipitating air parcels over Dumont d'Urville from Jullien et al. (2020) Fig. 10.

We add the following information in the revised version:

"Amsterdam Island is the only oceanic observatory dedicated to atmospheric studies in the southern hemisphere. This region in the Southern Indian Ocean is a significant moisture source for Antarctic precipitation, notably in the region encompassing Dumont d'Urville and Concordia stations (Wang et al., 2020; Jullien et al., 2020) ''

Figure 1 caption: What is meant with Magand? Add a link or reference?

>> We explain better with "photo taken by O. Magand"

P8, L223: What do you mean here with "low altitude"? That the observatory is located at low altitude? Or that the air from higher altitude is transported down to lower altitudes?

>> Indeed, this sentence if not clear. We propose to modify as:

"The identification of such observational processes (lower GEM concentrations in high-altitude air masses versus marine boundary layer ones) is used here to help indentifying possible intrusions of high-altitude air masses down to the surface in Pointe Benedicte Observatory."

P8, L228: Here now subscripts "v" used, but before not.

>> Actually, when we were writing either "water vapor $\delta^{18}O$" or simply " $\delta^{18}O_v$" all along the manuscript but we will simplify and put systematic $\delta^{18}O_v$, $q_v$ and $\delta D_v$ everywhere.

P9, Figure 2 caption: Why is here the anomaly used?

>> This is the standard way to correct the Picarro $\delta^{18}O_v$ and $\delta D_v$ measurements as detailed in the references given in the manuscript « (Tremoy et al., 2011; Leroy-Dos Santos et al., 2020)".

P9, L247: What exactly are these standards and how are these derived? Not clear! Are these typical relationships between these species? Are these documented somewhere else?

>> This is very classical when using isotopic composition of water vapor but we agree that it is difficult to follow when not working with this tool. We thus propose to explain better with the modified sentences:

"Here, we introduced two different water standards, EPB-AMS obtained from tap water at LSCE and GREEN-AMS obtained from melting Greenland surface snow. Theses standards are calibrated versus international water standards provided by IAEA. The water isotopic composition of these two standards lie on the global meteoric water line (Craig, 1961): they have respective values of (-5.66 ‰, -47.31 ‰) and (-32.65 ‰, -263.76 ‰) for the couple ($\delta^{18}O$, $\delta D$) which encompasses the range of isotopic values of $\delta^{18}O_v$ and $\delta D_v$ in Amsterdam Island. "

P9, L247: What exactly is denoted by these numbers? The data range?

>> These are the $\delta^{18}O$ and $\delta D$ values has explained in the sentence above (answer to previous comment)

P9, L254ff: I could not follow you. Why does the data need to be corrected? What did you find here in the relationship that is not as it should be?

>> From Figure 2 of the manuscript, you can see that the value of $\delta^{18}O_{measured}$ - $\delta^{18}O_{standard}$ is not flat as it should be when we inject the same standard at different humidity. This is a well known artefact of the water vapor analyzer which needs to be corrected as in all previous studies dealing with such kind of measurements.

We suggest the modified text:

"While we expect a constant null value for $\delta^{18}O_{measured}$- $\delta^{18}O_{standard}$ on Figure 2 because we always inject the same water standards at different humidity, the $\delta^{18}O$ measurements of both EPB-AMS and GREEN-AMS standards decrease with increasing humidity with the same amplitude. $\delta D_{measured}$-$\delta D_{standard}$ displayed on Figure 2 also shows variations but in contrast to the relative evolution of $\delta^{18}O$ with respect to water vapor mixing ratio, the $\delta D$ measurements of both EPB-AMS and GREEN-AMS standards exhibit different behavior: $\delta D$ of EPB-AMS increases by 1.5‰ and $\delta D$ of GREEN-AMS decreases by 2.5 ‰ over the same 6,000-24,000 ppmv range for water vapor mixing ratio $q_v$. "

P15, L402: How do you know that these peaks occurred during a cold front? No analyses ofmeteorological parameters indicating a front passage are shown or discussed here.

>> Indeed, we simply wanted to say that these $\delta^{18}O_v$ excursions occurred near the presence of a cold front as observed on weather synoptic maps for each excursion (see answer to reviewer 2, Figure R3). But as explained in the answer of reviewer 2, many cold fronts are also not associated with $\delta^{18}O_v$ excursions and we refrain now from referring to much to the cold fronts in the revised version. We agree that this was confusing and we explained in the answer to comments of reviewer 2 how we dealt with this issue.

P14, L403-404: You only picked the 11 excursions with low correlation coefficient between d18Ov and qv. However, according to Fig. 3, there are other negative excursions of d18O besides these 11 cases. If the goal of this study is to show the isotopic features associated with the passage of the cold fronts, the authors should rather pick up the events from the weather chart showing a cold front passage, not from the low correlations alone.

>> As mentioned above, we concentrate on water vapor $\delta^{18}O_v$ excursions and not cold fronts. So we will still pick the events using low correlation coefficient between water vapor $\delta^{18}O_v$ and $q_v$ and the occurrence of $\delta^{18}O_v$ excursions. As explained in our answer to reviewer 2, we give the followiong precisions in the revised manuscript :

« The 11 most abrupt events occurring when correlation coefficient R between $\delta^{18}O_v$ and d-excess$_v$ is larger than -0.5 are associated with $\delta^{18}O_v$ negative excursion larger than 3 ‰ (at 6h resolution) on a total length smaller than 24 h (as taken between the mid-slopes of the decrease and increase of the $\delta^{18}O_v$ respectively). The 11 selected negative excursions occur at a rate larger than -0.5‰.h$^{-1}$ and the $\delta^{18}O_v$ increase at the end of each excursion has an amplitude larger than half the amplitude of the corresponding initial decrease. »

Note that some excursions were also discarded because of a too large interruption in the water isotopic record (21$^{st}$ March 2020).

P16, Figure 4: Add a legend so that we can understand which color indicates which data just by looking at it.

>> This has been added, see answers to comments to reviewers 1 and 2.

P17, L454-455: Sentence not clear since it is grammatically incorrect. Please rephrase.

>> We propose this new sentence "Such mismatch makes the understanding of the processes at play during these events particularly important to test to further improve the performances of atmospheric general circulation models equipped with water isotopes."

P18, L475: I still have not seen how you can be sure that there was a cold front passage. How have the cold fronts been detected?

>> You are right. This sentence leads to confusion since we wanted to focus on excursions only. We modify this sentence as:

"Several hypotheses can be proposed to explain the negative excursions of $\delta^{18}O_v$."

Line 504-506: Rain evaporation occurs under the cloud base, moistening the boundary layer. So, the authors should not underestimate the role of rain evaporation because of the high relative humidity near the surface.

>> This is true, we actually can not rule out the effect of rain evaporation and we propose to be more cautious by stating :

" Surface relative humidity remains relatively high during these events (values given in Table 1 compared to a mean value of 77 %) which favors rain-vapor diffusive exchanges."

P19, Figure 6: This figure is not clear at all and needs more explanation. The plot represents the observations in the boundary layer, but the theoretical curves are the isotopic changes in the free troposphere? At least Noone (2012) used them to investigate processes in the tropical mid troposphere. What exactly is meant with "inspired" by Noone (2012)? How have these curves been derived? Do you take these from the Noone (2012) paper? Have you calculated/estimated these yourself?

>> The aim of figure 6 is to look at simple relationships between water vapor $\delta^{18}O_v$ and $q_v$ using simple relationships. Calculation of the curves has been done using the formulas given in the Noone (2012) paper. These formulas are quite simple and could be used for different applications. Of course, it is better adapted to studies in the free troposphere but it has also been used in the past for studies of the isotopic composition of water vapor near the surface (e.g. Guilpart et al., JGR, 2017). We explain in the revised manuscript that this simple representation is a first order approach for discussing the mechanisms. But because it is not enough, we then use atmospheric general circulation models equipped with water isotopes in a second part of the discussion.

We hence propose to add the following explanation to introduce our analysis:

" First, to test the hypothesis of vapor-droplet interactions, we looked at the $\delta^{18}O_v$ vs $q_v$ distribution following the approach already used by Guilpart et al. (2017) (Figure 6). We acknowledge that our approach is simple and should be taken as a first order approach since we can only look at the water vapor $\delta^{18}O_v$ vs $q_v$ distribution in the surface layer while it may be more adapted to look at this relationship in the free troposphere. "

>> We also changed the word "inspired" in the caption of Figure 6 and propose:

"The solid lines are theoretical lines whose equations are detailed in Noone (2012) "

*Reference* :

Guilpart, E., Vimeux, F., Evan, S., Brioude, J., Metzger, J. M., Barthe, C., ... & Cattani, O. (2017). The isotopic composition of near-surface water vapor at the Maïdo observatory (Reunion Island, southwestern Indian Ocean) documents the controls of the humidity of the subtropical troposphere. *Journal of Geophysical Research: Atmospheres*, *122*(18), 9628-9650.

P20, L523ff: Since subsidence is an important aspect of this study and you use Figure S1-S3 for the discussion, I don't understand why these figures are in the supplement instead of in the main part of the manuscript.

>> You are right and the modified figures according to the corresponding answer to reviewer 2 are now added in the main text (Figure 7) and appendices (figure A3 and A4).

P21, L623: I still haven't seen any proof that there has been a cold front passage.

>> As mentioned in other parts, this has been modified. Not all cold fronts are associated with $\delta^{18}O_v$ excursions (even if all water vapor $\delta^{18}O_v$ excursions occur in the vicinity of a cold front, Figure R4). We removed the expression "associated with cold fronts" here.

P25, L640: It is still not clear what the function of the models are. Are these only used to be evaluated or are these also used to understand the processes behind the peaks in the d18O time series?

>> Indeed, in this conclusion, we explain that we used the models for the two applications. In the second paragraph (l. 629 - 634), we say that the model-data comparison permits to explain the mechanism at play to explain the $\delta^{18}O_v$ excursion. And in the last paragraph (l. 635 - 640), we say that the kind of water isotopic records provided can be useful to evaluate model equipped with water isotopes.

We propose a new version of the last paragraph to improve clarity:

" We have shown that the isotopic composition of water vapor measured at the surface is a useful tool to identify aspects to be improved in the atmospheric component of the general circulation models. In our case, we used it to test different horizontal resolutions which influence the representativity of the vertical dynamics and have important implication in the simulation of surface variations of water vapor $\delta^{18}O_v$. Our study highlights the importance to have high-resolution models (e.g. mesoscale models) equipped with isotopes to further study such abrupt isotopic events.«

P26, L682: What do you mean with "of the s"?

>> Changed to "realized most of the simulations"

Supplement: The supplement contains too many figures. I think not all of them are really necessary and the number could be reduced. Further, the formatting should be the same as for the ACP paper, that means no underlined headers and the same style for the figures (no underline of the figure caption title and no italic text for the caption text.)

>> We will indeed reduce the numbers of figures and put some of them in the appendices.

Technical corrections

>> All technical corrections will be taken into account. Many thanks again for your careful reading

**Review 1**

I would like to compliment the authors for having prepared such a well-written paper and I strongly recommend the paper for publication after a very few adjustments/corrections. In general, the interpretation and the discussion of results is sounding and easy to follow. All figures are clear (see my only comment on Fig.4).

>> Many thanks for this general comment

I only have one comment/question about the interpretation of the results. Why d-excess has been (almost) left out of the discussion? The authors clearly state that during the depletion events both $\delta^{18}O_v$-q and $\delta^{18}O_v$-dexcess correlations break down. But how d-excess signal looks like during the event? If the d-excess doesn't change much, it would provide support to the hypothesis of rain-vapor interaction close to equilibrium than to rain-evaporation and to atmospheric subsidence, since evaporation of raindrops and free tropospheric air are associated with high d-excess.

>> d-excess of water vapor is indeed not changing much over the $\delta^{18}O_v$ excursion. We thus agree that it may not be explained by strong rain drop evaporation at the surface (even if rain drop evaporation is mostly affecting the rain) and it is also in agreement with the relatively high relative humidity at the surface. We thus agree that it may then support rain-vapor interaction close to equilibrium for the acquisition of this signal. Such rain-vapor interaction is indeed indicated on our summary on Figure 9 and we added in the manuscript that the stable d-excess signal supports this interpretation.

New sentence in the section 4.1 of the discussion:

« Such interpretation is also supported by the stable d-excess$_v$ during these events."

Minor comments:

L149 Please rearrange number of gasses (4) and monitoring years (3).

>> We have simplified this sentencev to keep only what is useful for our study as:

" Hg species have been continuously measured since 2012 »

L187 STP conditions: 273.15 K

>> Thank you, this will be changed.

Figure 4 Including a legend and reducing the size of the caption could improve readability.

>> Thank you, we propose the updated figure:

[Figure]

Figure 4: Data model comparison (January – March 2020); a- water vapor d$^{18}$O (light blue for data on hourly average, dark blue for data resampled at a 6-hour resolution); b- mixing ratio from our data set; c- vertical velocity; d- Precipitation amount. The grey shaded areas highlight the negative d$^{18}$O excursions (note that in this figure the excursions of the 3$^{rd}$ and 9$^{th}$ of January 2020 are distinct while the distinction could not be done on Figure 3 because of the scale).

**Review 2**

This manuscripts presents a unique time series of stable water isotopes in water vapour in the under-sampled area of the Southern Indian Ocean. The calibration of the water vapour isotope data follows established standards in the community. While the dataset presented here is of unique coverage, the relevance of the analysis and the conclusions is not convincing. Furthermore, the figures presenting the analysis are often difficult to read and many important information is in the supplement figures. In its current state, the manuscript lacks a motivation for the presented analysis and how it improves the understanding of the atmospheric water cycle. It might be worth considering a submission to Earth System Science Data (ESSD) instead of ACP.

>> The reason why we chose to submit to ACP is because we aim at combining for the first time water isotopes (data + 2 different models) and atmospheric species, here gaseous elemental mercury, over a long time series. The observation of concomitant water isotopes and Hg excursions suggests subsidence of air from high altitude.
We believe that water isotopes and elemental mercury records present an added value to the understanding of the dynamic of the atmospheric water cycle since they provide a surface diagnostic of processes happening higher in the atmosphere.
We agree with the reviewer that this objective was not clear enough in the manuscript, nor the added values of combining water isotopes, Hg and models and this is now hopefully better explain in the new version. Also, we worked on improving the readability (and colorscale) of the figures and moved some figures (backtrajectories for example) from the supplement to the main text and in appendices as suggested by the editor.

**Major comments:**
• **Model-measurement comparison**

The comparison of ECHAM6-wiso and LMDZ-iso with the measurements at Amsterdam Island leads to the main conclusion that "the isotopic composition of water vapor is a powerful tool to identify aspects to be improved in the general circulation models, such as the horizontal resolution which may influence the representativity of the vertical dynamics." Why are *isotope-enabled* models needed to illustrate that horizontal resolution influences vertical dynamics in GCMs? This is, for example, already evident when comparing the vertical wind fields in Fig.7. Which "aspects to be improved", other than the horizontal resolution, were identified in this study?

Our study aims at exploring the mechanism driving the variations of water vapor $\delta^{18}O_v$ and especially the 11 events identified in the water vapor $\delta^{18}O_v$. With the aim to understand the associated processes, the comparison of the data series with outputs of atmospheric components of AGCMs equipped with water isotopes is a useful tool to evaluate potentiel biais in models. In addition to propose a mechanism for the water vapor $\delta^{18}O_v$ excursions through our model – data comparison, we could also propose some ideas to explain why some models could not reproduce the water isotopic excursions at the surface. Isotope implementation is quite similar in the 2 models and is probably not the reason for the differences, a result that we now highlight in the revised version of the manuscript. Nudging is slightly different between the two models (vorticity nudging in ECHAM6-wiso, shorter relaxation time in LMDZ-iso) but as the reviewer rightly points out, the conclusion is that if isotopes are controlled during these events by vertical dynamics at the scale of a few tens of kilometers, then it is probably necessary to have high-resolution models to unravel the mechanisms. As detailed in the comments below, we added in the discussion and conclusion a couple of sentences saying that our model-data comparison validates the physics processes within the ECHAM6-wiso model as well as the implemented physics of water isotopes. Moreover, we stress in the conclusion the importance to have higher resolution models (mesoscale models) equipped with isotopes if one wants to study such events.

Finally, vertical wind fields from initial figure 7 are from reanalyses, themselves based on model. They were not measured. So we believe that our study, unique in this region, still has an added values by comparing a new measured data series (the records of water vapor $\delta^{18}O_v$ at the surface) with model outputs.

• **Cold front analysis**
The analysis is based on d18Ov excursions which are described as cold front events. The selection and analysis of these excursion with respect to cold frontal dynamics on a synoptic scale is missing. It is therefore difficult to interpret the described events with respect to the large-scale dynamics. In detail:
∘ The analysis focuses on 11 events of cold fronts. These fronts and their spatial structure is not described in the manuscript and there are no synoptic figures describing a typical situation during a cold front passage, except for a few weather analysis charts in the supplementary information. How are the cold fronts identified? What are the properties of the cold fronts? How much of the annual precipitation is represented by cold frontal precipitation?

>> This comment is very sound. Our aim was actually not to focus on cold fronts but on the isotopic record and especially on periods when the isotopic record is providing an added value to understand features associated with the atmospheric water cycle, that is to say a different signal than the meteorological signals (humidity, temperature). This is why we concentrate on the abrupt negative water vapor $\delta^{18}O_v$ excursions which are not seen in the humidity signal. The periods during which we observed water vapor $\delta^{18}O_v$ negative excursions are associated with a cold front but there are probably many cold fronts that are not exhibiting any water vapor $\delta^{18}O_v$ excursions. Because this is not the subject of this analysis, we better avoid now to refer to cold fronts when unnecessary as it was indeed misleading.

In the previous manuscript, we had identified the cold fronts from the synoptic figures. All weather charts corresponding to the 11 water vapor $\delta^{18}O_v$ negative excursions are shown below:

| | Day-1 of event at 00:00 UTC | Day of event at 00:00 UTC | Day+1 of event at 00:00 UTC |
|---|---|---|---|
| 03/01/2020 | | | |
| 09/01/2020 | | | |
| 24/01/2020 | | | |
| 05/03/2020 | | | |
| 10/05/2020 | | | |

[Figure]

[Figure]

**Figure R3** : Weather analysis charts provided once a day at 00:00 UTC by the Analysis Chart Archive service of the Australian Government Bureau of Meteorology http://www.bom.gov.au/australia/charts/archive/index.shtml. Red dot on weather charts displays Amsterdam Island location.

The idea was to check for the presence of a cold front in a distance of 100 km around Amsterdam Island in a 48h period covering the time of the event. We indeed see that we systematically have cold

fronts in the vicinity of the Amsterdam Island at the time of the water vapor $\delta^{18}O_v$ excursions. Still, in some cases, such as the 06/12/2021, there are cold fronts identified on the weather charts in the vicinity of the Amsterdam Island 1 day before and 1 day after the excursions but no clear cold front over Amsterdam Island at the exact time of the water vapor $\delta^{18}O_v$ excursion.

We explain now very simply the link to cold fronts as :

«These negative $\delta^{18}O_v$ excursions always occurred during low pressure periods (atmospheric pressure below 1005 mbar) and we observe the presence of a cold front in a distance of 100 km around Amsterdam Island in a 48h period covering the time of the event (Supplementary Material). »

To address the occurrence of cold fronts in a more detailed manner, we propose here a second synoptic analysis with the frontal passage, computed as the maximum gradient of 850 hPa potential temperature, when this gradient is greater than 2 K/100 km, following Schemm et al. (2015). These results are displayed in the figure below which is now part of the supplementary material (2 first columns only):

[Figure]

[Figure]

[Figure]

**Figure R4**: synoptic analysis using hourly ERA5 fields at the time of observed minimum $\delta^{18}O_v$ correspondint to the 11 events identified in the manuscript: (a) air temperature at 850 hPa, (b) precipitation, and (c) vertical velocity at 850 hPa. White and black lines represent frontal passage, located at the maximum gradient of 850 hPa potential temperature. Front is computed as the zero-line of the gradient of the magnitude of the gradient of 850hPa air temperature, when the gradient of 850hPa air temperature is greater than 2 K/100 km, following Schemm et al. (2015).

*Reference :*

Schemm, Sebastian, Irina Rudeva, et Ian Simmonds. « Extratropical Fronts in the Lower Troposphere–Global Perspectives Obtained from Two Automated Methods ». Quarterly Journal of the Royal Meteorological Society 141, no 690 (2015): 1686-98. https://doi.org/10.1002/qj.2471.

◦ The 11 events are chosen using the following criteria: " The green rectangles indicate the period with (1) correlation coefficient >-0.5 between d-excess and d18O of water vapor and (2) occurrence of a negative excursion in water vapor d18O." There are (from eye) other events that could fall into these criteria. For example, before the event in ~March 2021 (6th green rectangle in Fig. 3), there is an event agreeing with criteria (1) and showing a strong decrease in d18Ov. Why are other events not included? And how is a negative excursion in d18Ov defined?

>>> Many thanks for this comment and we agree that the definition of a negative excursion was not clear enough and we agree that it may be a bit subjective since we took only the 11 most prominent events. The reason why the event in March 2021 has not been selected is its slower d18Ov decrease at the start of the excursion (less than 0.5 permil.h-1) and the fact that, at the end of the excursion, the $\delta^{18}O_v$ does increase with an amplitude which is less than half the amplitude of the initial decrease (2.4 permil to be compared to 6 permil). This criterium is now added in the new version of the manuscript.

[Figure]

Figure R5 : Focus on the event of the 19-20 February 2021, not selected for our 11 most prominent events because one criterium in the evolution of the $\delta^{18}O_v$ was not fulfilled. Still, this event shares many of the main characterictics of the selected events.

We now precise in the revised manuscript that :

« The 11 most abrupt events occurring when correlation coefficient R between $\delta^{18}O_v$ and d-excess$_v$ is larger than -0.5 are associated with $\delta^{18}O_v$ negative excursion larger than 3 ‰ (at 6h resolution) on a total length smaller than 24 h (as taken between the mid-slopes of the decrease and increase of the $\delta^{18}O_v$ respectively). The 11 selected negative excursions occur at a rate larger than -0.5‰.h$^{-1}$ and the $\delta^{18}O_v$ increase at the end of each excursion has an amplitude larger than half the amplitude of the corresponding initial decrease. »

Note that some excursions were also discarded because of a too large interruption in the water isotopic record (21 March 2020).

◦ The analysis of the 11 events is mainly qualitative, and it is difficult to follow the description of the 4 events from 01-03/2020. All map plots and most of the vertical cross sections of these events are in the Supplementary Information, which makes it difficult to understand the synoptic situation during the events and the model performance. Further, it is not clear why these four events where chosen for a detailed description. Also, the description of the events is scattered in different paragraphs of Sections 3 and 4. The paragraphs should be better structure to lead the reader through the evolution of the cold front events.

The reason why we chose to show only the events from 01-03/2020 in the main text is simply to be able to read the figures easily and avoid too many repetitions. This period was favored because it encompasses several events which were then easily to see on a graph covering 3 months. As we wrote, « This period has been selected for display because it encompasses 4 out

the 11 negative excursions of $\delta^{18}O_v$, but the extended comparison over the whole 2 years period is displayed on Figure S1. » and the same conclusions can be drawn if we consider the 11 events (all discussed in Table 1). To support this assumption, we now provide in the appendices the figures corresponding to each event (such as current Figure 5 for the events of January 2020).

Following this comment, we have reorganised the section 4 with addition of subtitles and reduction of some paragraphs (sections 3 and 4) to improve readibility. We also added more figures from the supplement in the main text (backtrajectories) or appendices (Figure S1 has been moved in appendices, figure A1, and we have added a focus on each event as Figure A2).

**Further general comments:**

• The section headings are not very specific. While Section 3 "Data description" has many short subsections, Section 4 "Discussion" has no subsection while introducding a lot of information and new analyses.

>>> We have included subsection titles in the discussion (4.1 to 4.5) so that it will be easier to follow the argumentation. Some paragraphs have also been rewritten for clarity as explained in answer to specific comments listed below.

• Water isotope measurements:
Various information is missing in the description of the water isotope measurements:
◦ What was the material and length of the inlet line to the Picarro instrument? Was the inlet line heated ?

>>> The inlet line was indeed heated (40°C) and the 5 m inlet tube was of PFA. This is explained in the new manuscript.

**«** The instrument has been installed in a temperature-controlled room at the observatory on the Amsterdam Island and the sampling of water vapor is done outside at ~ 6 m above ground level (or 76 m above sea level) through a 5 m long inlet tube made of PFA (perfluoroalkoxy alkanes) and heated at 40°C."

◦ Was the water vapour mixing ratio measured by Picarro calibrated? How does it compare to other humidity measurements on the Island?

>>> The calibration of water vapour mixing ratio was done in the laboratory before sending the instrument and this protocol is valid as this calibration depends only on the laser cavity configurations. On the field, we found an excellent agreement between mixing ratio measured by the Picarro and by the weather station (the difference between the two records always stays below 2% and there is no systematic shift between the two records). This is added in the revised version :

« The calibration of water vapour mixing ratio was performed in the laboratory before sending the instrument. On the field, we found an excellent agreement between mixing ratio measured by the Picarro instrument and mixing ratio measured by the weather station (the difference between the two records always stays below 2% and there is no systematic shift between the two records). "

◦ When was the humidity-isotope dependency calibration done and what kind of calibration device was used?

>>> The humidity-isotope dependency calibration was checked every year and the calibration device is the standard delivery module by Picarro. These explanations are given in the new manuscript.

« The calibration of the water isotopic data is performed in several steps following previous studies (Leroy-Dos Santos et al., 2020; Tremoy et al., 2011) and using a standard delivery module by Picarro. »

« This calibration step has been performed every year over the whole range of mixing ratio values and provided very similar results from one year to the other.”

**Detailed comments:**
Lines 68-69: What is (18O/16O) and (D/H)? Does this represent the isotopic ratio?

>> Indeed, these are isotopic ratios between heavy and light isotopes. This is now added.

Lines 90-93: "For this objective, several instruments have been installed either in observatory stations ... or on boat …".
Is there a reason that this summary omits aircraft measurements?

>> Many thanks for this comment, we now also mention aircraft measurements adding a reference to the following study as an example :

Henze, D., Noone, D., and Toohey, D.: Aircraft measurements of water vapor heavy isotope ratios in the marine boundary layer and lower troposphere during ORACLES, Earth Syst. Sci. Data, 14, 1811–1829, https://doi.org/10.5194/essd-14-1811-2022, 2022.

Lines 97-99: "Such data comparison enables one to test the performances of the models either in the simulation of the dynamic of the atmospheric water cycle or in the implementation of the water isotopes."
I agree with this statement but I don't see how this study adds any new knowledge on model performance or isotope parametrisations. Can you elaborate further?

>>> In the two models presented here, a very similar approach has been followed for water isotopes implementation. Because one model is able to reproduce well the observed water vapor $\delta^{18}O_v$ excursions, we can conclude that the isotopes parameterisation is appropriate (at least in this region). It means that the reason why the other model is not able to reproduce the excursion is not due to isotopic parameterisation but to the modeled atmospheric dynamics whose horizontal resolution is likely too coarse. We have rewritten some part of the manuscript to adress this comment in the discussion section:

Discussion :

« Still, the fact that at least ECHAM6-wiso is able to reproduce every negative $\delta^{18}O_v$ excursions (whether they are associated or not with subsidence or rain-water vapor reequilibration) shows that not only the patterns of atmospheric water cycle are correctly reproduced (a validation

which can also be performed using humidity and precipitation data) but also that the isotopic processes are correctly implemented in this model. Such abrupt $\delta^{18}O_v$ events can hence be used as a test bed of the skills of water isotopes enabled general circulation models. "

Conclusion :

"The good agreement between modeled and measured $\delta^{18}O_v$ when using ECHAM6-wiso validates the physics processes within the ECHAM6-wiso model as well as the implemented physics of water isotopes.

[….]

This study highlighted the added value of combining different data from an atmospheric observatory to understand the dynamics of the atmospheric circulation. The 2-year records are also a good benchmark for model evaluation. We have especially shown that the isotopic composition of water vapor measured at the surface is a powerful tool to identify aspects to be improved in the atmospheric component of the general circulation models. In our case, we used it to test different horizontal resolutions which influence the simulation of the vertical dynamics and have important implication in the simulation of surface variations of water vapor $\delta^{18}O_v$. Our study highlights the importance to have high-resolution models (e.g. mesoscale models) equipped with isotopes to further study such abrupt isotopic events."

Line 101-102: "This region is poorly documented with present-day observations despite its primary importance in governing CO2 sinks"
Do you have a reference for this statement?

>>> This sentence has been removed for simplicity, cf next comment.

Line 102-105: "Moreover, we lack precise descriptions of atmospheric processes associated with cloud microphysics and surface-atmosphere exchange in polar regions, and the evolution of westerly wind locations and strength (Fogt and Marshall, 2020)."
Why is this relevant for the presented study? The study site lies in the mid-latitudes.

>> We agree that this sentence is more general and not really adapted for this study in particular (it mostly refered to our other studies in Antarctica). For simplification, we propose to remove « This region is poorly documented with present-day observations despite its primary importance in governing $CO_2$ sinks. Moreover, we lack precise descriptions of atmospheric processes associated with cloud microphysics and surface-atmosphere exchange in polar regions, and the evolution of westerly wind locations and strength (Fogt and Marshall, 2020)."

Line 134: "Climate is temperate, generally mild with frequent presence of clouds."
What do you mean with frequent presence of clouds?

>> We precise by stating that « the average total sunshine hours is 1581 hours per year from the period 1981 – 2010 » (https://donneespubliques.meteofrance.fr/FichesClim/FICHECLIM_98404002.pdf)

Lines 149-150: "$CO_2$, CO, $CH_4$ and Hg species have been continuously measured since 1980, 2014, and 2012 respectively."There are four species but only three years are mentioned. It is not evident which species belongs to which year.

Because of the need to simplify this section (cf other comments) and keep only what is necessary for our study, the paragraph has been simplified as :

« Numerous atmospheric compounds and meteorological parameters are and were continuously monitored at the site since 1960 (Angot et al., 2014; El Yazidi et al., 2018; Gaudry et al., 1983; Gros et al., 1999, 1998; Polian et al., 1986; Sciare et al., 2000, 2009; Slemr et al., 2015; Slemr et al., 2020). In particular, the Amsterdam (AMS) site hosts several dedicated atmospheric observation instruments notably at the Pointe Bénédicte atmospheric observatory (70 m above sea level) where greenhouse gases concentrations and mercury (Hg) are monitored. . Hg species have been continuously measured since 2012. "

Wind speed and direction, atmospheric pressure, surface temperature and relative humidity data are currently obtained at a minute resolution. Another meteorological station is based on the island and is operated by Météo France at Martin-de-Viviès life base around 27 m above sea level, about two kilometers east from the Pointe Bénédicte observatory collecting air temperature, humidity, precipitation, wind speed and direction, pressure and solar radiation "

Line 165: What is IGE?

>> IGE is the "Institut des Geosciences de l'Environnement", it has been removed in the new version (for simplification as requested in previous comment)

Line 187: What is STP?

>> It means "Standard temperature and pressure" and is written as such in the new manuscript. Also there was a mistake in the first manuscript since the temperature should be 273.15 K.

Line 202: "high altitude air masses (lower/ upper troposphere, or even above)" This is very unspecific. What do you mean with lower/upper troposphere?

>> There is not a precise altitude above which the GEM profile shows a decrease and is replaced by Hg oxidized species. The observations show that when we are above the free troposphere (in general above 5-6 km) and in the low stratosphere and when there is no biomass burning transportation from Africa, the GEM concentrations decrease with height (and this is the inverse for oxidized species). We thus propose to replace the following text:

"In this study, atmospheric GEM is used as potential tracer of intrusion and/or subsidence of high altitude air masses (lower/ upper troposphere, or even above) that may possibly impact the atmospheric records in Pointe Benedicte Observatory which collects marine boundary layer most of the time"

by:

"In this study, and even if long-range transport and a variable tropopause height may modulate, atmospheric GEM is used as potential tracer of stratosphere-to-troposphere intrusion and/or subsidence of upper troposphere (above 5-6 kms) that may impact the atmospheric records in Pointe Benedicte Observatory which collects marine boundary layer most of the time"

Lines 204-205: " As mentioned above, mercury in the atmosphere is detected in three defined forms:"This has not been properly introduced earlier.

>>> We propose to replace the first sentence of section 2.2 by:

"Mercury (Hg) in the atmosphere consists of three forms: gaseous elemental Hg (GEM as defined above), gaseous oxidized Hg and particulate-bound Hg. »

Line 233: "outside at ~ 6 m above ground level." What is this relative to m agl/asl, i.e. compared to the other measurements?

>> See previous answers to comment. We now precise :
« . The instrument has been installed in a temperature-controlled room at the observatory on the Amsterdam Island and the sampling of water vapor is done outside at ~ 6 m above ground level (or 76 m above sea level)."

Line 279: What do mean with "quadratic error"?

>> Because d-excess is defined as d-excess = $\delta D-8 \times \delta^{18}O$, the "quadratic error" on d-excess ($\sigma^2$) is calculated as the root square of ($\sigma_{\delta D}^2 + 64 \times \sigma_{\delta 18O}^2$). The term "quadratic error" is confusing and is replaced by "uncertainty" as for the $\delta^{18}O$ and $\delta D$ uncertainties given above.

Line 290: Why are you starting the trajectories at 100m a.s.l.?

>> Because our instrument is at the surface and the Pointe Bénédicte observatory is 70 m above the ground, we chose the 100 m level. But you are right that we could also have taken a lower level. We have checked that we obtain exactly the same pattern when starting the back-trajectories at 100 m or 50 m high.

Line 324: "high spatial": 0.9° horizontal resolution is high compared to the LMDZiso simulation of this study but low compared to convection permitting climate simulations. I would therefore skip "high".

>> OK

Line 389: "d-excess of the precipitation". I don't see this in Fig. 3.

>> Indeed, we did not show this record because it does not add much to the study. We precise in the new version « No significant seasonal variations are observed in the record of d-excess of precipitation (not shown). »

Line 395, 395, 399: What is $R_2$? The correlation coefficient R to the power of 2? What kind of correlation are you calculating? $R_2$ is used before R is introduced.

>> R2 is the coefficient of determination for a linear regression between the time series at hourly resolution. We now precise "with R2 being the coefficient of determination for a linear regression".

Line 399-400. "...(R is calculated continuously from hourly records in 8 consecutive days)..."
Do you mean that you used an 8-day moving window?

>> Yes, this is correct and the sentence has been changed

Line 402-403: "d-excess$_v$" has not been introduced.

>> It is now introduced as"d-excess$_v$ = $\delta D_v$ - $8 \times \delta^{18} O_v$"

Lines 434 – 436: "...the agreement with measured precipitation amount is better for ECHAM6-wiso (R2 = 0.45) than for LMDZ-iso (R2 = 0.08 – 0.13 for VLR – LR)…"
The correlation of LMDZ-iso with the measurements is close to zero, i.e. there seems to be nearly no agreement. The statement that the agreement with measured precipitation is better for ECHAM6-wiso than for LMDZ-iso seems weak in this context.

>>> We changed the wording and wrote that the correlation for LMDZiso is close to zero while there is a correlation for ECHAM6-wiso.

« The correlation between modeled and measured precipitation is close to zero for LMDZ-iso ($R^2$ = 0.08 – 0.13 for VLR - LR) while there is a better agreement when comparing measured precipitation amount to outputs of ECHAM6-wiso ($R^2$ = 0.45). »

Lines 437-439: "...they are in general more strongly expressed in the data series than in the model series which is only partly due to the hourly resolution of the d18Ov record compared to the 3h and 6h resolution of the outputs of the LMDZ-iso…)." What is the basis of this conclusion?

>> Our idea is simply to see if part of the disagreement between data and model is linked to the fact that the models have only a 3h or 6h resolution while the data were displayed at 1h resolution. To test this, we have reinterpoled our data at 6h resolution in the figure 4 and calculated the amplitude of the water vapor $\delta^{18} O_v$ excursions with data resampled at 6h resolution. We will better explain that our aim was here to look if the difference resolution between the models and between models and data can explain the bad agreement between model outputs and data for the LMDZ-iso model.

We propose this new formulation :

« Part of this disagreement can be explained by the fact that the $\delta^{18} O_v$ record has a higher temporal resolution (1h) than the model outputs (3h for LMDZ-iso and 6h for ECHAM6-wiso) . However, when interpolating the $\delta^{18} O_v$ record at a 6h resolution, the negative excursions are still clearly visible which cannot be captured by the LMDZ-iso model (Figure 4 and Table 1 ). »

Linbes 450-451: " They always occurred during low pressure periods (atmospheric pressure below 1005 mbar)." What is the synoptic situation leading to this low pressure and cold fronts?

>> We have removed the numerous references to cold fronts because it was confusing and we actually wanted to focus on the water vapor $\delta^{18}O_v$ excursions. We thus do not discuss the synoptic situations which is not covered by our study. We rewrote this part as :

« «These negative $\delta^{18}O_v$ excursions always occurred during low pressure periods (atmospheric pressure below 1005 mbar) and we observe the presence of a cold front in a distance of 100 km around Amsterdam Island in a 48h period covering the time of the event (Supplementary Material Figure S1). »

Lines 501-503: "However, for the 11 events highlighted above, the d18Ov vs qv evolution follows an evolution characteristic of remoistening processes, i.e. a curve standing below the curve of the d18Ov vs qv evolution observed for the rest of the series…"
The single events show a much steeper evolution in the d18Ov-qv diagram than the remoistening curve. Why is this?

>> You are right. As mentioned as answer to the editor, this figure is a "first order" approach following previous study of Guilpart et al. (2017) but is not 100% appropriate since the simple modeling curves are idealized $\delta^{18}O_v$ vs $q_v$ trajectories. Even if we have adapted our initial conditions to the observation of the surface isotopic composition of the water vapor, we do not expect the curves to be aligned with our events and we simply use them at first approach to show that enhanced remoistening or water-rain interactions may be a good candidate to explain the $\delta^{18}O_v$ vs $q_v$ relationship during the events.

We propose to better explain as:
" Even if the water vapor $\delta^{18}O_v$ vs $q_v$ evolution is rather steep, there is some resemblance with the idealized theoretical curve for remoistening initially calculated for the free troposphere (Noone, 2012) and adapted here with initial conditions corresponding to the surface water vapor isotopic commposition. Even if the analogy between our measurements and this simple modeling approach should be taken with cautious, the fact that the water vapor $\delta^{18}O_v$ vs $q_v$ evolution lies below the idealized curve for condensation processes supports the depleting effect of vapor-rain interactions for our negative water vapor $\delta^{18}O_v$ excursions (Noone, 2012; Worden et al., 2007). "

Lines 504-506: "Since relative humidity is relatively high during these events (values given in Table 1 compared to a mean value of 77 %), it more likely reflects rain-vapor diffusive exchanges than rain evaporation." Are you referring to the relative humidity at the surface? How about relative humidity above that will also influence the interaction of the rain with its surroundings?

>> This is correct, we were referring to relative humidity at the surface. Actually, high relative humidity will favor the diffusive exchange between rain and vapor but we can not exclude reevaporation at the surface nor in the upper atmosphere since both can have an influence as shown in Figure 8 and in l. 585 – 588 of the previous manuscript. We thus propose the new text :

« Surface relative humidity remains relatively high during these events (values given in Table 1 compared to a mean value of 77 %) which favors rain-vapor diffusive exchanges. »

Line 519ff.: As the trajectories are only shown in the supplement, it is difficult to follow this paragraph. The beginning of the paragraph leads to think that the trajectories indicate that subsidence is import but in the end the conclusion is that "back trajectories are however not supporting systematic subsidence for other cases".

>> Indeed, the backtrajectories do not evidence systematic air subsidence for the water vapor $\delta^{18}O_v$ excursions and this is the reason why the figures were initially put in the supplement but we agree that it is better to have some of them in the main text. Actually, when looking in details the atmospheric dynamic, it is clear that the $\delta^{18}O_v$ excursions occur at the transition between ascendence and subsidence and this is probably the reason why we could not easily detect it from 10 days back-trajectories. This will be better explained in the text with back-trajectories of 5 days only (cf next answer to comment).

Lines 526-527: "... the maximum altitude of the envelope of the back trajectories increases from 5,000 to 8,000 m..."

What it the mean/standard deviation of the maximum trajectory height? How many days before arrival are the trajectories at their maximum height? How relevant is this for the isotopic composition upon arrival? E.g. if the trajectories descend over the ocean and take up moisture, their maximum height before the moisture take-up is less relevant for the isotopic composition at Amsterdam Island. If you are using the full 10-days backward trajectories to calculate the maximum altitude, I don't think that the maximum altitude is a good measure of subsidence in front of the cold front.

>> Actually, what we wanted to test initially was if there is a change of the origin of the air mass during the excursion which may explain different isotopic signature of the water vapor. To answer the questions listed here, we have developed a new representation of the back-trajectories on only 5 days and with the indication of the location of the average (using humidity weighting) back-trajectory. We propose also to better explain that back-trajectories are used to mainly study the change of air origin.

The new paragraph in the main text reads :
« To further explore the processes leading to the decoupling of humidity and $\delta^{18}O_v$ as well as sharp negative excursions of $\delta^{18}O_v$ during the 11 events identified here, we also use information from the ERA5 reanalyses. In particular, the influence of atmospheric circulation (vertical and horizontal advection) and moisture origin can be studied through back trajectories. The back trajectories performed over the 5 previous days (Figures 7 and A3) confirm the information from wind directions that there is no systematic change in the horizontal origin of the trajectories for the different events. No systematic pattern is also identified in the vertical advection even if we note that for the event of the 3rd of January, the average altitude of the envelope of the 5-day back trajectories increases when comparing the situation before the excursion and the situation during the most negative water vapor $\delta^{18}O$ value. This observation may support the occurrence of air subsidence indicated by the GEM record on this particular event (Figure 7). "

Figure 7 is the following:

«

[Figure]

**Figure 7** : FLEXPART footprints of 5-day back trajectories for the event of the 3rd-4th of January. (a) Latitude-longitude projection of the FLEXPART back trajectory footprint for the 3rd of January 2020 at 13h30. The yellow to green colors on each grid point of these projections represent the density of particles. The white to blue colors indicate the water vapor mixing ratio on the humidity weighted average back-trajectory. Each red point indicates the location of the average back-trajectory for each of the 5 days before the date of the considered event. (b) Same as a for the 3rd of January 2020 at 22h30. (c) Top shows the evolution of the water vapor mixing ratio of the back trajectories for the 3rd of January 2020 at 13h30; bottom shows the altitude evolution of the back trajectory for the 3rd of January 2020 at 13h30. (d) same as (c) for the 3rd of January 2020 at 22h30. »

Lines 543-555: " There is no evidence for changes in the horizontal advection of air over the 11 particular events from the observation of wind direction around these cold front events."

How is the cold front identified? Does it divide different air masses? A cold front normally implies a horizontal transport of air, why is this not the case for these cold fronts?

>> The cold front were initially identified with the synoptic weather charts as explained in answers to other comments. As mentioned elsewhere, the reference to these cold fronts are muted in the new version and we have removed « around these cold front events » from this particular sentence.

Lines 556-558: "Such abrupt d18Ov events can hence be used as a test of the performances of general circulation models equipped with water isotopes."

What was learned about the performance of the GCMs involved in this study? Was it necessary to include d18O in such a performance test instead of just using traditional humidity variables (e.g. relative humidity, specific humidity, precipitation)?

>> You are right that many skills could already be tested using only meteorological data. This is the reason why we focused here only on the periods when water vapor $\delta^{18}O_v$ was showing a different signal than the one inferred from humidity to study what we can learn from this signal. Also, the combination with Hg measurements suggesting subsidence was useful to complement the traditional variable records over these excursions. Finally, because more and more models are equipped with water isotopes, we wanted to show an example of the peculiar signal seen in the water isotopes and not in traditional humidity variable to test the skills of these models, both on the implementation of water isotopes and on the dynamic of the atmospheric water cycle.

We propose this new formulation to take into account this comment :

« Still, the fact that at least ECHAM6-wiso is able to reproduce every negative $\delta^{18}O_v$ excursions (whether they are associated or not with subsidence or rain-water vapor reequilibration) shows that not only the patterns of atmospheric water cycle are correctly reproduced (a validation which could also be performed using humidity and precipitation data) but also that the isotopic processes are correctly implemented in this model. Such abrupt $\delta^{18}O_v$ events can hence be used as a test bed of the skills of water isotopes enabled general circulation models.«

Line 562: What is "SOM"?

>> Supplementary Online Material, it is not in the new version.

Lines 559-584: As both isotope-enabled models were nudged to ERA5 dynamics, it is to be expected that the GCMs reproduce the ERA5 reanalysis wind fields rather well with some caveats due to the lower horizontal resolution. This paragraph (and Fig.7) is mostly describing the smoothing of ERA5 due to the coarser resolution of the isotope-enables GCMs. Why do we need isotope measurements to see the effect of a coarser horizontal resolution? What do we learn about the GCM performances by decreasing the horizontal resolution?

>> It is true that we do not need isotopes to show the effect of decreasing resolution and we have removed the sentences which may convey this idea (especially the first sentence of the paragraph). Still, the idea with the use of water isotopes is to have a record at the surface of what is happening above in the atmosphere (without relying on reanalyses only) and this is the reason why we think that they have an added value to a pure comparison with reanalyses.

In addition, testing the different resolutions shows that good horizontal resolution is essential to correctly simulate front dynamics (as was already pointed by Ryan et al. 2000) and associated isotopic variations.

Ryan, B. F., and Coauthors, 2000: Simulations of a Cold Front by Cloud-Resolving, Limited-Area, and Large-Scale Models, and a Model Evaluation Using In Situ and Satellite Observations. *Mon. Wea. Rev.*, **128**, 3218–3235

We propose the modified paragraph :

« To further explore the $\delta^{18}O_v$ data-model comparison and associated processes, we compare the skills of the ECHAM6-wiso and the LMDZ-iso models over the first months of 2020 in term of atmospheric dynamic (the whole series is displayed in Figure A1). First and as expected because of the nudging, the two models reproduce rather well the evolution of the vertical velocity from the ERA5 reanalyses with a stronger ascent for the model predicting the strongest precipitation amount (e.g. LMDZ-iso for 24$^{th}$ of January 2020). The event of the 3$^{rd}$ of January is the only one reproduced by both ECHAM6-wiso and the two versions of the LMDZ-iso model: the three simulations show a clear subsidence over the isotopic event and a clear negative $\delta^{18}O_v$ excursion (Figure 4). For the other events, neither LMDZ-iso nor ECHAM6-wiso show a clear signal of subsidence neither at 500 nor at 850 hPa (Figure 4). However, the horizontal distribution of vertical velocity obtained with ECHAM6-wiso and LMDZ-iso are significantly different (Figure 7 for the event of the 9$^{th}$ of January, Figures S5 for the other events). While the LMDZ-iso modelled vertical velocity displays a rather strong homogeneity on the vertical axis, ECHAM6-wiso modelled vertical velocity highlights subsidence of air below the ascending column at the exact location of the negative $\delta^{18}O_v$ anomaly (Figure 7c). This subsidence of depleted $\delta^{18}O_v$ below the ascending column is responsible for the sharp negative $\delta^{18}O_v$ excursion in the ECHAM6-wiso model. The fact that subsidence of air occurs just below uplifted air, at the limit between ascendance and subsidence (Figure 7j), permits to reconcile the GEM data suggesting subsidence and the sign of the vertical velocity of the ERA5 reanalyses at Amsterdam Island. Since the isotope implementation was done similarly in the two models, the reason why the LMDZ-iso model does not reproduce the water isotopic anomaly is its too coarse resolution as also supported by the comparison between skills of the LMDZ-iso model at low resolution and very low resolution for the event of the 24$^{th}$ of January (Table 1 and Figure 4). As already pointed by Ryan et al. (2000), a fine resolution is necessary to correctly simulate front dynamics and we extend this result here to the high resolution temporal patterns of surface $\delta^{18}O_v$. "

Lines 586-590: "A rain event is associated with a strong ascending column in which d18Ov is depleted by progressive precipitation during the ascent and by interaction between rain and water vapor. This ascending column is coupled to the subsidence of d18Ov depleted air at the rear of the event which is pushed toward Amsterdam Island through a south west advection of cold air."

How is the isotopic composition of the subsiding air behind the cold front connected to the progressive precipitation during the ascent? Can the ascending and descending column be differentiated in the d18O excursions?

>> This is a very valid question which is not so simple to answer with the model outputs since we do not have water tagging. Still, we did a simple analysis using the outputs of the ECHAM6-wiso model looking at the water vapor $\delta^{18}O_v$ vs $q_v$ in front of the event (ascending column), during the event and at the rear of the event (subsiding column) (Figure R6).

[Figure]

**Figure R6**: (left) Evolution of the water vapor $\delta^{18}O_v$ vs $q_v$ in ECHAM6wiso, at the surface at the nearest grid cell to Amsterdam island for all time steps between 01/01/2020 and 31/03/2020 (grey dots), at the surface for event 3 (24th of January 2020) at the latitude of Amsterdam and for all longitudes between 50°E and 100°E (black dots), and for three atmospheric columns at the time of the event (plain colored lines). The three vertical atmospheric columns are taken as best representations of the situation before the water vapor $\delta^{18}O_v$ anomaly of the 24th of January 2020, as an example. Ascending column is represented by the vertical atmospheric column at 81.6°E (upstream of / before the anomalous $\delta^{18}O_v$ event, green line); the situation during the event can be visualized as the vertical atmospheric column at 77.8°E, orange line; the vertical atmospheric column after the event (downstream) can be visualized by the column at 73.1°E, blue line. Vertical velocity directed downward is represented by a thick line (only present for the blue line). The black line indicates the distribution $\delta^{18}O_v$ vs $q_v$ for marine mixing and the dashed lines show Rayleigh distillation distributions. (right) as in Figure 7b of the article, but for event 3, and showing the location of the three atmospheric columns with same color lines as in the right.

The period before the $\delta^{18}O_v$ anomaly corresponds in most cases to the end of the rain event. This period is associated with a strong lift of the moist air in which we see a water vapor $\delta^{18}O_v$ vs $q_v$ distribution (low $\delta^{18}O_v$ with high humidity, green curve on Figure R6) which looks like an extreme case of remoistening (super Rayleigh as described in the Figure 6 of the main manuscript). We can

interpret this as ongoing rain-steam exchanges. The situation is different from what happens at the surface where water vapor $\delta^{18}O_v$ vs $q_v$ is not showing any anomalous behavior yet.

During the event (orange curve on Figure R6), the water vapor $\delta^{18}O_v$ vs $q_v$ evolution is completely vertical and really difficult to be explained by only remoistening effect. We thus believe that a dynamic aspect (mixing) is also involved in bringing in the surface boundary layer low water vapor $\delta^{18}O_v$ with relatively high humidity.

After the event (blue curve on Figure R6), we are back to a classic Rayleigh situation, so the water vapor $\delta^{18}O_v$ returns to its initial value.

We did not add anything on this analysis in the new version but if the reviewer and the editor thinks that this is an interesting added value, we propose to ad dit in the supplementary.

Line 621: "hours/days". Is there a cold front passage that has a duration of several days?

>> You are right, we do not have cold fronts lasting several days over the Amsterdam Island. As noted in previous answers to comments, we did put too much emphasis on cold fronts which was misleading. We removed "associated with cold fronts" on l. 623.

Lines 635-640: "This study highlighted the added value of combining different data from an atmospheric observatory to understand the dynamic of the atmospheric circulation. The two-year records are also a good benchmark for model evaluation. We have especially shown that the isotopic composition of water vapor is a powerful tool to identify aspects to be improved in the general circulation models, such as the horizontal resolution which may influence the representativity of the vertical dynamics."

As also mention above, why are stable water isotopes needed to show that the horizontal resolution may influence the vertical dynamics? The vertical cross sections of vertical wind speed (Fig. 7) illustrates this already quite well.

>> This comment has been addressed in the list of major comments above.

on the Zenodo platform, and the dataset does not include the water vapour mixing ratio.

>>> We will complete the dataset with a version 2 after acceptation of the manuscript including water vapour mixing ratio.

**Figures**:

General: The figures are often difficult to read, especially the described phenomena are small (e.g. d18O excursions of a few hours in a 2-year or 3 month timeline). Additionally, the colors are not color-blinded friendly and the caption are not concise.

>> We agree that it is difficult to see the excursions on a 2-year timeline and this is the reason why it has been decomposed in several figures of 3-months (now in appendix, see answer to comment above). In addition, some excursions are detailed in the main text (and all are now provided in the

appendix, see comment above). Figure 4 (and corresponding figures in appendix) is now color-blinded friendly and the captions concise (see below for details).

Fig.1: Is this figure needed? Fig.1 is not mentioned in the text.

>> We mention it in the text at first instance.

« In order to complete the picture of the atmospheric water cycle over the Indian basin of the Southern Ocean already measured by these three analyzers, we installed a new water vapor isotopic analyzer in the mid-latitude of the south Indian Ocean on Amsterdam Island (Figure 1) in November 2019. »

Fig.3: x-axis too coarse, light green shading difficult to see.

We change to the following figure:

[Figure]

Fig. 4: Colors red/blue/green are not colorblind friendly. The caption text includes many repetition and should be improved. A legend in the figure could improve the readability.

>> We propose this modified figure (new color code) which will also enable to shorten the caption (since information is on the legend in the figure):

[Figure]

On this panel, we also removed the vertical velocity at 500 hPa from panel c to improve readability and because it did not bring much additional information (the information is given in Table 1).

New caption :

« **Figure 4**: Model-measurement comparison (January – March 2020); a- $\delta^{18}O_v$ (light blue for data on hourly average, dark blue for data resampled at a 6-hour resolution); b- water vapor mixing ratio from our data set; c- vertical velocity; d- Precipitation amount. The grey rectangles highlight the negative $\delta^{18}O$ excursions (note that in this figure the excursions of the 3[rd] and 9[th] of January 2020 are distinct while the distinction could not be done on Figure 3 because of the scale).”

Fig. 6: What is $\phi=0.025$?

This value is indicated together with the "remoistening curve" which has been obtained following the expression in Noone (2012). Basically, the idea is to express remoistening through a modification of the equilibrium fractionation coefficient between water vapor and rain ($\alpha_e$) so that the effective fractionation factor will be $\alpha=(1+\phi)\times\alpha_e$

This effective fractionation coefficient is then introduced in the Rayleigh distillation equation to deduce the link between $\delta^{18}O_v$ and mixing ratio:

$(\delta-\delta_0)=(\alpha-1)*\ln(q/q_0)$

We provide this information in the revised version of the manuscript as :

« Remoistening is described through a modification of the equilibrium fractionation coefficient between water vapor and rain ($\alpha_e$) so that the effective fractionation factor is $\alpha=(1+\phi)\times\alpha_e$. This effective fractionation coefficient is then introduced in the Rayleigh distillation equation to deduce the link between $\delta^{18}O_v$ and mixing ratio as:

$$(\delta^{18}O_v-\delta^{18}O_{v,0})=(\alpha-1)\times\ln(q_v/q_{v,0}) \qquad \text{(Eq 8)''}$$

Figure 8: What is SBL? The ascent of air in front of the cold front rises nearly vertical at a constant longitudinal position. As a cold front is moving system (mostly associated with an extratropical cyclone), the ascent does not occur at a constant location (in latitude and longitude). Further, all precipitation seems to fall in front of the cold front, which is unlikely.

SBL is Surface Boundary Layer, it has been changed to Marine Boundary Layer.

We agree that we need to include a more detailed analysis of the synoptic situation during the events. In Figure R4 shown above, we show front location, precipitation and 850 hPa vertical velocities from ERA5 at the time of the events. Precipitation generally falls just ahead of the cold front.

Figure 8 (now Figure 9) scheme is based on the profiles modelled by ECHAM for event of the 9th of January 2020. We show these profiles below (Figure R7). We now make this clear in the new legend to Figure 9 and add these profiles to the Supplement (they share similar patterns for all events).

[Figure]

[Figure]

**Figure R7 (Figure S5 of the Supplementary Material)**: ECHAM6wiso profiles during event of the 9th of January 2020, used to design Figure 8 of the initial manuscript.

New caption for Figure 8 (now figure 9):

« **Figure 9:** Scheme of the mechanism explaining the sharp negative excursion of $\delta^{18}O_v$ recorded at the surface for cold front events associated with precipitation. The scheme is based on the profile modelled by ECHAM6-wiso for event of the 9th January 2020. The top panel show the altitude vs longitude dynamics of air masses with vertical saturated lifting in the center and subsidence at the rear of the lifting. The bottom panel shows the associated evolution of $\delta^{18}O_v$ and precipitations on the same longitude scale than on the upper panel. "

How is the subsidence at 100°E and ascent at 90°E related to the cold front?

The subsidence at 100°E seems to be linked to background conditions, while ascent at 90°E is caused by the cold front moving eastwards, causing precipitations just ahead of the front (Figures R4 and R7)

What does a composite of precipitation and d18Ov for all cold front events look like?

>> We already tried to make a composite before the first version of the manuscript but because the water $\delta^{18}O_v$ anomalies have different amplitude and durations (Table 1) and precipitation amount is very different from one event to the other (some event being associated with no rain, Table 1), a composite does not show any useful additional value.

Can it reproduce the schematic as shown in the "surface box"?

Because it is not possible to make a composite, the surface box of the schematic follows the surface state modeled by ECHAM for event of the 9th of January 2020 (Figure R7, lower right plots).

We propose this new version for the Figure 8:

[Figure]

**Technical comments**

Generally: there are many abbreviations in the text that are only used a few times. Can you reduce the number of abbreviations?

>> This is true and has been modified in the revised version.

Line 60-61 (and many others): The references are not in chronological order.

>> We changed it.

Line 65-66: "We express the abundance of the heavy isotopes D and 18O with respect to the amount of light isotopes 16O and H in the water molecules…" should be "We express the abundance of the heavy isotopes D and 18O with respect to the amount of light isotopes **H and 16O, respectively,** in the water molecules…"

>> This has been modified

Line 68: Eq. 1 has strange symbols (squares).*

>> It is a problem from the word to pdf conversion, it can not be changed at this stage but should be possible when finalising the paper for production.

Line 88-89: "water cycle processes such as water cycle processes such as "

>> This has been corrected

Lines 106-109: "Over the previous years, we have installed 3 water vapor analyzers on Reunion Island at the Maido observatory (21.079°S, 55.383°E, 2160m) (Guilpart et al., 2017) and in Antarctica (Dumont d'Urville and Concordia; (Leroy-Dos Santos et al., 2021; Breant et al., 2019; Casado et al., 2016). " Check usage of brackets.

>> This has been corrected

Line 133: "from the nearest lands, Madagascar**,** and"

>> It has been corrected

Lines 140-141: "…and were continuously monitored at the site **since** 1960…"

>> This has been corrected

Lines 145-150: The section is very difficult to read, the websites and datasets should better be included as references. Same for link to AERIS on line 178 and 200.

>> datasets are now only included in the section « data availability » and we removed useless reference to program not used in this study.

Lines 180-181: "instrument models (Tekran Inc., Toronto, Canada) (Angot et al., 2014; Slemr et al., 2015, 2020; Sprovieri et al., 2016; Li et al., 2023). "

>>> This is simplified in the new version.

Line 202: "may possibly" Doubling, omit either.

>> We have removed "possibly"

Lines 211-221: This sentence is too long. Can you divide into two sentences?

>>> Sure, we have this new formulation:

"Chemical cycling and spatiotemporal distribution of mercury in the air is still poorly understood whatever atmospheric layer considered (surface, mixed or free troposphere, stratosphere), and complete GEM oxidation schemes remain still unclear (Shah et al., 2021 and associated references). Still, several studies provided evidence that vertical distribution of atmospheric mercury measurements from boundary layer to lower/upper troposphere and stratosphere shows a decreasing trend in GEM concentration with increasing altitude, in parallel with an increase in the concentration of divalent mercury (GOM + PBM) resulting from GEM oxidation mechanisms (Murphy et al., 2006 ; Swartzendruber et al., 2006, 2008 ; Talbot et al., 2007 ; Fain et al., 2009 ; Sheu et al., 2010 ; Lyman and Jaffe, 2012 ; Brooks et al., 2014 ; Fu et al., 2016 ; Koenig et al., 2023). "

Line 228: "The isotopic composition of near-surface water vapor (d18Ov and dDv in ‰ versus SMOW)"

>> It has been modified

Line 242: "The calibration of the data is performed in different**several** steps following previous studies"

>> It has been modified

Lines 301-303: "…identical **to** the atmospheric setup of IPSL-CM6A (Boucher et al., 2020) used for phase 6 of the Coupled Model Intercomparison Project (CMIP6, (Eyring et al., 2016)). "

Line 373: "… very close to the one observed in Angot et al. (2014)."

Line 374: "During the period (2020-2021) of water vapor isotope measurements in AMS…"

Do you mean: During the period 2020-2021 of water vapor isotope measurements in AMS.. ?

Lines 390-391: "The annual cycles are also not visible…" Do you mean: "An annual cycle is not visible…" ?

---

## Referee Report (RR1)

Review egusphere-2023-1617

**Abrupt excursion in water vapor isotopic variability during cold fronts at the Pointe Benedicte observatory in Amsterdam Island**
*Amaëlle Landais,\*, Cécile Agosta,\*, Françoise Vimeux, Olivier Magand, Cyrielle Solis, Alexandre Cauquoin, Niels Dutrievoz, Camille Risi, Christophe Leroy-Dos Santos, Elise Fourré, Olivier Cattani, Olivier Jossoud, Bénédicte Minster, Frédéric Prié, Mathieu Casado, Aurélien Dommergue, Yann Bertrand, Martin Werner*

Thank you for the detailed replies. Many of my concerns have been addressed and I think that this manuscript fits to ACP, especially when highlighting that the combination with mercury measurements allows for new conclusions regarding vertical air movements. I think that in the new manuscript, this message is improved. But mercury is only marginally mentioned in the introduction. There are many explanations in the data section which might better motivate the study if included in the introduction.

I still have some concerns regarding the interpretation of the cold front dynamics and the figures highlighting these dynamics. Even though the term "cold front" is less dominant throughout the manuscript, the cold front dynamics are still prominent in the conclusion and synthesis figure. This makes sense as most of the d18O excursions are related to the passage of a cold front. But Figures 8, 9, S2 and S3 show a very wide longitudinal range that does not allow to see features along the cold fronts. Some detailed comments on this issue in the following:

- why is is LMDZiso-VLR only reproducing d18O excursion on 3 Jan, what is special about this event?
- Figure R6: Thanks for this analysis! *In front of* and *during event* seem to be in a very similar dynamic environment (ascent; both during precipitation?) and show a similar evolution in Fig R6a. While *after event* shows a very different evolution. So, could this mean that *during event* is an enhanced signal of the *before event* at the rear of the precipitation event?
- The locations where you chose before, during and after event in Fig. R6 are within 10° around AMS. Why do you show a 50° (or 60°) window for Fig. 8, 9, S2 and S3 if the relevant processes occur within these 10°? This aspect is mentioned again for several of the following points.
- Lines 573-575: *"However, we note that when negative d18Ov excursions are not concomitant with subsidence, they occur right after an ascending movement and are generally followed by subsidence (Figures A1 and A2)."*
  What does "after" and "generally followed" mean? It seems that this is no longer referring to the trajectory calculations. Does this mean that large-scale subsidence (as represented by the trajectories) is not important?
- Lines 601-604: *"While the LMDZ-iso modelled vertical velocity displays a rather strong homogeneity on the vertical axis, ECHAM6-wiso modelled vertical velocity highlights subsidence of air below the ascending column at the exact location of the negative d18Ov anomaly (Figure 8c)."*
  Which subsidence below the ascending colum do you mean? Do you mean the strong subsidence behind the cold front between 65-75°E? This does not correspond with a d18Ov excursion at the surface. The x-axis scale makes it difficult to see these small feature at the AMS location.
- Lines 605-609: *"The fact that subsidence of air occurs just below uplifted air, at the limit between ascendance and subsidence (Figure 8j and Supplementary Material Figure S4),*

*permits to reconcile the GEM data suggesting subsidence and the sign of the vertical velocity of the ERA5 reanalyses at Amsterdam Island."*
I don't understand this sentence. What do you mean with "permits to reconcile"?

- Lines 621-623: *"This ascending column is coupled to the subsidence of d18Ov depleted air at the rear of the event, which is pushed toward Amsterdam Island through a south west advection of cold air."*
  What do mean with "coupled"? I don't understand what you mean with subsidence and south west advection. Do you refer to large-scale advection within the cold sector? Is the horizontal advection an important process for the d18Ov excursions? This has not been mentioned so far. Also, the trajectory analysis did not show any important signals from large-scale advection for the selected events.

- Fig. 8
  - The cold front appears as a vertical line due to the large longitudinal window. Therefore, no typical features along the cold front can be seen.
  - Isentropes in Fig.8 could help to see the cold front in vertical profiles
  - Why is subsidence > 10° away from the front important for the isotopic signature during front passage? The surface isotopic composition between 60 and 75° in Fig. 8 b,e,h shows a distinctly higher signal than the water vapour above ~2km and at the AMS position. Is it important to show this to understand the processes leading to the d18Ov excursions? I recommend to choose a smaller window around the cold front for Fig.8.

- Fig.9 does not help to understand the described processes leading to the d18Ov excursion. New phrases are mentioned (e.g moist and dry subsidence, marine boundary layer) but they were not introduced in the manuscript in the context of the d18Ov excursions. It is not evident from the manuscript why processes more than 10° away from the front are important for d18Ov excursion.

Minor comments:

- Lines 597/598 state *"For the other events, neither LMDZ-iso nor ECHAM6-wiso show a clear signal of subsidence neither at 500 nor at 850 hPa (Figures 4 and A1)."*
  Neither captions of Fig 4 nor A1 state at which level the vertical velocity is shown.

- 582-586: *"Still, the fact that at least ECHAM6-wiso is able to reproduce every negative d18Ov excursion (whether they are associated or not with subsidence or rain- water vapor reequilibration) shows that not only the patterns of atmospheric water cycle are correctly reproduced (a validation which can also be performed using humidity and precipitation data) but also that the isotopic processes are correctly implemented in this model."*
  Not all aspect of the atmospheric water cycle can be assessed with humidity and precipitation data only, e.g. the residence time of water in the atmosphere cannot be seen with a precipitation field, but can be traced with isotopes. This is one of the strength of an isotope measurements and isotope-enabled models.

- Lines 659-660: *"They are most of the time characterized by a decrease in water vapor mixing ratio. "*
  There is an increase in qv during the d18Ov excursions in Fig.4.

- Lines 673-674: *"This study highlights the added value of combining different data from an atmospheric observatory to understand the dynamics of the atmospheric circulation."*
  This is a very broad statement? Can you be more specific what you highlight about "the dynamics of the atmospheric circulation"?

- Lines 675-677: *"We have especially shown that the isotopic composition of water vapor measured at the surface is a powerful tool to identify aspects to be improved in the atmospheric component of the Earth system models. "*
  Which aspect of the atmospheric component of Earth System models should be improved according to this study? I would say that different model setups have been used but the atmospheric component of the models stayed the same.
- Please, check again the chronological order of the references in the text.
- Fig. S4: check caption.

---

## Author Response (AR2)

Answer to Reviewer 1.

Many thanks for the additional comments. We have addressed the different comments as detailed below.

Thank you for the detailed replies. Many of my concerns have been addressed and I think that this manuscript fits to ACP, especially when highlighting that the combination with mercury measurements allows for new conclusions regarding vertical air movements. I think that in the new manuscript, this message is improved. But mercury is only marginally mentioned in the introduction. There are many explanations in the data section which might better motivate the study if included in the introduction.

>> We have added an additional sentence in the introduction :
« Indeed, previous studies have shown that gaseous elemental mercury decreases with increasing altitude in marine environment suggesting that gaseous elemental mercury can be used as a tracer of subsidence of air from the high altitude (e.g. Koening et al., 2023). »

I still have some concerns regarding the interpretation of the cold front dynamics and the figures highlighting these dynamics. Even though the term "cold front" is less dominant throughout the manuscript, the cold front dynamics are still prominent in the conclusion and synthesis figure. This makes sense as most of the d18O excursions are related to the passage of a cold front. But Figures 8, 9, S2 and S3 show a very wide longitudinal range that does not allow to see features along the cold fronts.

We agree that it makes sense to zoom closer to the front location and we modified Figures 8, 9, S2 and S3 accordingly. As fronts are usually oriented in the North-West/South-East direction, we also changed the cross sections to be oriented perpendicular to this direction, as shown below in new Fig. 8 and in new Fig. S1.

[Figure]

[Figure]

**Figure 8:** Modelled $\delta^{18}O_v$ and vertical velocity for the event of January 9th 2020. (a) Surface air $\delta^{18}O_v$ (~83 m, latitude vs longitude), with yellow line indicating -15 ‰ contour level and grey lines indicating precipitation contours at 0.5, 10, and 50 mm day$^{-1}$ (thin, medium and thick lines respectively); (b) $\delta^{18}O_v$ plotted on a vertical cross-section (altitude vs longitude) along the transect indicated by the white line on panel (a), with yellow lines indicating $\delta^{18}O_v$ contours at -30 ‰ and -15 ‰, blue lines indicating the contour of –0.05 Pa s$^{-1}$ vertical velocity (ascendance), and the vertical black line denoting the longitude of Amsterdam Island; (c) Vertical velocity plotted on a vertical cross-section as for (b), with same contour lines. (a), (b) and (c) are drawn using outputs of the ECHAM6-wiso model; (d), (e) and (f) are the same as (a), (b) and (c) but obtained from the LMDZ-iso model at low resolution (LR); (g), (h) and (i) are the same as (a), (b) and (c) but obtained from the LMDZ-iso model at very low resolution (VLR). (j) ERA5 air temperature at 850 hPa, with white lines marking front locations (see Supplementary Material S1); (k) ERA5 vertical velocity plotted on a vertical cross-section (altitude vs longitude) along the transect indicated by the black dotted line on panel (j)."

| Date of events | (a) Air temperature at 850 hPa | (b) Precipitation |
|---|---|---|
| 03/01/2020 | | |
| 09/01/2020 | | |
| 24/01/2020 | | |
| 04/03/2020 | | |

[Figure]

| | |
|---|---|
| 10/05/2020 | |
| 09/08/2020 | |
| 08/03/2021 | |
| 07/06/2021 | |
| 23/06/2021 | |

[Figure]

| | |
|---|---|
| 08/11/2021 |
[Figure]
 |
| 06/12/2021 | |

**Figure S1:** Synoptic analysis using hourly ERA5 fields at the time of observed minimum $\delta^{18}Ov$ corresponding to the 11 events identified in the manuscript: (a) air temperature at 850 hPa, (b) precipitation, and (c) vertical velocity at 850 hPa. White and black lines represent frontal passage, located at the maximum gradient of 850 hPa potential temperature. Front is computed as the zero-line of the gradient of the magnitude of the gradient of 850 hPa air temperature, when the gradient of 850 hPa air temperature is greater than 2 K/100 km, following Schemm et al. (2015). The black dotted line shows the transect location used for the vertical cross-section in Figure S3.

| Date of events | LMDZ6-VLR (~3°) | LMDZ6-LR (~2°) | ECHAM6 (~1°) |
|---|---|---|---|
| 03/01/2020 |
[Figure]
 | | |
| 09/01/2020 | | | |

[Figure]

| | | | |
|---|---|---|---|
| 23/06/2021 | | | |
| 08/11/2021 | | | |
| 06/12/2021 | | | |

[Figure]

**Figure S2:** δ¹⁸Ov plotted on a vertical cross-section (altitude vs. longitude) as modeled by LMDZ6 at very low resolution (left), low resolution (middle) and ECHAM6-wiso (right). Location of the extracted transects are indicated by the white line in Fig. 8a) for ECHAM6wiso, 8d) for LMDZ6iso-LR and 8g) for LMDZ6iso-VLR. Yellow contours indicate –30‰ (upper) and –15‰ (lower) contours of surface δ¹⁸O$_v$. Black contours indicate contours of –0.05 Pa s$^{-1}$ vertical velocity (ascendance). The vertical black line denotes Amsterdam Island latitude.

| Date of events | LMDZ6-LR (~2°) | ECHAM6 (~1°) | ERA5 (~0.25°) |
|---|---|---|---|
| 03/01/2020 | | | |

[Figure]

| | | | |
|---|---|---|---|
| 09/01/2020 | |
[Figure]
 | |
| 24/01/2020 | | | |
| 04/03/2020 | | | |
| 10/05/2020 | | | |
| 09/08/2020 | | | |
| 08/03/2021 | | | |

[Figure]

**Figure S3:** Vertical velocity plotted on a cross section of longitude (x) versus altitude (y) at the Amsterdam latitude as modeled by LMDZ-iso at low resolution (1st column), ECHAM6-wiso (2nd column) and ERA5 (3rd column). Location of the extracted transects are indicated by the white line in Fig. 8d for LMDZ6iso-LR, Fig. 8a for ECHAM6wiso and Fig. 8j or S1 for ERA5. Yellow contours indicate –30‰ (upper) and –15‰ (lower) contours of surface $\delta^{18}O_v$. Black contours indicate contours of –0.05 Pa s$^{-1}$ vertical velocity (ascendance). The vertical black line denotes Amsterdam Island latitude.

Some detailed comments on this issue in the following:

why is is LMDZiso-VLR only reproducing d18O excursion on 3 Jan, what is special about this event?

The reason for that is probably that this event is associated with a strong subsidence, stronger than for other events (see Figure 5).

Figure R6: Thanks for this analysis! In front of and during event seem to be in a very similar dynamic environment (ascent; both during precipitation?) and show a similar evolution in Fig R6a. While after event shows a very different evolution. So, could this mean that during event is an enhanced signal of the before event at the rear of the precipitation event?

Many thanks for this comment. We agree with this interpretation that, during the event, we see an enhanced signal of the situation before the event at the rear of the precipitation event. However, it is difficult to go further without additional tools, such as water isotope tagging analyses. We thus prefer not speculating more on this in the article.

The locations where you chose before, during and after event in Fig. R6 are within 10° around AMS. Why do you show a 50° (or 60°) window for Fig. 8, 9, S2 and S3 if the relevant processes occur within these 10°? This aspect is mentioned again for several of the following points.

We agree and we have modified all figures accordingly.

Lines 573-575: "However, we note that when negative d18Ov excursions are not concomitant with subsidence, they occur right after an ascending movement and are generally followed by subsidence (Figures A1 and A2)."
What does "after" and "generally followed" mean? It seems that this is no longer referring to the trajectory calculations. Does this mean that large-scale subsidence (as represented by the trajectories) is not important?

Indeed, this paragraph is based on the ERA5 vertical velocity as mentioned at the beginning of the paragraph. We have rephrased the sentence to make it more clear:

« However, we note that when negative $\delta^{18}O_v$ excursions are not concomitant with subsidence, they occur at the end of an ascending movement which is generally followed by subsidence (Figures A1 and A2). »

Lines 601-604: "While the LMDZ-iso modelled vertical velocity displays a rather strong homogeneity on the vertical axis, ECHAM6-wiso modelled vertical velocity highlights subsidence of air below the ascending column at the exact location of the negative d18Ov anomaly (Figure 8c)."
Which subsidence below the ascending column do you mean? Do you mean the strong subsidence behind the cold front between 65-75°E? This does not correspond with a d18Ov excursion at the surface. The x-axis scale makes it difficult to see these small feature at the AMS location.

We agree it was not clear, we wanted to highlight subsidence between 75°E and 77°E, just below the ascending column. We made a zoom on Fig. 8 and we changed the text to clarify this.

" While the LMDZ-iso modelled vertical velocity displays a rather strong homogeneity on the vertical axis, ECHAM6-wiso modelled vertical velocity highlights subsidence of air below the ascending column, with the maximum of negative $\delta^{18}O_v$ anomaly at the surface located just at the limit between ascendance and subsidence (between 75°E and 77°E in Figure 8c)"

Lines 605-609: "The fact that subsidence of air occurs just below uplifted air, at the limit between ascendance and subsidence (Figure 8j and Supplementary Material Figure S4), permits to reconcile the GEM data suggesting subsidence and the sign of the vertical velocity of the ERA5 reanalyses at Amsterdam Island."
I don't understand this sentence. What do you mean with "permits to reconcile"?

The sentence has been rewritten as follow to clarify this point:
« The fact that subsidence of air occurs just below uplifted air, at the limit between ascendance and subsidence (Figure 8j and Supplementary Material Figure S2), permits to reconcile the GEM data suggesting subsidence and the sign of the vertical velocity of the ERA5 reanalyses at Amsterdam Island suggesting that many excursions start with ascendance. »

Lines 621-623: "This ascending column is coupled to the subsidence of d18Ov depleted air at the rear of the event, which is pushed toward Amsterdam Island through a south west advection of cold air."
What do mean with "coupled"? I don't understand what you mean with subsidence and south west advection. Do you refer to large-scale advection within the cold sector? Is the horizontal advection an important process for the d18Ov excursions? This has not been mentioned so far. Also, the trajectory analysis did not show any important signals from large-scale advection for the selected events.

We wanted to highlight that cold fronts are usually moving from south west to north east. Cold fronts are associated with ascendance in front of the front, and subsidence at the rear of the front. We changed the text to clarify this point:

« This ascending column is generally associated with a cold front moving from South-West to North-Est (Figure 8j and Supplementary Material S1), with subsidence and $\delta^{18}O_v$ depleted air at the rear of the front (Figure 8 and Supplementary Material S2 and S3) »

Fig. 8
◦ The cold front appears as a vertical line due to the large longitudinal window. Therefore, no typical features along the cold front can be seen.

We agree, it has been fixed.

◦ Isentropes in Fig.8 could help to see the cold front in vertical profiles
We decided not to add additional lines in this figure for clarity issues, but instead we performed all vertical cross sections in the South-West to North-East direction, generally orthogonal to the front direction, for a better visualization of the front features.

◦ Why is subsidence > 10° away from the front important for the isotopic signature during front passage? The surface isotopic composition between 60 and 75° in Fig. 8 b,e,h shows a distinctly higher signal than the water vapour above ~2km and at the AMS position. Is it important to show this to understand the processes leading to the d18Ov excursions? I recommend to choose a smaller window around the cold front for Fig.8.

We agree that the vertical structure of $\delta^{18}O_v$ just around Amsterdam is more important that the broader window initially shown, so we reduced the window to a +-10° window around Amsterdam island. The depletion is also marked at higher altitude than at the surface in ECHAM6wiso on Fig8, but it is variable depending of the events, so we decided not to comment on that in the article.

Fig.9 does not help to understand the described processes leading to the d18Ov excursion. New phrases are mentioned (e.g moist and dry subsidence, marine boundary layer) but they were not introduced in the manuscript in the context of the d18Ov excursions. It is not evident from the manuscript why processes more than 10° away from the front are important for d18Ov excursion.

You are right, we removed notions that are not introduced in the manuscript and were related to previous versions of the text. We zoomed the figure closer to the event as suggested, which indeed enables a more focused view on the depletion event at the surface. This scheme aims at summarizing many information from cross-sections into a synthetic figure, we hope it will help the reader better understand the link between water isotopes and vertical structure of the atmosphere.

You will find the updated Figure 9 below.

[Figure]

Minor comments:
• Lines 597/598 state "For the other events, neither LMDZ-iso nor ECHAM6-wiso show a clear signal of subsidence neither at 500 nor at 850 hPa (Figures 4 and A1)."
Neither captions of Fig 4 nor A1 state at which level the vertical velocity is shown.

Many thanks for noting this. Indeed, we simplified these figures following the comments of the previous round of reviews. So we removed now reference to Figures 4 and A1.

582-586: "Still, the fact that at least ECHAM6-wiso is able to reproduce every negative d18Ov excursion (whether they are associated or not with subsidence or rain- water vapor reequilibration) shows that not only the patterns of atmospheric water cycle are correctly reproduced (a validation which can also be performed using humidity and precipitation data) but also that the isotopic processes are correctly implemented in this model."
Not all aspect of the atmospheric water cycle can be assessed with humidity and precipitation data only, e.g. the residence time of water in the atmosphere cannot be seen with a precipitation field, but can be traced with isotopes. This is one of the strength of an isotope measurements and isotope-enabled models.

We modified the sentence as:

« Still, the fact that at least ECHAM6-wiso is able to reproduce every negative $\delta^{18}O_v$ excursion (whether they are associated or not with subsidence or rain-water vapor reequilibration) shows that (1) the patterns of atmospheric water cycle are correctly reproduced, a validation which can be performed using humidity and precipitation data for some aspects but benefits from water isotopes implementation for the residence time of water and (2) the isotopic processes are correctly implemented in this model. »

Lines 659-660: "They are most of the time characterized by a decrease in water vapor mixing ratio. " There is an increase in qv during the d18Ov excursions in Fig.4

There should be a misunderstanding since there is clearly a long term decrease of qv during the excursions, qv is lower after the excursion than before in Figure 4.
We propose the modified sentence:

« They are most of the time occurring during a decrease in water vapor mixing ratio. »

Lines 673-674: "This study highlights the added value of combining different data from an atmospheric observatory to understand the dynamics of the atmospheric circulation."
This is a very broad statement? Can you be more specific what you highlight about "the dynamics of the atmospheric circulation"?

>> We propose to be more specific with this modified sentence:

« This study highlights the added value of combining different data from a surface atmospheric observatory to understand the dynamics of the atmospheric circulation, e.g. subsidence in the higher atmosphere. »

Lines 675-677: "We have especially shown that the isotopic composition of water vapor measured at the surface is a powerful tool to identify aspects to be improved in the atmospheric component of the Earth system models. "
Which aspect of the atmospheric component of Earth System models should be improved according to this study? I would say that different model setups have been used but the atmospheric component of the models stayed the same.

We have modified the sentence to better reflect the results of our work:

« We have especially shown that the isotopic composition of water vapor measured at the surface is a powerful tool to test the vertical dynamic of atmospheric models and the implementation of water isotopes for those that are equipped with them. »

Please, check again the chronological order of the references in the text.

The chronological order of references have been checked

Fig. S4: check caption.

Indeed, the qv plot has been removed. The caption has been changed to:

« Figure S4: Surface signal ($\delta^{18}O_v$ on the top and precipitation on the bottom) as modeled by LMDZ-iso with very low resolution (left), low resolution (middle) and ECHAM6-wiso (right). »

Answer to editor comments:

Many thanks for these comments, there are addressed as detailed below.

P5, L115: "in the mid-latitude of the south Indian Ocean" -> please rephrase

>> This has been corrected as « at mid-latitude in the south Indian Ocean »

P5, L128: "for evaluation of to evaluate atmospheric components...." -> check sentence and correct

>> We removed « to evaluate »

P6, L155: change "Magand" to "from O. Mangand"

>>Done

P6, L155: correct parenthesis around reference of Angot et al.

>> Done

P7, L191: "In this study, and even if ......" -> please rephrase to improve readability

>> We replace « if » by « though »

P7, L193: add "air" after "upper troposphere"

>> Done

P7, L194: also here, add "air" after "marine boundary layer"

>> Done

P8, L214: Check sentence, somthing is missing here.

>> Rewritten as
« The identification of such observational processes (lower concentration of GEM in high-altitude air masses compared to those in the marine boundary layer ones) is used here to help

characterize possible intrusions of high-altitude air masses at the low altitude Pointe Benedicte observatory. »

P8, L220: I would suggest to put the text part following after "and" in an extra sentence for better readability.

>> Done

P9, L236: add here to where the instrument where send.

>> Done « to Amsterdam Island »

P13, L297-298: density probability -> vice versa? Probability density?

>> Changed

P12, L314: Is the doubling of "surface" correct here?

>> Indeed, we can remove one – done.

P12, L334: Move reference at the end of the sentence.

>> Done

P13, L35: one "the" obsolete.

>> Done

P18, L475: One full stop obsolete.

>> Done

P18, L476: we indicate this "-0" -> this is indicated in the table as "-0"

>>Done

P20, L502. was taken with -> rather "calculated with" or "has a"?
>> Changed to : « has a »

P21, L516: Sentence not clear, please check.
>> Changed to
« Although the evolution of the water vapor $\delta^{18}O_v$ vs $q_v$ is rather abrupt, there is a certain resemblance with the idealized theoretical remoistening curve initially calculated for the free troposphere (Noone, 2012) and adapted here with initial conditions corresponding to the isotopic composition of surface water vapor. »

P21, L528: Such -> This
>> done

P27, L664: With high resolution less events were reproduced? Didn't you state before the opposite?

>> I think that it is what we state – we reproduce only 1 event at very low resolution and 7 at low resolution (the « low » resolution model has a highest resolution than the « very low » one as described in the methods).

P32, L717: Do you mean with "a" panel (a)? Then you should write "(a)" instead of just "a".

>> Changed.